# STRATEGIST: SELF-IMPROVEMENT OF LLM DECISION MAKING VIA BI-LEVEL TREE SEARCH

**Jonathan Light**[1]  **Min Cai**[2]  **Weiqin Chen**[1]  **Guanzhi Wang**[5]  **Xiusi Chen**[3]
**Wei Cheng**[4]  **Yisong Yue**[5]  **Ziniu Hu**[5]
[1]Rensselaer Polytechnic Institute,  [2]Shenzhen University,
[3]University of California, Los Angeles,  [4]NEC laboratories America,
[5]California Institute of Technology
https://llm-strategist.github.io

## ABSTRACT

Traditional reinforcement learning and planning typically requires vast amounts of data and training to develop effective policies. In contrast, large language models (LLMs) exhibit strong generalization and zero-shot capabilities, but struggle with tasks that require detailed planning and decision-making in complex action spaces. We introduce STRATEGIST, a novel approach that integrates the strengths of both methods. Our approach leverages LLMs to search and update high-level strategies (as text), which are then refined and executed by low-level Monte Carlo Tree Search (MCTS). STRATEGIST is a generalizable framework to optimize the strategy through population-based self-play simulations without the need for any training data. We demonstrate the effectiveness of STRATEGIST in learning optimal strategies for competitive, multi-turn games with partial information, including Game of Pure Strategy (GOPS) and multi-agent, hidden-identity discussion games like The Resistance: Avalon. Our results show that agents equipped with STRATEGIST outperform those trained with traditional RL methods, other LLM-based skill acquisition techniques, pre-existing LLM agents across both game environments and achieves comparable performance against human players.

## 1 INTRODUCTION

Recent studies have shown the potential of Large Language Models (LLMs) for learning skills and improving decision-making in interactive environments Wang et al. (2022; 2023a; 2024); Xi et al. (2023); Zhao et al. (2024); Liu et al. (2023). However, adversarial, multi-agent environments present a significant challenge. LLMs must reason about opponent actions, plan ahead strategically, and navigate a vast policy space Kambhampati et al. (2024); Valmeekam et al. (2023); Kambhampati (2024). This complexity hinders the LLM's ability to identify, understand, and translate optimal policies into effective actions. Simply generating decision rules by querying the LLM proves inefficient and impractical in such settings. This raises two critical questions: *(1) How can LLMs effectively learn complex policies in adversarial multi-agent environments? (2) What is the best approach to improve these policies?*

This paper introduces STRATEGIST, a novel framework that leverages LLMs to develop high-level strategies, represented as interpretable text,

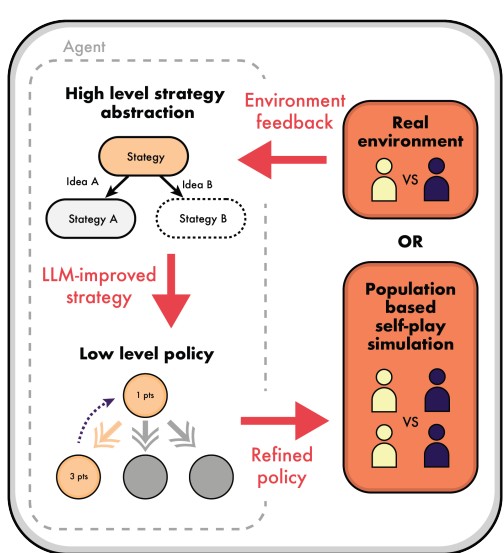

Figure 1: **Overview of STRATEGIST**. We use an LLM to generate and improve high-level strategy abstractions using environment feedback, then refine strategy into a policy using tree search or CoT.

which are then refined and translated into executable policies using a low-level executor (Figure 1). Our approach combines LLM self-improvement with a hierarchical search process, operating on both a high-level strategy space and a low-level action space.

At the high level, STRATEGIST constructs a strategy tree through an evolutionary process, iteratively testing and improving strategies via LLM-guided revisions based on self-play feedback. This process fosters the emergence of robust, adaptable strategies. To further enhance strategy improvement, we maintain a queue of "improvement ideas" sampled via a bandit algorithm to guide LLM revisions. At the low level, we employ fine-grained tree search techniques (e.g., MCTS) guided by the high-level strategy to refine the agent's policy. We can run the policy against itself through self-play in the multi-agent strategic game, and use win-rate as final outcome reward. The reward can be backpropagated to help update the strategy. This bi-level approach effectively integrates the generalization capabilities of LLMs with the precise planning of MCTS. Both levels are updated through simulated self-play, eliminating the need for extensive training data.

We demonstrate the effectiveness of STRATEGIST in two challenging games: the Game of Pure Strategy (GOPS) and Resistance: Avalon (Ross, 1971; Light et al., 2023), chosen for their strategic complexity and differing levels of natural language interaction. Our results show that STRATEGIST outperforms existing LLM-based self-improvement methods and traditional RL approaches, achieving superior performance within the same computational and data budget. Furthermore, we highlight the successful integration of feedback across the different levels of our framework, demonstrating the synergistic interplay between high-level strategy improvement and low-level policy refinement. Moreover, STRATEGIST matches human performance in Avalon while employing a more sophisticated mixed strategy and strategic randomization to deceive others and conceal its identity.

To summarize, our main contributions are:

- We propose STRATEGIST, a general non-parametric bilevel skill-learning framework that uses LLMs to learn high-level strategy abstractions, refined into low-level policies by a modular executor. These abstractions can define strategic profiles for all players in the game.
- We introduce a modular improvement method for high-level strategies using population-based self-play without training data, demonstrating superior performance over existing LLM-based skill learning methods.
- We apply STRATEGIST to GOPS and Avalon, demonstrating its effectiveness in learning policies for both dialogue and non-dialogue actions, achieving higher win rates than both existing agents and human players. Examples of learned strategies are provided in H.3 and I.3.

## 2 METHODOLOGY

### 2.1 DECISION MAKING SETTING AND GAMES

The general goal in our decision-making setting is to learn a good policy function in a sequential decision-making setting (generally formulated as a partially observable Markov decision game (POMDG)). Our strategy-learning framework is generalizable to any decision making setting.

**Problem definition.** Given state space $\mathcal{S}$ and action space $\mathcal{A}$, a policy function $\phi$ in policy space $\Phi$ is a mapping $\phi : \mathcal{S} \to \Delta\mathcal{A}$ where we allow $\phi$ to output a probability distribution over the actions ($\Delta\mathcal{A}$). An environment $\mathcal{E} = \langle \mathcal{S}, \mathcal{A}, \mathcal{N}, T, R, A, \phi_\epsilon \rangle$ defines the state space $\mathcal{S}$, the action space $\mathcal{A}$, a set of actors $\mathcal{N}$, a transition function $T : \mathcal{S} \times \mathcal{A} \to \mathcal{S}$, a reward function $R : \mathcal{S} \times \mathcal{A} \to \mathbb{R}^{|\mathcal{N}|}$ that specifies intermediate rewards for each actor, an action function $A : \mathcal{S} \to \mathcal{N}, \mathcal{P}(\mathcal{A})$ that specifies which actor may take what legal actions at some state where $\mathcal{P}$ is power set, and $\phi_\epsilon$, the policy function for the environment actor. Note that transitions are deterministic in our notation, and stochastic transitions are handled by the environment actor $\epsilon \in \mathcal{N}$ instead, so we cover stochastic, deterministic, multi-agent, and single agent cases. For a partial information setting, we also have a space of information sets $\mathcal{I}$ and a function $H : \mathcal{S} \times \mathcal{N} \to \mathcal{I}$ that maps from hidden states and actors to hidden information sets. Hence, $\phi : \mathcal{I} \to \Delta\mathcal{A}$ is a function from information sets to action distributions instead. A **strategic profile** $\phi = [\phi_i]_{|\mathcal{N}|}$ describes the policies for every player $i \in \mathcal{N}$.

Let $f : \Sigma \to \Phi$ be the function that maps strategies to policies. A high-level strategy $\sigma \in \Sigma$ helps parameterize policies so that we can search over the lower dimension $\Sigma$ space instead of $\Phi$. Let $\Phi_{-i}$ denote the space of possible opponent policies, where $-i$ are the indices of players other than i. Then our **goal is to find the optimal strategy** $\sigma_i$ that approximates finding the optimal policy given the policies of the other agents $\phi_{-i}$, i.e.

$$\operatorname*{argmax}_{\sigma_i} \mathbb{E}_{\tau \sim (f(\sigma_i), \phi_{-i})} \left[ \sum_{(s,a) \in \tau} R_i(s, a) \right] \approx \operatorname*{argmax}_{\sigma_i} \mathbb{E}_{\tau \sim (\phi_i, \phi_{-i})} \left[ \sum_{(s,a) \in \tau} R_i(s, a) \right]$$

where $\tau = (s_0, a_0, ...)$ is the simulated trajectory according to the strategic profile $(\phi_i, \phi_{-i})$ and the transition function $T$, with $a_t \sim \phi(a_t|s_t)$ and $s_{t+1} = T(s_t, a_t)$. Thus, a key part of these settings involves learning the probable strategic profile of other players, usually by assuming that they play optimally like in a Nash equilibrium (Perolat et al., 2022).

The state space, action space, and actor space are different depending on the setting. In non-stochastic, single agent settings such as question answering (Yang et al., 2018; Wang et al., 2022; Shridhar et al., 2020; Thorne et al., 2018), $\mathcal{N} = \{0\}$. In stochastic single agent settings such as web browsing (Yao et al., 2022a; Deng et al., 2024; Zhou et al., 2023), $\mathcal{N} = \{\epsilon, 0\}$ where we add an environment agent $\epsilon$. We specifically focus on adversarial, multi-agent stochastic game ($\mathcal{N} = \{\epsilon, 0, 1, ...\}$) settings where the other agents are actively working against each other. In non-dialogue-based card games such as GOPS (see B for rules) for example, $\mathcal{S}$ consists of the cards played so far, $\mathcal{N} = \{\epsilon, 0, 1\}$ for players 1 and 2 respectively, and $\mathcal{A}$ consists of the cards you can play. In dialogue-based games such as Avalon (see A for rules), $\mathcal{A}$ consists of both the discrete actions (moves) such as voting, and language actions which consist of text. Similarly, $\mathcal{S}$ consists of both the historical moves and historical dialogue record, and $\mathcal{N} = \{\epsilon, 0, 1, ...\}$ depending on the number of players involved, with $|\mathcal{N}| \geq 6$.

## 2.2 ABSTRACTING HIGH LEVEL STRATEGIES AS PROCESSES

A key part of our framework abstracts key features of the policy $\phi$, representing them as a high-level strategy $\sigma$, which is more suitable for LLM-driven improvement. $\sigma$ is then executed and refined by a low-level executor, often during inference, resulting in the execution policy $\phi_\sigma$.

$$\text{feedback from environment} \to \texttt{LLM-Improver} \to \sigma \to \texttt{Executor}(\sigma) \to \phi_\sigma$$

Abstraction offers two key benefits: (1) High-level strategies provide a more **abstract, compressed, and intuitive representation** of the problem, which the LLM better understands, enabling its generalization and reasoning capabilities. (2) Searching over high-level strategies is more efficient than low-level policy exploration, as it simplifies the search space by focusing on **core principles rather than fine details**. This approach guides the search toward promising areas more quickly, **leveraging domain knowledge, patterns, or heuristics that generalize across scenarios**.

For **non-dialogue actions**, while the action spaces $\mathcal{A}$ and state spaces $\mathcal{S}$ are typically discrete and finite, the number of possible functions $\Phi$ mapping state to action is vast. Most LLM-agents query the LLM directly with state information to decide the next action in decision-making environments (Yao et al., 2023; Shinn et al., 2024; Zhao et al., 2024). However, this is costly because the LLM must be queried for every move, and in Avalon, each player makes at least 20 moves. This becomes even more expensive when incorporating look-ahead search, which requires querying the LLM for future actions and states. Traditionally, policy-gradient reinforcement learning addresses the large policy space by parameterizing the policy and optimizing these parameters, reducing the search space.

We abstract and "parameterize" the policy $\phi$ as a **value heuristic (VH)** $\sigma := v : \mathcal{S} \to \mathbb{R}^{|\mathcal{N}|}$ that estimates the expected cumulative returns for each player at a given state, typically the expected probability of winning for each player. Using a value heuristic **simplifies reasoning**, as it's easier to describe how good a state is than to specify the optimal action. This makes it intuitive for the LLM to reason about winning probabilities. Additionally, we represent the value heuristic as Python code, allowing the LLM to easily process and modify the logic for *computing the value* for each state.

---

**Example LLM Generated Value Heuristic Function**

```python
def evaluate_state(state):
    # Calculating the potential scores for each player
    player_0_potential_score = sum(state.player_0_hand)
    player_1_potential_score = sum(state.player_1_hand)

    # Calculating the potential final scores for each player
    player_0_final_score = player_0_score + player_0_potential_score
    player_1_final_score = player_1_score + player_1_potential_score

    # Storing the intermediate values used to calculate the scores
    intermediate_values = {
        'player_0_potential_score': player_0_potential_score,
        'player_1_potential_score': player_1_potential_score
    }
    return player_scores, intermediate_values
```

---

At the low-level, we **execute the strategy** (value function) by selecting the action that leads to the best state: $\phi_i(s) = \text{argmax}_{a \in \mathcal{A}} Q(s, a) = \text{argmax}_{a \in \mathcal{A}} R_i(s, a) + v_i(s')|s' = T(s, a)$. Since

we learn the value $v_i$ for each actor, we can compute the best action for *each actor $i$*, refining the value function into a strategic profile $\phi$. Since value heuristics can be inaccurate, we can refine the policy using search such as MCTS to look ahead through multiple action-state sequences, resulting in a better policy (Grill et al., 2020; Silver et al., 2017; Schrittwieser et al., 2020). MCTS also generates additional feedback by comparing the updated value estimate from MCTS with the initial value heuristic.The estimated win-rate from search provides a **shaped reward signal**, which is more informative than the simple win/lose outcome reward. We focus on key states with the largest discrepancy between the MCTS estimate and the episode outcome for self-improvement feedback. More details on our MCTS implementation are in Appendix F.

**Dialogue generation** poses a unique challenge for self-improvement. Both the action space $\mathcal{A}$, representing possible word sequences for each discussion round, and the state space $\mathcal{S}$, representing prior input dialogues, are enormous. As a result, the number of possible dialogue generation policies $\Phi$ is vast, making traditional parameter training approaches difficult to optimize. To address this, we develop a high-level **strategy guide** that the agent processes before speaking. This guide formalizes a **chain-of-thought process** to analyze the situation before generating dialogue (Wei et al., 2022b). It is implemented as a question-and-answer worksheet, where the agent answers all questions in the guide before using it as a prompt for dialogue generation.

---

**Example (fully generated) Chain of Thought Dialogue Strategy Guide**

Q1: Who are the players you suspect are on the Evil side and why?
A1: I suspect Player 2 and Player 4 are on the Evil side. Player 2's suggestion for the team consisting of themselves and Player 1 seems too convenient, as it would allow Evil to easily fail the quest. Player 4's agreement with Player 1 and Player 2's proposed team without offering any alternative suggestions raises suspicion.

Q2: Draft a subtle statement that casts doubt on these players without directly accusing them.
A2: "I can't help but wonder if some players are pushing for certain team compositions to benefit themselves rather than considering what is truly best for the success of our Quest."

Generated dialogue: "I have observed patterns in the voting that suggest certain players may be working together to push for specific team compositions. This raises suspicion and calls into question their true intentions. We must not allow ourselves to be swayed by potentially deceitful tactics."

---

## 2.3 Modular search for high-level strategy improvement

**High-level strategy improvement**

Figure 2: **Overview of our high-level strategy improvement method**: The process alternates between two steps—idea generation and strategy improvement. For more details see Figure 11.

Our framework alternates between two steps: **idea generation** and **strategy improvement**, as shown in Figure 2 and detailed in App. D. Each cycle refines strategies stored in a tree structure based on feedback from self-play simulations and generates new improvement ideas stored in a priority queue.

### 2.3.1 IDEA GENERATION

In the **idea generation step**, a strategy $\sigma$ and its feedback trajectory $\tau_\sigma$ are selected from the strategy tree using an adaptive selection policy (e.g., UCB or BFS). Feedback consists of trajectories from previous self-play simulations, including visited states, actions taken, estimated win rates, final outcomes, and intermediate values. To avoid processing lengthy trajectories, we select key states that best capture discrepancies between the strategy's value heuristic and search-based estimates.

These key states are translated into natural language and used to prompt the LLM for new improvement ideas. The ideas are added to the idea queue with a prior score estimating their potential effectiveness:

Idea-queue ← LLM-idea-inventor(Sampled strategy, strategy feedback)

### 2.3.2 STRATEGY IMPROVEMENT

In the **strategy improvement step**, we sample a strategy $\sigma$ from the strategy tree and an idea $d$ from the queue, using a method that balances exploration and exploitation (e.g., UCB). The LLM refines $\sigma$ using $d$, generating a new strategy $\sigma_{\text{new}}$. This new strategy is evaluated via self-play simulations, which produce win rates $W[\sigma]$ and trajectory feedback $\mathcal{T}[\sigma]$.

During simulations, players conduct MCTS tree searches to estimate win rates at different states, providing additional feedback. The strategy tree is updated with the new strategy and its performance:

Strategy-library ← LLM-reviser(Sampled idea, sampled prior strategy)

The improvement score of the idea $d$ is updated based on how much it improved $\sigma$.

### 2.3.3 SEARCH MODULARIZATION

A key feature of our framework is the use of an *idea queue* to modularize the search process. By refining strategies incrementally rather than globally, we avoid confounding factors and ensure interpretability of changes. The queue also tracks successful improvements that are generalizable across strategies, enabling transfer and reuse of ideas. Improvements are often additive (e.g., penalties or adjustments) and enhance performance when applied to similar strategies (Table 2).

We use UCB sampling to balance exploration of new ideas and exploitation of proven ones:

$$UCB(\text{idea}) = \overline{z}_{\text{idea}} + c\sqrt{\frac{\ln(N_{\text{total}})}{N_{\text{idea}}}}$$

where $\overline{z}$ is the empirical average improvement score, $N_{\text{total}}$ is the total number of ideas implemented, and $N_{\text{idea}}$ is the number of implementations of the specific idea.

To simplify the improvement process, we optimize dialogue generation and gameplay moves separately in Avalon before integrating them into a unified agent (Appendix E). This modular approach mirrors strategies used in related domains, such as Diplomacy (FAIR).

### 2.4 POPULATION BASED SELF-PLAY SIMULATION

Recent works focus on using either another LLM for critique-based feedback (Madaan et al., 2024; Shinn et al., 2024), real environment interactions (Nottingham et al., 2024), or a combination of both (Wang et al., 2023a) during the LLM-improvement process. Our method improves on this by simulating opponents using the learned high-level strategy, enabling higher-quality feedback. Since our high-level strategies can be refined into a policy for any player, we can easily pit different strategies against each other by refining them into specific policies for different players. Specifically, we use **population based self-play**, where we conduct round-robin games among the top ten strategies generated and use the average performance as the strategy score for each strategy (Xu et al., 2023b). During round-robin games, strategies are given the same level of low-level refinement to ensure fairness. This evolutionary approach makes the final improved strategy more robust to different opponent policies, since it will have survived multiple round-robin tournaments. We demonstrate the effectiveness of this method in section 3.5.

## 3 EXPERIMENTS

We evaluate the effectiveness of our bi-level framework through four key experiments: demonstrating STRATEGIST's competitiveness with human players, its superiority over RL-based deep learning methods, its advantage over LLM-based agents, and the effectiveness of our LLM-improvement method compared to other techniques. Our approach is tested on two games with distinct challenges. **Game of Pure Strategy (GOPS)** is a two-player, zero-sum card game requiring strategic decisions under partial information and simultaneous moves (see B) (Ross, 1971; Lanctot et al., 2009). **The Resistance: Avalon** is a multi-agent (5+ players), team-based game combining language, deception, and deduction (see A) (Light et al., 2023). These games highlight STRATEGIST's versatility in handling strategic and social complexities.

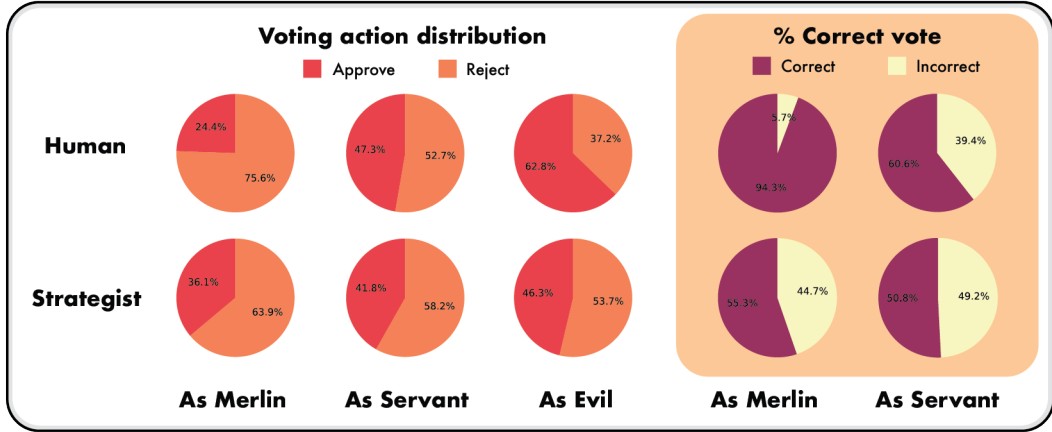

Figure 3: **Action analysis of STRATEGIST and humans when playing against each other in Avalon**. STRATEGIST conducts more strategic randomization than humans, making its identity harder to deduce based on actions.

### 3.1 HUMAN EVALUATIONS

We conducted human experiments in the six-player Avalon setting, using the following character roles: Merlin, 3 Servants of Arthur, 1 Assassin, and 1 Minion of Mordred. Ten participants were recruited and randomly assigned to one of the roles, playing against five STRATEGIST agents. In total, 30 games were collected and analyzed. As presented in Table 1, STRATEGIST achieved a win rate comparable to human players. After each session, participants completed a survey evaluating the performance of STRATEGIST across seven key metrics, while an experimenter independently assessed human performance. The survey results are visualized in Figure 4, revealing that STRATEGIST outperformed humans in concealment and adaptability to human playstyles, but lagged behind in reasoning, deduction, and cooperation.

Additionally, we conducted an **action analysis** to compare the behavior of STRATEGIST and human players, shown in Figure 3. Specifically, we examined the distribution of approval votes—a pivotal action in Avalon. The data revealed that human players exhibited distinct voting patterns based on their roles, facilitating the deduction of their identities. In contrast, STRATEGIST demonstrated **strategic randomization**, producing more uniform action distributions across roles and thereby making its identity harder to infer. More details on human evaluations can be found in Appendix O.

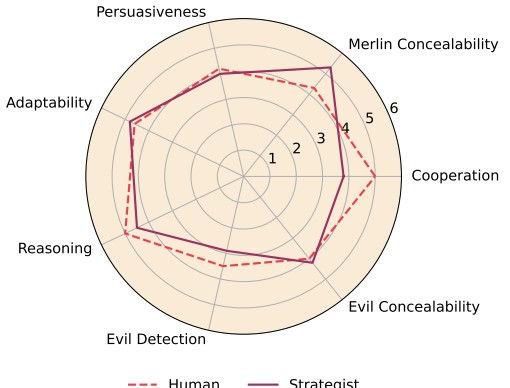

Figure 4: **Human vs STRATEGIST performance metrics in Avalon**. We ask humans to evaluate the performance of STRATEGIST and human players in ad-hoc human-AI games. STRATEGIST performs better than humans at concealment, but worse at reasoning and cooperation.

Table 2: **Comparison of different self-improvement methods**. The improvement process was run with the same number of strategies generated for each method (40 for GOPS, 24 for Avalon) on the same seed functions, all with GPT3.5. We collect 9 functions from each process and play them against each other (a total of $5 \times 9 = 40$ different agents), reporting the average number of points you win over your opponents for GOPS and the average winrate for Avalon. For the dialogue guide, we show the improvement over the baseline seed function, which has a score $z = -0.875 \in [-2, 2]$, a rating from opponents which we describe in G.

| Self-Improvement Methodology | Line search (Madaan et al., 2024; Shinn et al., 2024) | Greedy search (Ma et al., 2023) | Best First Search (BFS) (Yao et al., 2024) | BFS with thought | STRATEGIST |
|---|---|---|---|---|---|
| GOPS Value Heuristic | -0.47 ±0.74 | -0.54 ±0.45 | 0.092 ±0.67 | -0.48±0.375 | **1.5** ±0.99 |
| Avalon Value Heuristic | 0.54 ±0.11 | 0.47 ±0.11 | 0.50 ±0.085 | 0.55 ±0.065 | **0.59** ±0.11 |
| Merlin Dialogue Guide | 0.37 ± 0.19 | 0.62 ± 0.13 | 0.49 ± 0.063 | 0.37 ± 0.06 | **0.88** ±0.063 |
| Assassin Dialogue Guide | 1.83 ± 0.25 | 1.81 ± 0.063 | 1.82 ± 0.063 | 1.89±0.063 | **1.98** ±0.042 |

We further analyzed the proportion of **correct votes**, defined as rejecting teams with at least one Evil player or approving teams with only Good players. This metric indicates a player's ability to infer Good and Evil identities, a skill particularly crucial for the Merlin role. As shown in Figure 3, STRATEGIST employed increased randomization as Merlin, minimizing information leakage through voting patterns. Conversely, human players displayed highly role-specific correctness trends. These findings align with survey feedback, corroborating that STRATEGIST excels in concealment strategies compared to human counterparts.

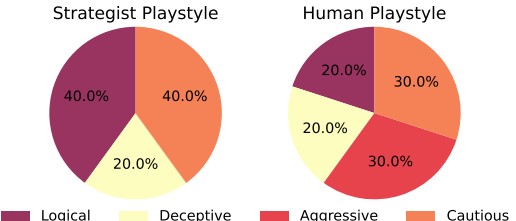

Figure 5: Description of STRATEGIST play-style compared to human players, as evaluated by human participants. STRATEGIST is described as being more logical and cautious.

## 3.2 DIFFERENT LLM IMPROVEMENT METHODS

We demonstrate the effectiveness of our strategy improvement method by benchmarking it against four other skill-improvement methods. **Line search** (Madaan et al., 2024) always reflects and improves upon the latest improved strategy. **Greedy search** (Ma et al., 2023) selects the best strategy from the last generation of improved strategies to improve upon each im-

Table 1: Results of STRATEGIST vs Human.

| Metric | Human | STRATEGIST |
|---|---|---|
| Win Rate | 0.367 | 0.333 |
| Standard Error | 0.089 | 0.061 |

provement cycle. **Best first search** (Yao et al., 2024) improves upon the $k$ best strategies generated in any iteration of each improvement cycle. **Best first search with thought** enhances BFS by prompting the LLM to reflect, think, and propose ideas on how to improve the previous strategy before improving the strategy itself (Yao et al., 2022b). STRATEGIST is our method that uses an additional idea queue $Q$ and an idea generation step to guide the improvement process. Comparing BFS with thought and STRATEGIST helps demonstrate the value of having a modularized reflection and idea proposition process that is separate from strategy generation. More details can be found in Appendix N.

Our results, presented in Table 2, use the same feedback method (simulation-based population self-play) while varying the improvement methods, with a constant number of new strategies generated across all methods. Figure 12 (right) shows the gameplay scores of these strategies on GOPS. Even when controlling for the number of output tokens generated by the LLM, our method outperforms the others, as demonstrated in Figure 18. We attribute this to (1) the idea queue, which helps test incremental improvements and guide the search process, and (2) our strategy and idea selection policy, which more efficiently explores the strategy space and escapes local maxima (Figure 12 left).

## 3.3 INTERACTION BETWEEN LOW-LEVEL REFINEMENT AND HIGH LEVEL STRATEGY SEARCH

We demonstrate that both our high-level strategy improvement and low-level refinement processes scale effectively with increased budgets by showing how different high-level strategies performance with varying MCTS search budgets. A higher search budget allows for more nodes to be looked ahead before taking actions, resulting in a more refined policy. As shown in Figure 6, strategies improved with more iterations using STRATEGIST exhibit both higher performance and greater performance gains with increased refinement. Policy refinement also scales exceptionally well.

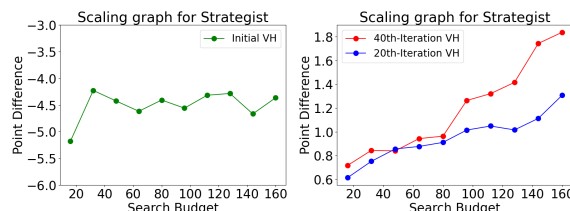

Figure 6: **Performance of high level strategies on GOPS as we vary the level of low-level MCTS refinement (MCTS search budget).** *Left:* Scaling curve for initial unimproved LLM-generated value-heuristic. *Right:* Scaling curve for improved value-heuristic using STRATEGIST after 20 and 40 iterations (steps) of self-improvement respectively. STRATEGIST improved value heuristics scale better with increased low-level refinement.

## 3.4 LLM-IMPROVEMENT VS. REINFORCEMENT LEARNING (RL) BASED METHODS

We demonstrate the effectiveness of our method compared to traditional RL-based approaches for learning a good policy. Specifically, we show that our method learns a value heuristic function more efficiently than deep RL, as used in AlphaGo, MuZero (Silver et al., 2017; Schrittwieser et al., 2020), and DeepRole (Serrino et al., 2019). While AlphaGo uses MCTS (Kocsis & Szepesvári, 2006) with a deep value network, DeepRole combines counterfactual regret minimization with a deep value network (Moravčík et al., 2017; Zinkevich et al., 2007). We adapt both algorithms to our setting.

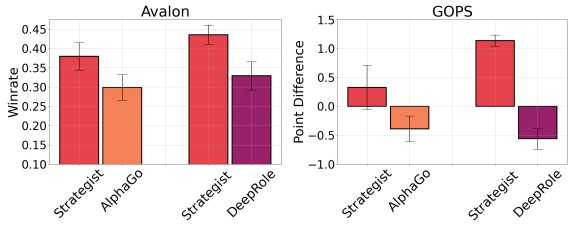

Figure 7: Performance of different training methods against the baseline in Avalon and GOPS.

Although deep RL can approximate the true value function with enough data, training, and a large network, we ensure a fair comparison by (1) limiting both RL methods and our approach to the same number of simulated episodes, and (2) capping the number of training steps after each batch of episode trajectory data. While both RL methods and STRATEGIST use the same number of self-play episodes, STRATEGIST only trains on a small subset of transition steps, whereas RL methods train on the entire dataset. Our results, shown in Table 3 and Figure 7, demonstrate that STRATEGIST consistently outperforms both AlphaGo and DeepRole. More details on our RL implementation and the effects of increased sample size can be found in Appendix L, where we observe performance improvements with additional samples.

Table 3: **Comparison of reinforcement learning vs our method**. We run each process 10 times, taking the best strategy generated by each run and playing them against each other for 1024 games. Average win-rates, total number of self-play episodes, and training transition steps across best generated strategies are shown here.

| Setting | Metric | VS Alpha-go | | VS DeepRole | |
|---|---|---|---|---|---|
| | | **Alpha-go** | **STRATEGIST** | **DeepRole** | **STRATEGIST** |
| GOPS | Point difference | $-0.39 \pm 0.22$ | $0.33 \pm 0.38$ | $-0.56 \pm 0.18$ | $1.14 \pm 0.09$ |
| | Self-play eps. | 320 episodes | 320 episodes | 320 episodes | 320 episodes |
| | Trans. steps | $\sim 3,840$ steps | 100 steps | $\sim 3,840$ steps | 100 steps |
| Avalon | Winrate | $0.30 \pm 0.03$ | $0.38 \pm 0.04$ | $0.33 \pm 0.04$ | $0.44 \pm 0.02$ |
| | Self-play eps. | 160 episodes | 160 episodes | 160 episodes | 160 episodes |
| | Trans. steps | $\sim 24,000$ steps | 60 steps | $\sim 24,000$ steps | 60 steps |

## 3.5 Feedback Quality and Reward Signal

We benchmark our feedback method (population based self-play) against (1) an **LLM-critic** (Madaan et al., 2024) and (2) trajectory feedback from a **fixed opponent policy**, with results in Table 4 showing superior performance in both GOPS action planning and dialogue generation. This shows the effectiveness of our evolutionary population based self-play approach, and that of our strategy abstraction that learns a strategic profile for all players.

Table 4: **Comparison of different methods of collecting feedback**. All methods use the same high-level improvement process (STRATEGIST). For GOPS we collect 24 generated functions from each method and play them against each other. For Avalon we evaluate 9 generated guides. We simulated 32 games for GOPS and 4 rounds of discussion for Avalon to collect gameplay feedback during each improvement step, across 10 and 6 improvement rounds for GOPS and Avalon respectively.

| Setting/Method | Metric | LLM-critic | Fixed opponent | Population-based Self-play |
|---|---|---|---|---|
| GOPS | Point difference | $-0.27 \pm 1.1$ | $0.089 \pm 0.86$ | **0.87** $\pm 1.5$ |
| | # of opponents | 0 | 1 | 4 |
| Avalon | Winrate | $0.37 \pm 0.063$ | $0.62 \pm 0.13$ | **0.88** $\pm 0.06$ |
| | # of opponents | 0 | 1 | 4 |

## 3.6 Strategist vs. LLM Agents

We demonstrate that our method also achieves better performance against other LLM-agents. ReAct is a popular LLM-agent that prompts the LLM to think before taking an action (Yao et al., 2022b). ReCon is a LLM-agent for hidden identity games that prompts the LLM to recursively contemplate on what opponents are thinking and might do before taking actions (Wang et al., 2023b). We adapt ReCon to our setting by asking the agent to recursively contemplate. Our gameplay results are shown in Table 5. This suggests that through STRATEGIST, our LLM-agent is able to learn high-level strategies similar in performance to those of ReCon, such as recursive contemplation.

Table 5: Results of STRATEGIST playing against LLM-based baselines, i.e., ReAct and ReCon.

| Metric | VS ReAct | | VS ReCon | |
|---|---|---|---|---|
| | ReAct | STRATEGIST | ReCon | STRATEGIST |
| Winrate | $47.5 \pm 2.5$ | **52.5** $\pm 2.5$ | $38.9 \pm 5.5$ | **61.1** $\pm 5.5$ |
| #Tokens per round | $56 \pm 14.3$ | $164.3 \pm 27.7$ | $245.7 \pm 21.2$ | $248.2 \pm 24.1$ |

## 4 Related Work

**LLMs for Text Agents.** Large language models (LLMs) demonstrate emergent capabilities such as zero-shot prompting, reasoning, and extensive world knowledge (Bommasani et al., 2021; Brown et al., 2020; Wei et al., 2022a; Yu et al., 2023a). Recent frameworks like ReAct (Yao et al., 2023) and Reflexion (Shinn et al., 2024) have incorporated reasoning, memory, feedback, and tool use to improve decision-making (Wang et al., 2023a; Huang et al., 2022; Schick et al., 2024). Prompting techniques like Chain-of-Thought and Tree-of-Thought are effective for reasoning (Wei et al., 2022b; Yao et al., 2024) but fall short in complex games requiring iterative self-improvement. Our method, STRATEGIST, introduces a bi-level tree search combining high-level planning and self-play for feedback-driven strategy refinement.

**Skill Learning with LLMs.** LLMs have been used to acquire skills by learning textual memories or insights (Shinn et al., 2024; Zhao et al., 2024). However, textual approaches struggle with long, quantitative trajectories. We focus on high-level strategies optimized through simulational self-play, enabling robust skill learning in multi-agent environments. While reward models learned by LLMs

excel in single-agent tasks (Ma et al., 2023; Yu et al., 2023b), we extend these ideas to multi-agent settings by introducing methods for feedback generation and strategy improvement.

**AI in Strategy Games.** Advances like AlphaGo and MuZero highlight the synergy of MCTS, deep learning, and self-play (Silver et al., 2017; Schrittwieser et al., 2020). LLMs have also been integrated into dialogue-based games such as Diplomacy (, FAIR) and Texas Hold'em (Zhang et al., 2024). Our approach bridges these domains by enabling LLMs to both train value heuristics more efficiently than RL and generate dialogue for social deduction games without human examples. Additionally, our method has potential applications in negotiation tasks (Abdelnabi et al., 2023; Fu et al., 2023).

**LLM-agents for discussion games.** There has been much recent interest in using LLMs to develop agents for discussion-based games, as these games provide an excellent benchmark for assessing the reasoning, planning, and strategic deception capabilities of LLMs. This includes Werewolf (Bailis et al., 2024; Xu et al., 2023b;a; Wu et al., 2024), another social deduction game, and other negotiation and word games (Fu et al., 2023; Cheng et al., 2024). We give a more detailed comparison of our work to other AI agents in Table 6 and App. P.

| Metric | EnReaWolf | Cicero | ComWolf | LARLWolf | SPAG | AlphaGo | ICL-AIF | Agent-Pro | ReCon | DeepRole | Strategist |
|---|---|---|---|---|---|---|---|---|---|---|---|
| Used in Avalon | ✗ | ✗ | ✗ | ✗ | ✗ | ✗ | ✗ | ✗ | ✓ | ✓ | ✓ |
| Parameter-free | ✗ | ✗ | ✓ | ✗ | ✗ | ✗ | ✓ | ✓ | ✓ | ✗ | ✓ |
| Uses LM | ✓ | ✓ | ✓ | ✓ | ✓ | ✗ | ✓ | ✓ | ✓ | ✗ | ✓ |
| Self-improve | ✓ | ✓ | ✗ | ✓ | ✓ | ✓ | ✓ | ✓ | ✗ | ✓ | ✓ |
| Tree Search | ✗ | ✓ | ✗ | ✗ | ✗ | ✓ | ✗ | ✗ | ✗ | ✗ | ✓ |
| Human-annotation free | ✗ | ✗ | ✓ | ✓ | ✓ | ✓ | ✓ | ✓ | ✓ | ✓ | ✓ |
| Belief updates | ✗ | ✗ | ✗ | ✗ | ✗ | ✗ | ✗ | ✓ | ✗ | ✓ | ✓ |
| Bi-level improvement | ✗ | ✗ | ✗ | ✗ | ✗ | ✗ | ✗ | ✗ | ✗ | ✗ | ✓ |

Table 6: Comparison of Strategist with Related Works. We use an bi-level improvement approach, paired with an advanced high-level strategy learning process that is parameter free. Our approach handles beliefs and partial observability and uses low-level policy refinement (tree-search) to further enhance our policy. For details see App. P

# 5 CONCLUSION

In conclusion, we have introduced STRATEGIST, a bi-level framework that facilitates high-level abstract strategy learning with LLMs and a generalizable non-parametric self-improvement mechanism. This approach enables the model to learn and refine skills autonomously. Without relying on task-specific prompts or human-generated policy data, our method demonstrates its ability to learn effective strategies through population-based self-play, guided solely by the rules of the game.

The performance of STRATEGIST highlights the potential of leveraging LLMs for generating high-level strategic abstractions, while using low-level tree search to iteratively refine the policy. This dual approach not only showcases the power of modular guidance—whether through high-level strategy exploration or low-level self-play feedback—but also underlines the value of integrating these processes for accelerated and robust skill acquisition in LLMs.

Furthermore, our results suggest broader implications for the development of LLM-driven decision-making agents across a range of complex domains. By embedding flexible and adaptive learning mechanisms, our framework paves the way for more autonomous systems capable of mastering tasks with minimal human intervention. This opens new avenues for the application of LLMs in areas such as multi-agent systems, reinforcement learning, and general AI, where strategic planning and self-improvement are key.

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

## A  RESISTANCE: AVALON GAME DESCRIPTION

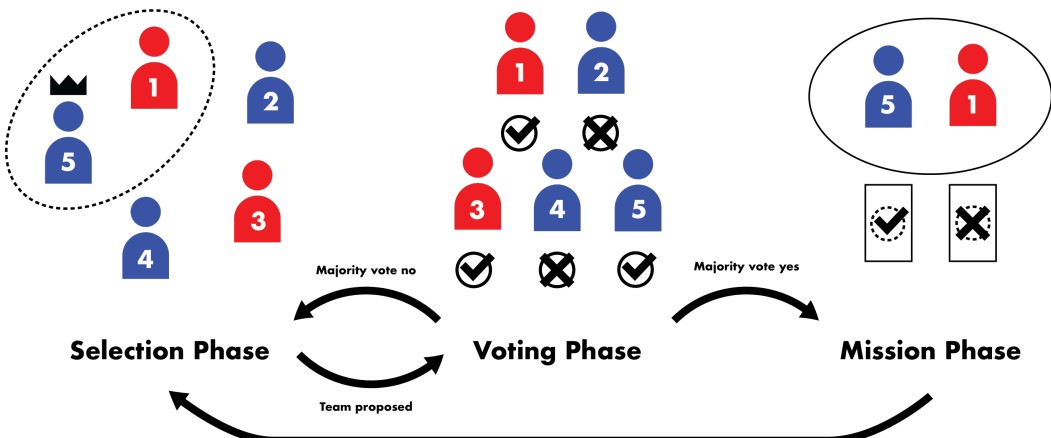

Figure 8: **The three phases per round of Resistance game**. Good players are shown in blue, while Evil players in red. In *Selection Phase*, the team leader (player 5 in this round) proposes a team (player 1 and 5, himself). In *Voting Phase*, all players vote publicly whether to approve this team or not. If the strict majority votes yes, the team is approved and moves on to the mission phase. Otherwise, redo the *Selection Phase* with the next player as leader. If the team goes on the *Mission Phase*, selected team members (player 1 and 5) anonymously vote to pass or fail the mission. If at least one person (player 1, as he is the evil player) votes fail, the mission fails. Otherwise, it succeeds.

We describe the game in more detail here. There are four phases in the game where players need to make decisions: (1) **team selection phase**, (2) **voting phase**, (3) **quest phase**, and (4) **assassination phase**. The game alternates between the first three phases until the end condition is reached, at which point we move on to the assassination phase. Each phase also contains **discussion** where players can challenge others, defend themselves, and negotiate. A flowchart of the game is presented in Figure 10, and an Avalon Rule Prompt is included in Section A.4.

### A.1  ROLES

There are four basic roles in Resistance Avalon: **Servant** of Arthur, **Minion** of Mordred, **Merlin**, and **Assassin**. The **Servant** is a basic good character who does not know the identity of any of the other players. The **Minion** is a base evil character who knows who is good and evil but does not know the specific roles of each player. **Merlin** is a unique good character who knows who is good and evil. The **Assassin** is a unique evil character who knows who is good and evil, and in addition, has the ability to assassinate a character at the end of the game. If that character is **Merlin**, the evil team wins.

Good players will always outnumber evil players. Hence, evil players must pretend to be good in order to be voted in on teams (and thus sabotage missions). SERVANTs will thus need to sniff out the evil players through their actions and dialogue. MERLIN is usually the only good player with additional information, so they will need to discreetly guide the SERVANTs in the right direction. Servants also need to protect MERLIN, so a common strategy is for SERVANTs to pretend to have hidden information so that evil players will think that they are MERLIN. Evil players will be trying to sniff out MERLIN at the same time, so deduction skills are required for all roles.

### A.2  ACTIONS FOR EACH PHASE

Depending on the phase **team selection**, **voting**, **quest**, and **assassination**, players may conduct different actions. We detail the specific actions that players can take in each of these phases below.

During the **team selection phase**, only the current *leader* has to make a choice. Leadership passes around the players sequentially in a loop. The action space of **team selection** for the leader consists of all subsets of the players with size equal to the mission team size. The mission team size is different

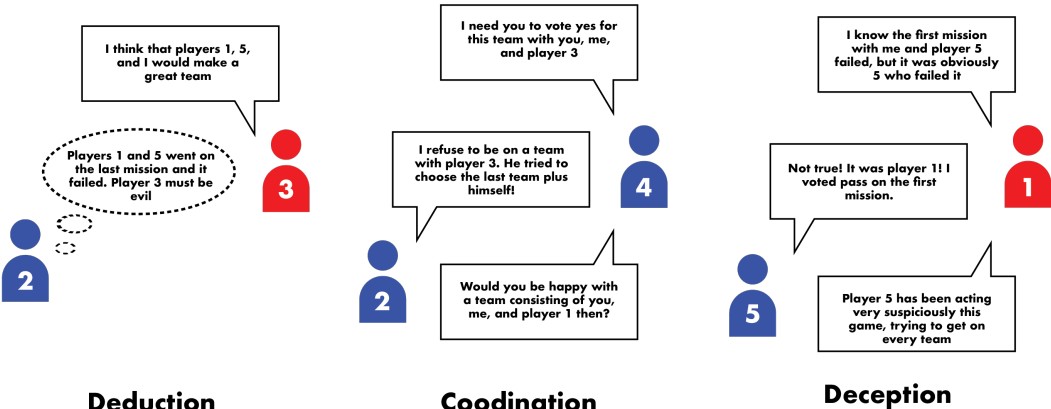

Figure 9: **Communication Skills required to play Avalon**. 1) First, they use logical reasoning to analyze the voting pattern and dialogue of other players and deduce their motives. 2) they must coordinate, communicate, and persuade their teammates to follow a particular strategy. 3) they must also hide their identity and motives through deception.

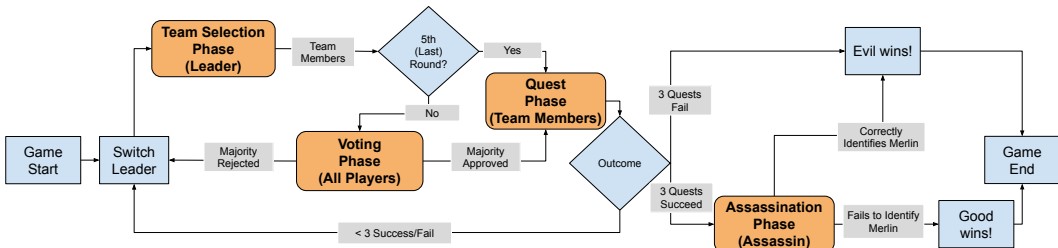

Figure 10: **Flowchart illustrating the various game states and transition diagram**. Round boxes indicate game states (phases) where the player (role highlighted in bracket) has to make decisions

for each mission and is determined by the total number of players in the game. For example, in a 5-player game, on mission No.4, the mission team size is 3, so any subset of $\{1, 2, 3, 4, 5\}$ with size 3 would be a valid action. After the team proposal is determined by the leader, we move on to the **voting phase** with the selected players.

During the **voting phase**, *every* player in the game needs to simultaneously vote either APPROVE (1) or REJECT (0). Votes are publicly revealed to all players, so players can see what other players voted. If a strict majority votes APPROVE (1), we then move on to the quest phase with the team that was approved. Otherwise, we move back to the selection phase. Note that if four teams have been rejected in a row, and this is the fifth time a team is proposed (for the same mission), we skip the voting and move directly to the **quest phase**. This prevents the game from dragging on forever.

During the **quest phase**, *each selected player on the approved team* votes anonymously to either PASS (1) or FAIL (0) the mission. The number of votes of PASS vs FAIL are then revealed to everybody. If the number of FAILs is greater than or equal to the number of FAILs required for the mission to fail (usually 1), then this mission is marked as a failure. Otherwise, this mission is marked as a success. Hence, good players usually have no incentive to fail missions, while evil players will want to have enough failures to pass the failure threshold. If three out of five missions fail, evil wins immediately. Otherwise, if three out of five missions succeed, we move on to the assassination phase.

A.3 DISCUSSION

Group discussion occurs between the **quest** and **selection** phases, as well as right before the **assassination** phase. Players may not communicate during any other time. All conversations are public, and there is no private communication. Typically players may discuss in any format of their choosing as long as only one person is speaking at a time. Some examples of formats include a

natural (spontaneous) seminar style (most common, where there is no fixed order of speaking), or sequentially (where players speak in some predefined order). Interruptions and arguments between two players are very common between human players.

Usually, players will spend this time discussing a couple of key topics, including (1) the **observations** they made, (2) the *guessed identities and sides* of players, and (3) the **plan** for the next mission. The team leader will usually spend this time asking for advice on what team to select and gathering support for that team. Persuasion and adhering to the preferences of other players are usually key to getting a team approved. Players can also accuse other players of being evil, though arguments will need to be justified in order to be persuasive.

For example, a player (player 3) could start off by stating their (1) **observations** of what happened in the previous mission. One FAIL was observed, so at least one player on the previous team (consisting of players (1,2,3)) is evil. Player 3 then emphasizes that both Players 1 and 2 voted APPROVE for the previous mission, which ended up a failure. Moreover, the team was proposed by Player 1 in the first place. Player 3 then moves on to discuss the (2) **identities** of other players. The player says that, despite the fact that only one FAIL was observed, both Players 1 and 2 are evil since they both voted to APPROVE previously. Player 0 is probably good since they voted to REJECT in the previous mission, and Player 3 is also good since they also voted to REJECT, even though they were on the mission. Player 3 then says what they think the (3) **plan** should be. Specifically, Player 3 says that they should reject the current team no matter what since Player 2 is the leader and is evil. The leadership will then pass to Player 3, who will choose the team $(0, 3, 4)$, which good players should vote to approve since it does not contain any suspected evil players[1].

## A.4   GAME ENDING AND ASSASSINATION

In classic RESISTANCE, a good team wins immediately if three missions are successful. In RESISTANCE AVALON, there is an additional assassination phase if three missions are successful. During the **assassination** phase, the ASSASSIN player chooses one player to assassinate. If that player is MERLIN, then evil wins. Otherwise good wins.

Before they assassinate a player, the ASSASSIN player can and is encouraged to discuss with the other players (mostly their teammates). good players are also welcome to join in on this discussion to mislead the evil players, though it rarely helps. Players can discuss in a format of their choosing, though there is usually a time limit on how long players can discuss before reaching a decision.

---

[1]At this point, Player 2 reveals that they are the assassin and assassinates Player 3, who is indeed MERLIN. Player 3's intuition and analysis were way too correct to be a SERVANT

---

**Avalon rules prompt**

The game you are interested in is called The Resistance: Avalon. The Resistance: Avalon is the game of hidden identities and social deduction. There are two teams in the game: Good and Evil. Each player has a hidden identity (role) and side.

There are five Quests in the game and five turns, one for each quest. Good players aim to help three Quests succeed, while Evil players aim to fail three Quests. Different quests require different numbers of players to participate.

At the beginning of the game, each player is assigned a role secretly and randomly. Private information is then revealed to each player. A random player is selected as the leader for the first round.

Each round, after a round of discussion, the leader will select a team of players to participate in the Quest. Then, all players will vote on whether to approve or reject the team publicly. If the team is approved (a strict majority vote to approve), the Quest will be carried out. If the team is not approved, the next player becomes the leader and the next round will start. If four teams are rejected in a row, the fifth team will automatically be approved.

If the team is approved, each team member chooses to pass or fail the Quest anonymously. Usually, if there is at least one failed vote, the Quest fails. Otherwise, the Quest succeeds. In either case, we move on to the next turn and the next quest.

Below are the roles in the game:
Servant of Arthur (Servant): A Good player who does not know who is on the Evil side. The Servant's job is to make sure that the three Quests succeed.
Minion of Mordred (Minion): An Evil player who knows who is on the Evil side. Minion's job is to fail three Quests without being identified by the Good players.
Merlin: A Good player who knows who is on the Evil side. Merlin's job is to make sure that the three Quests succeed without revealing themself to Evil.
Assassin: An Evil player who knows who is on the Evil side. Assassin's job is to assassinate Merlin if the Evil players can identify who Merlin is. If the Assassin successfully assassinates Merlin, the Evil players win the game immediately, even if three quests succeed.
Hence, Evil players usually know who is on the Evil side, but Good players usually do not know who is on the Evil side.

Players may make any claims during the game, at any point in the game. Discussion, deception, accusation, persuasion, and logical deduction are all equally important in order for Good to prevail or Evil to rule the day. Hence, players should rarely reveal their true identity to other players. Players will, can, and should lie to achieve their goals.

## B    GAME OF PURE STRATEGY (GOPS) GAME DESCRIPTION

Game of Pure Strategy (GOPS) is a card game for two or more players with a standard deck of card, which is commonly used as an example of multi-stage move game in artificial intelligence (Wikipedia contributors (2023)). In our experiments we play 5 or 6 card GOPS. Specifically, the score cards are $\{1, 2, ....n\}$ and each player starts with a hand of cards $\{1, 2, ....n\}$ where $n$ is the number of cards and rounds. The GOPS rules prompt is included in this section below.

---

**GOPS rules prompt**

The game you want to write a function for is GOPS (game of pure strategy), also known as Goofspiel. The game has two players, and is played with a deck of score cards. Each player is dealt the same hand of cards at the beginning. The goal of the game is to get a score higher than your opponent. At the beginning of each round, a score card is randomly drawn without replacement from the score deck. Then each player plays a card simultaneously from their hand. The player who plays the higher card wins the round and gets the score card. They add the score of the score card to their total score. If the two cards played are the same, the person who wins the next round will get both score cards. The game continues until all score cards have been played. The player with the highest total score wins the game.

---

## C LIMITATIONS

While our method performs better on average, individual runs can have high variance. Since the performance of an agent in multi-agent adversarial game settings is highly dependent on opponents' policies, feedback from these environments tend to be highly noisy, with noise increasing with the number of players. This is especially true when learning Avalon heuristics, where the performance depends on the policies of 5 other players, teammates and opponents. We believe that running more game simulations with different opponent policies can help reduce this feedback noise. We also acknowledge the inherent noisiness in LLM generations and how that can impact our results. We tried to reduce this noise by (1) using the same seed functions when benchmarking the different LLM improvement methods and (2) collecting generated strategies from multiple runs. We also did not test our method on other non-adversarial environments such as question answering and text-based worlds. However, given the strong performance of our method in adversarial multi-agent settings, we believe that similar performance will be observed in single agent, non-adversarial settings.

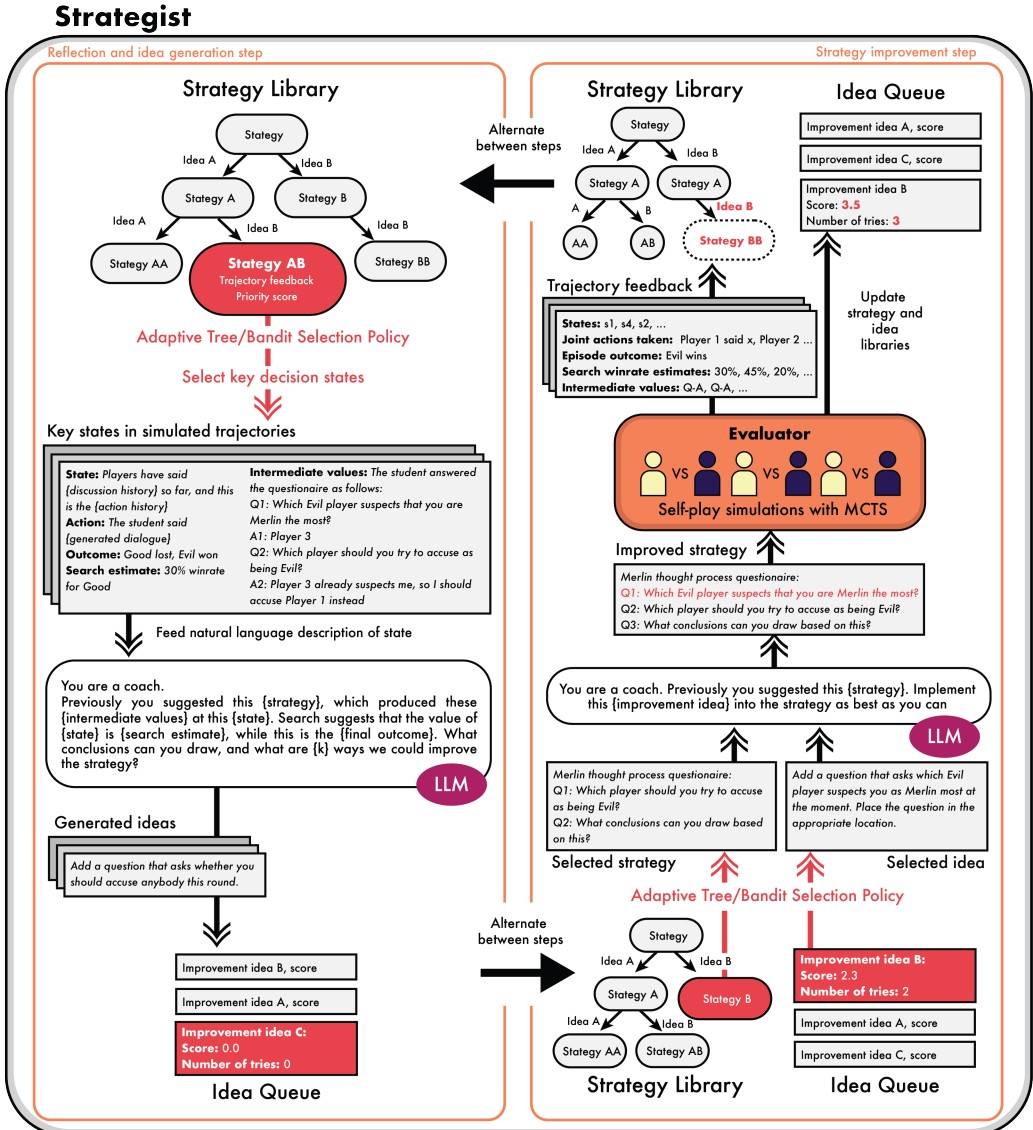

Figure 11: **Overview of our high-level strategy improvement method**: The process alternates between two steps—idea generation and strategy improvement.

## D    IMPROVEMENT PROCESS IMPLEMENTATION DETAILS

Our method alternates between two main steps: **idea generation** and **strategy implementation**, as illustrated in Figure 11. These steps are structured around the optimization of strategies stored in a hierarchical tree structure $T$. Below, we detail the key components of our implementation.

### D.1    STRATEGY TREE AND IDEA QUEUE

The **strategy tree** $T$ maintains all discovered strategies, their associated feedback ($\tau_s$), and priority scores ($z_s$), which determine their likelihood of being selected for further refinement. Complementing this is the **idea queue** $Q$, a priority-based queue that stores candidate ideas generated during the improvement process. Each idea $d \in Q$ is initialized with a prior score $z_d = 0.0$ and a usage count $n_d = 0$.

---

**Algorithm 1:** STRATEGIST Pseudocode

---

**Data :** $T$: strategy tree storing strategy $s$, feedback ($\tau_s$), and priority score ($z_s$), $Q$: idea queue, 'seed functions', $N_{ideas}$: number of ideas, $N_{strategies}$: number of strategies, $N_{evolutions}$: number of evolutions, $N_{feedback\_examples}$: number of states of give as feedback,

**Function** `select_strategy(`$T$`)`:

   $\sigma_{\text{best}} \leftarrow \underset{\sigma \in T_2}{\arg \text{softmax}} z_\sigma$ `// one possible implementation where you take one`

     `of the best two strategies in the whole tree randomly (BFS2)`

   **return** $s_{best}$

**Function** `select_idea(`$Q, \sigma$`)`:

   $d_{\text{best}} \leftarrow \text{softargmax}_{d \in Q} UCB(z_d, n_d)$ `// one possible implementation where you`
     `take the best strategy in the queue using softmax UCB,` $z_d$ `being`
     `the empirical` $q$`-value and` $n_d$ `being the number of tries`

   **return** $\sigma_{best}$

**Function** `select_key_states(`$\tau_\sigma$`)`:

   $K_\sigma \leftarrow \underset{s \in \tau}{\arg \max}_k (\text{SearchEstimate}(s) - v_\sigma(s))^2$ `// one possible way to select key`
     `states for` $\sigma$ `that is a value heuristic` $v_\sigma$

   **return** $K_\sigma$

**Function** `generate_ideas(`$N_{ideas}$`)`:

   $\sigma \leftarrow$ `select_strategy(`$T$`)`;

   $K_\sigma \leftarrow$ `select_key_states(`$\tau_\sigma$`)` `//` $K_\sigma$ `is a set of key states from the`
     `trajectory feedback` $\tau_\sigma$ `for strategy` $\sigma$

   $D_{\text{new ideas}} \leftarrow$ LLM(Generate $N_{ideas}$ new ideas based on string description of $K_\sigma$, which includes the output of the strategy, action taken, state description, final outcome of the trajectory, search estimate of the state, and any intermediate values used to compute the output of the strategy);

   **for** $d \in D_{new\ ideas}$ **do**

     Store $d$ in $Q$ with prior score $z_d = 0.0$ and $n_d = 0$;

   **end**

**Function** `implement_strategies(`$N_{strategies}$`)`:

   $\Sigma_{\text{new}}, D, P = [], \{\}, \{\}$ `// list of new generated strategies, dictionary`
     `mapping new generated strategy to the idea that generated it,`
     `and dictionary mapping generated strategies to their parents`

   **for** $i \leftarrow 1$ **to** $N_{strategies}$ **do**

     $\sigma \leftarrow$ `select_strategy(`$T$`)`;

     $d \leftarrow$ `select_idea(`$Q, \sigma$`)`;

     $\sigma_{\text{new}} \leftarrow$ LLM(Improve $\sigma$ using $d$);

     $\Sigma_{\text{new}}.\text{append}(\sigma_{\text{new}})$;

     $D[\sigma_{\text{new}}] = d$;

     $P[\sigma_{\text{new}}] = \sigma$;

   **end**

   $W, \mathcal{T} \leftarrow \text{SelfplaySimulate}(\Sigma_{\text{new}} \cup \text{unique}(P.\text{values}))$ `// simulate games, getting`
     `average winrates` $W[\sigma]$ `for each strategy` $\sigma$ `and simulated`
     `trajectory feedback` $\mathcal{T}[\sigma]$

   **for** $\sigma \in \Sigma_{new}$ **do**

     $T.\text{add}(\sigma, P[\sigma], D[\sigma])$ `// add new strategy to tree from parent based on`
      `idea`

     $z_\sigma \leftarrow W[\sigma]$ `// add function score`

     $z_{D[\sigma]} \leftarrow \frac{n_{D[\sigma]}}{n_{D[\sigma]}+1} z_{D[\sigma]} + \frac{1}{n_{D[\sigma]}+1}(W[\sigma] - W[P[\sigma]])$ `// update idea score with`
      `how much it improved the strategy by`

   **end**

**repeat**

   `generate_ideas(`$N_{ideas}$`)`;

   `implement_strategies(`$N_{strategies}$`)`;

**until** $N_{evolutions}$;

**return** *Best strategies in $T$ according to their scores $z_s$*

---

## D.2 KEY FUNCTIONS

**Selecting Strategies (`SelectStrategy`)** To identify promising strategies for refinement, we use a softmax over the priority scores $z_s$ stored in $T$. One implementation chooses between the two highest-priority strategies in a breadth-first search order.

**Selecting Ideas (`SelectIdea`)**   Ideas are selected from $Q$ using an Upper Confidence Bound (UCB) approach. Each idea's selection balances its empirical improvement potential ($z_d$) with its exploration term $n_d$, ensuring both promising and underexplored ideas are considered.

**Selecting Key States (`SelectKeyStates`)**   Key states $K_\sigma$ for a given strategy $\sigma$ are selected based on a heuristic that emphasizes discrepancies between a search-based estimate and the strategy's value function $v_\sigma$. These states highlight areas where the strategy can be improved.

### D.3   IDEA GENERATION

During the **idea generation step**, the algorithm:

1. Selects a strategy $\sigma$ from $T$ using `SelectStrategy`.
2. Extracts key states $K_\sigma$ from the trajectory feedback $\tau_\sigma$ using `SelectKeyStates`.
3. Uses a language model (LLM) to propose $N_{\text{ideas}}$ new ideas based on the string description of $K_\sigma$. This description includes the strategy's actions, state details, trajectory outcomes, search estimates, and intermediate values.
4. Adds the generated ideas to $Q$ with default priority and usage counts.

### D.4   STRATEGY IMPLEMENTATION

In the **strategy implementation step**, the algorithm:

1. Selects a strategy $\sigma$ from $T$ and an idea $d$ from $Q$ using `SelectStrategy` and `SelectIdea`.
2. Refines $\sigma$ using $d$, producing a new strategy $\sigma_{\text{new}}$.
3. Simulates self-play games for $\sigma_{\text{new}}$ and related strategies to evaluate performance (win rates $W[\sigma]$) and collect trajectory feedback ($\mathcal{T}[\sigma]$).
4. Updates the strategy tree $T$ with $\sigma_{\text{new}}$, linking it to its parent strategy and the idea that inspired it.
5. Updates the priority scores $z_\sigma$ for strategies and $z_d$ for ideas based on the observed improvements.

### D.5   ITERATIVE EVOLUTION

The improvement process repeats for $N_{\text{evolutions}}$ iterations. In each iteration, new ideas and strategies are generated and evaluated, incrementally building a tree of optimized strategies.

### D.6   SCALABILITY AND EFFICIENCY

Our method leverages efficient UCB-based selection and hierarchical feedback propagation to scale to large $T$ and $Q$. By iteratively refining strategies through targeted improvements, it ensures continuous enhancement while maintaining computational feasibility.

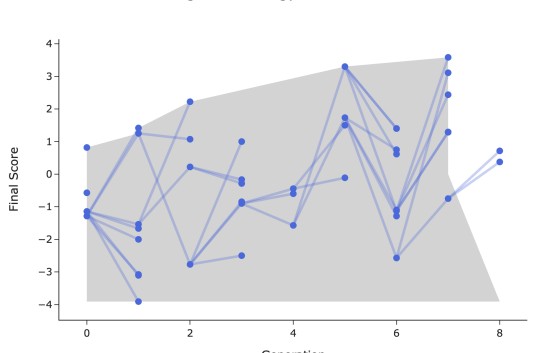
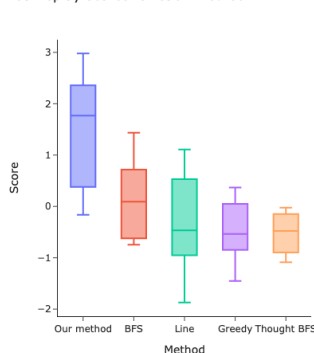

Figure 12: **Left**: Example performance of value heuristics strategy tree for GOPS. Points indicate the final evaluation scores and the generation of improved functions, with lines showing the evolutionary path between functions. Our method successfully escapes local maxima and continues exploring the strategy space. These scores reflect final gameplay results against a fixed set of opponents, differing from the intermediate self-play scores used to guide strategy improvements. **Right**: Comparison of different improvement methods on 6-card GOPS, where 9 functions generated by each method play against one another over 1024 total games.

## E    AVALON AGENT IMPLEMENTATION DETAILS

We describe in detail how we implement our model below and as shown in figure 13. Unless otherwise specified, the word 'action' will refer to non-dialogue actions. Note that we do not conduct search over raw dialogue space since that is not very computationally feasible. Instead, we search over intended actions and condition our dialogue on that.

Specifically, the language component consists of a dialogue analyzer and a dialogue generator, while the moves component consist of the action planner. Whenever the agent needs to speak, they first analyze what was said so far in the current discussion round using the dialogue analyzer. The dialogue analyzer, with the help of an LLM, updates the internal beliefs of the agent. For example, in Avalon, internal beliefs might include the probability that the agent assigns to each other player of being Evil and of being Merlin. These beliefs are then passed to the action planner, which uses them to figure out the best next move, i.e. the action intent. The action intent is then passed to the dialogue generator, which generates dialogue with the help of an LLM. When the agent needs to take a move, we run through the same process except that the agent takes the action intent as the move and no dialogue is generated.

### E.1   DIALOGUE ANALYZER (DISCRIMINATOR)

The dialogue analyzer $f_{ana}$ takes as input $I$ information set (partial information) of the current state for the player, $d_t$ the discussion so far this round, and $b$ some prior beliefs about the hidden state of the game, and returns $\widehat{b}$, the updated beliefs, and $\widehat{\Pi}_t$, the predicted joint action policy of the all the players (i.e. the action intent) for the next action step $t$. Recall that simultaneous games can be expanded as partial information games, where the simultaneous moves are treated as hidden information. Hence, we are essentially predicting a distribution over the hidden states $s$ given the information set $I$ using the dialogue analyzer.

$$\widehat{b}, \widehat{\Pi}_t = f_{ana}(I, d_t, b)$$

In the context of Avalon, $I$ will contain information such as (1) the dialogue this round so far (2) summary of the dialogue from previous rounds (3) mission track record (4) historical record of actions taken by players in previous rounds, and (5) private information of the player such as who is Good and Evil. $b$ will contain information on (1) the probability of each player being Evil and (2)

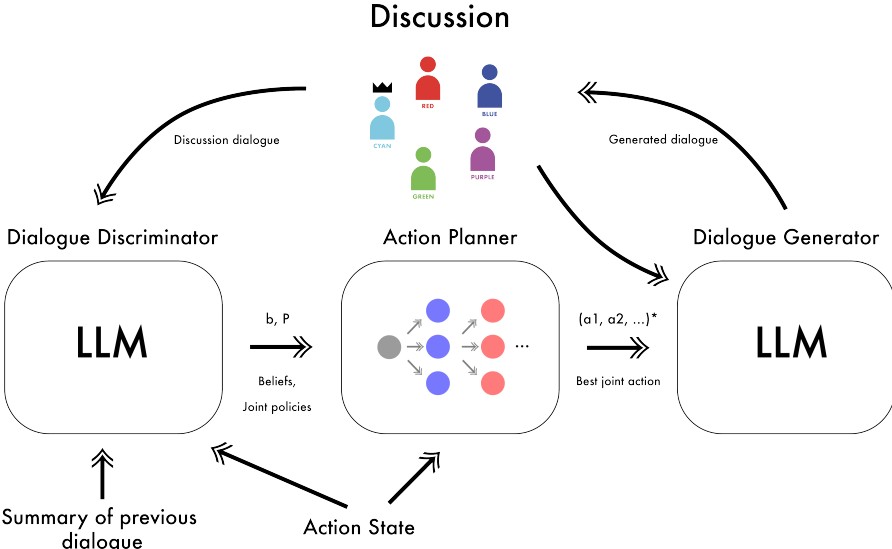

Figure 13: Overview of the LLM-powered agent, including the three main modules that we use to generate dialogue during discussion

the probability of each player being Merlin, both conditioned on the private information contained in $\boldsymbol{I}$. While a full treatment of the distribution over the hidden state space $\mathcal{S}$ we require assigning probabilities to each possible combination of Good and Evil players, not just assessing the marginal probability of each player being Good individually, in practice

We implement $f_{ana}$ using an LLM, which is fed $\boldsymbol{I}$, $\boldsymbol{d}$, $\boldsymbol{b}$ (converted to natural language form) as prompts, along with some instruction prompt $\phi_{ana}$ that prompts it to produce $\widehat{\boldsymbol{b}}, \widehat{\boldsymbol{\Pi}}_t$. Specifically,

$$f_{ana}(\boldsymbol{I}, \boldsymbol{d}_t, \boldsymbol{b}) = f_{LLM}(\phi_{dis}, \boldsymbol{I}, \boldsymbol{d}, \boldsymbol{b})$$

We show examples of such prompts in Appendix J.

### E.2 ACTION PLANNER

Given $\widehat{\boldsymbol{b}}$ the belief prior, $\widehat{\boldsymbol{\Pi}}_t$ the predicted joint action policy for all players, and $\boldsymbol{s}$ the representation of the current state, the action generation model $f_{act}$ generates a probability distribution over possible actions $\boldsymbol{\pi}^i$ for the main player $i$ that is the best response to $\widehat{\boldsymbol{\Pi}}_t$. We do so by using search techniques to look ahead and find the best response.

$$\boldsymbol{\pi}^i = f_{act}(\widehat{\boldsymbol{b}}, \widehat{\boldsymbol{\Pi}}_t, \boldsymbol{I})$$

More specifically, in our search implementation, at the first layer, we first sample across possible hidden states $\boldsymbol{s} \sim \widehat{\boldsymbol{b}}$ according to the belief prior. At the second layer (i.e. the first action stage $t$), we calculate expected $q$-values for each action $a \in \mathcal{A}$ that the main player can take if the other players play actions $\boldsymbol{a} \sim \widehat{\boldsymbol{\Pi}}_t$ according to the predicted joint distribution. In subsequent action stages, the search process will assume that other players play according to their policy simulated and induced by the value heuristic that is not dialogue dependent. We then take the best response action $a_i^* = \max(\boldsymbol{\pi}^i)$ as the intended action. Since this is a partial information game, expected $q$-values are taken across information sets, not states. We describe how our action planner is implemented in more detail in Appendix F.

### E.3 DIALOGUE GENERATION

The dialogue generator $f_{gen}$ takes as input $\boldsymbol{I}$ some representation of the current information set and $a_i^*$, the intended best response action, and outputs dialogue $d$.

$$d = f_{gen}(\boldsymbol{I}, a_i^*)$$

We will implement $f_{gen}$ using an LLM, which is fed $\boldsymbol{I}$ and $a_i^*$ directly as prompts, along with some instruction prompt $\phi_{gen}$ that prompts it to produce realistic sounding dialogue that helps it achieve its intended action.

For example, perhaps the player wants to approve the next team. Then it should try to generated dialogue that convinces the other players to also approve.

We show examples of such prompts in Appendix J.

## F    VALUE HEURISTIC IMPLEMENTATION DETAILS

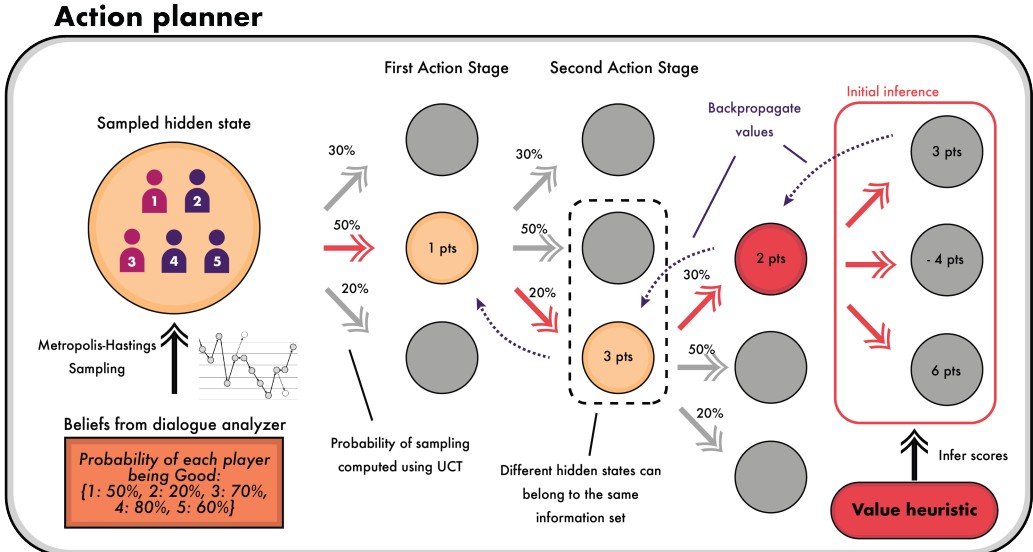

Figure 14: **Overview of how we utilize the trained value heuristics in MCTS tree search to get a non-dialogue based policy**. While we only display the values for a single player in the diagram, note that in practice we infer and update the values for all players at the same time. Next states are sampled using the PUCT formula we described. We sample an initial hidden state based on the internal beliefs of the agent. Metropolis-Hastings is used to sample since it may be difficult to calculate the probability density specified by the internal beliefs. Note that values estimated using MCTS are also passed as feedback to the evaluator.

The MCTS search process is depicted in Figure 14, where we simulate a trajectory from the hidden state we are at until we reach some unexpanded state $s$. The probability of transitioning to a state during simulations is computed assuming that each player samples from their optimal actions according to their PUCT (polynomial upper confidence trees) values (and $\phi_e$ for the environment actor) (Schrittwieser et al., 2020). Since in some environments players may only be able to observe information sets, when computing the PUCT values we average over all expanded states in that information set. Moreover, the initial hidden state can be sampled according to a prior (or empirical prior) over the states in the information set that the player observed. Then, using our value heuristic, we compute the values of each of the next hidden states. We then backpropagate our new values back up the simulated trajectory, updating the intermediate states. After running a few MCTS simulations (roll-outs) like the one we described, the planner then outputs the action which leads to the highest value next state. We show our information set PUCT formula below, where $N(s, a)$ is the number of times we took action $a$ at state $s$ during MCTS rollouts, $P(s, a)$ is the prior prior probability of selecting action $a$ from state $s$, $C$ is the exploration constant, $Q_{emp}$ is the empirical average of MCTS roll-out outcomes, $\widehat{Q}(s, a)$ is the prior computed by our value heuristic, $\alpha$ controls how much weight be put on the prior (often $\alpha = 1$), and $\pi_B$ is the distribution across hidden states in the information set given our **beliefs** $B$, some parametrization of $\pi_B$. Since $\pi_B$ is often hard to compute, we can simply set $\pi_B(s|I) = \frac{\sum_b N(s,b)}{\sum_{s' \in I} \sum_b N(s',b)}$ to be the empirical roll-out distribution, given that we sample initial states $s_0 \sim \pi_B(s_0|I)$ according to our beliefs. For example, in Avalon, we can sample the hidden roles according to our beliefs $B$ using Metropolis-Hastings for the initial state $s_0$.

$$Q(s, a) = \frac{N(s, a) \cdot Q_{\text{emp}}(s, a) + \alpha \cdot \widehat{Q}(s, a)}{N(s, a) + \alpha} \tag{1}$$

$$\text{PUCT}(I, a) = \sum_{s \in I} \pi_B(s|I) \left[ Q(s, a) + C \cdot P(s, a) \cdot \frac{\sqrt{\sum_b N(s, b)}}{1 + N(s, a)} \right] \tag{2}$$

## G DIALOGUE GUIDE IMPROVEMENT EVALUATION IMPLEMENTATION DETAILS

We provide more details on our dialogue improvement evaluation process here and as shown in figure 15. The improvement method (skill coach) remains the same as we described before.

We first generate a synthetic dataset by simulating a game of Avalon with initial dialogue and move policies $\phi$. Given the dialogue guide $\sigma$ we want to evaluate, we then sample 'scenarios' from the dataset. A scenario consists of a game state, intended action, and private information in the simulated trajectory. We create an Avalon agent like the one we described in E for each player in the game, initialized with their corresponding private information. The Avalon agent is then asked to generate dialogue using the dialogue guide $\sigma$.

Using this new generated dialogue, we then simulate the next round of dialogue analysis for each Avalon agent. This produces analysis scores based on how likely they think the player is to be Merlin $z_{merlin}$, and how likely they think the player is to be Evil $z_{evil}$, where $z_{merlin}, z_{evil} \in [-2, 2]$. For evaluating Merlin, we get the average $z_{merlin}$ scores from the Evil players, $\overline{z}_{merlin}$, along with the average $z_{evil}$ scores from the Good players $\overline{z}_{evil}$. We then take the minimum of these two as the feedback score $z = \min\{\overline{z}_{evil}, \overline{z}_{merlin}\}$. This is because Merlin wants to both minimize the probability of being detected by the Evil players, and also minimize the probability of being identified as Evil by the Good players.

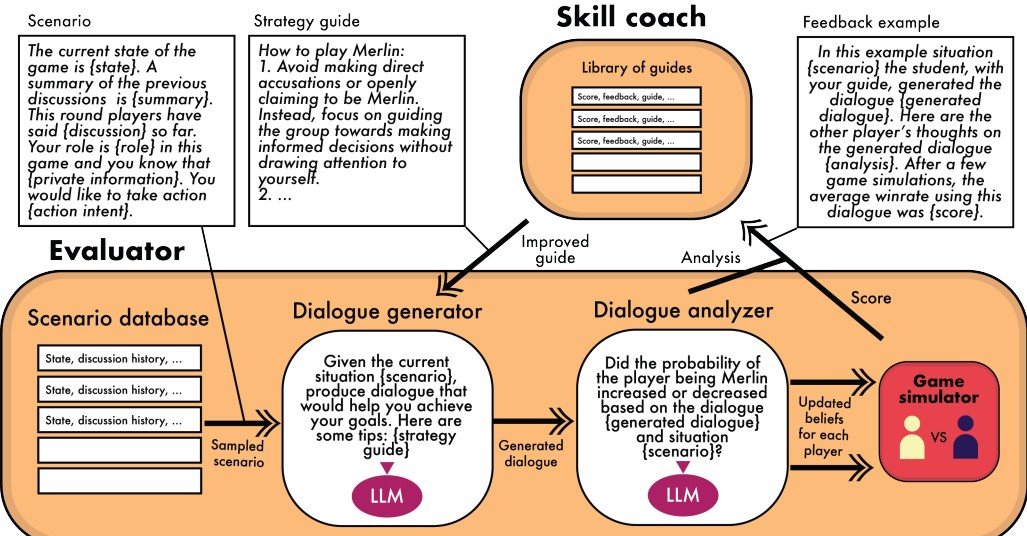

Figure 15: **Overview of our improvement process for learning dialogue generation strategies**. This includes how we evaluate the dialogue and how we collect feedback. The skill coach here can be implemented as either our improvement method, STRATEGIST, or any of the baseline methods we described.

The dialogue analyzer (discriminator) is described in more detail in Appendix E and the specific generation and analysis prompts are shown in Appendix J.

# H  VALUE HEURISTIC LLM PROMPT AND OUTPUT EXAMPLES

## H.1  SYSTEM PROMPTS

System prompt are guidelines for LLM to generate outputs align with the intended goals. In our case, the goal is to generate a function that evaluates the value of a state in a game under low cost.

> **Value heuristic system prompt**
>
> You are a function engineer trying to write a function that can evaluate the value of a state in a game. This is known as a value heuristic, and will be used in look-ahead search algorithms to evaluate the value of unexplored states. Your goal is to develop a heuristic that is as accurate as possible without being too expensive to compute. Hence, you are not allowed to runs simulations in the function.

The following example is a detailed prompt telling the LLM how to format the value heuristics specifically in the GOPS game. The format of input and output are clearly defined in the prompt with illustrations, examples and structures.

---

**GOPS value heuristics function signature**

The function (written in python) should be named 'evaluate state' and take in a tuple called 'state' of the game state as input. Specifically, the input tuple will be of length 9, and it should return 2 elements. The first element should be a tuple with 2 floats: the first element being the score you expect player 0 will get at the end of the game, and the second element being the score you expect player 1 will get at the end of the game. The second element should be a dictionary of any important intermediate values that you used to calculate the scores. For example, if you think player 0 will win 12 total points by the end of the game and player 1 will win 8 total points, the function should return (12, 8).

Make sure your output only includes the code of the function itself in plain text such that it is executable using exec() in python. Any helper functions should be defined within the scope of the function 'evaluate state'. Include comments in your code so that it is readable, but everything should be implemented.

The signature of the function should be as follows:

```python
def evaluate_state(state) -> tuple[tuple[float, float], dict]:
    score_cards = state[0] # a python list of the score cards (integers) that have been played, in the
            order they were played
    player_0_played_cards = state[1] # a python list of the cards (integers) player 0 has played, in the
            order they were played.
    player_1_played_cards = state[2] # a python list of the cards (integers) player 1 has played, in the
            order they were played.
    is_turn = state[3] # bool, true if it is you and your opponent's turn to play, false if it is time
            to draw a new score card
    player_0_score = state[4] # float or integer, player 0's score so far
    player_1_score = state[5] #  float or integer, player 1's score so far
    score_deck = state[6] # a python set of the score cards (integers) left in the deck, either same
            length as player_0_hand and player_1_hand or one less since the score card appears before the
            players play. May be empty
    player_0_hand = state[7] # a python set of the cards (integers) left in player 0's hand. May be
            empty
    player_1_hand = state[8] # a python set of the cards (integers) left in player 1's hand. May be
            empty
    # explanation of what we do next
    ...
    <intermediate_value1> = value1
    # explanation of what we do next
    ...
    <intermediate_value2> = value2
    # explanation of what we do next
    ...
    player_scores = (player_0_expected_score, player_1_expected_score)
    intermediate_values = {'<intermediate_value1>': intermediate_value1, '<intermediate_value2>':
            intermediate_value2, ...}
    return player_scores, intermediate_values # make sure the return is exactly in this format
```

Where you can use your own names for the intermediate values and the values themselves. Please start with "def evaluate state(state):"

## H.2 IDEA GENERATION EXAMPLES

The idea generation prompt included system prompt, game rules, previous guide and feedback reflections. Following those four components, we construct the format and an example of ideas to guide the generation of LLM.

---

**Prompt for idea generation**

```
<System prompt>
<Game rules>
<Previous guide>
<Feedback reflections>
```

Based on the function, feedback, and conclusions you drew, what are 2 improvements that you can make to the function that you think will have the most impact? Be as specific and concrete as possible, and write them out in the following format:

- Thoughts: <your thoughts here>
- Idea 1: <your idea here>
- Idea 2: <your idea here>

...

Here's an example of what this might look like for 3 improvement ideas:

- Thoughts: I should consider the number of cards left in the deck when evaluating the value of a state.
- Idea 1: I should add a term to the value function that penalizes states where there are fewer cards left in the deck.
- Idea 2: I should add a term to the value function that rewards states where the player has more cards in their hand than the opponent.
- Idea 3: I should add a term to the value function that rewards states where the player has more cards in their hand than the opponent and there are fewer cards left in the deck.

---

Below is an instance of Feedback of GOPS game, showing the setup of two players, the intermediate values involved in the computation, and the actual scores.

---

**Feedback example**

Example 9:

The state you were trying to estimate a value for is:

The current state of the game is as follows:

- The score cards that have been revealed are: (2, 4, 5, 1, 3)
- The cards that player 0 has played are: (1, 2, 4, 3, 5)
- The cards that player 1 has played are: (3, 5, 1, 2, 4)
- Player 0's score so far is: 9
- Player 1's score so far is: 6
- The score cards left in the deck are: set()
- The cards left in player 0's hand are: set()
- The cards left in player 1's hand are: set()

The function you generated returned the following values:

{0: 3, 1: -3}

for the expected end of game scores of the players.

Some intermediate values that you used to calculate the scores were:

{'player_0_expected_score': 9, 'player_1_expected_score': 6, 'dynamic_penalty': 0.0, 'player_0_hand_reward': 0, 'player_1_hand_reward': 0, 'player_0_adjustment': 0, 'player_1_adjustment': 0, 'player_0_strategic_adjustment': 0, 'player_1_strategic_adjustment': 0}

The estimated end of game scores of the players using lookahead search with your function was:

{0: 0.0, 1: 0.0}

The actual scores of the players at the end of the game in the simulation were:

{0: 3.0, 1: -3.0}

---

Recall that feedback reflection refers to feedback for strategy, which is used to generate ideas. For prompting, We specify that game rules, previous guide and feedback examples are generated previously. The following is the prompt for feedback reflections.

Prompt for feedback reflections

```
<System prompt>
<Game rules>
```
Previously you generated the following function to evaluate the value of a state in the game.
```
<Previous guide>
```
Below is some feedback on how the function you generated performed when we tested it. Note that simulations involve high variance and the actual scores may not match the expected scores exactly. Hence, you should focus on trying to get the scores produced by your function to match those predicted by look-ahead search as closely as possible.
```
<Feedback examples>
```
Based on the feedback given and the function you generated previously, what are some conclusions you can draw from the feedback? Make sure to cite the specific examples in the feedback to justify your analysis.

Below is the feedback reflections from the LLM.

Feedback reflections

From the feedback provided, we can draw several conclusions regarding the performance of the function in evaluating the value of a state in the GOPS game:

1. **Discrepancy in Expected Scores**: In several examples (e.g., Example 4, Example 7, Example 8), the expected end-of-game scores returned by the function do not match the scores predicted by look-ahead search closely. This indicates that the function may not be accurately estimating the value of the states in the game.

2. **Inconsistencies in Strategic Adjustments**: The function includes a strategic adjustment based on the opponent's played cards, but in all cases, the strategic adjustments are zero. This suggests that the strategic adjustments made in the function may not be capturing the true strategic implications of the opponent's moves effectively.

3. **Dynamic Penalty Adjustment**: The function includes a dynamic penalty adjustment based on the remaining high-value score cards in the deck. However, the dynamic penalty adjustments may not be accurately reflecting the impact of high-value score cards left in the deck on the game outcome.

4. **Limited Hand Size Reward**: The function rewards states where a player has more cards in their hand than the opponent. While this reward is included, it might not be capturing the full strategic implications of having more cards in hand effectively.

5. **Simulation vs. Expected Scores**: The actual scores at the end of the game in the simulations do not match the expected scores returned by the function precisely. This indicates that the function's evaluation may not be aligning well with the actual game outcomes.

In conclusion, the function may need further refinement in terms of capturing strategic adjustments, dynamic penalties, and hand size rewards to improve the accuracy of the state evaluation in the GOPS game. Additionally, considering the discrepancies between the expected and actual scores, further testing and adjustments may be necessary to enhance the function's performance in evaluating the value of states accurately.

Also, with the prompts of idea generation, below is the generated new ideas by LLM.

---

**Generated new ideas**

Thoughts: The function needs to better capture the strategic implications of the opponent's moves and adjust the expected scores accordingly. Additionally, the dynamic penalty adjustment and hand size rewards could be improved to better reflect their impact on the game outcome.

Idea 1: Enhance the strategic adjustment component of the function by analyzing the opponent's played cards more deeply. This could involve considering patterns in the opponent's moves, potential card combinations, and predicting future moves based on past actions.

Idea 2: Revise the dynamic penalty adjustment to more accurately reflect the impact of high-value score cards left in the deck. This could involve adjusting the penalty dynamically based on the remaining high-value cards and their likelihood of being drawn in future rounds. This adjustment could help in better assessing the risk associated with certain states in the game.

---

### H.3 STRATEGY IMPLEMENTATION EXAMPLES

During the strategy implementation step, we first select a strategy and an idea from the libraries using the adaptive selection policy. Then, we prompt the LLM to implement the idea on the strategy, generating a new improved strategy.

In this section, we showcase two examples. Each example illustrates strategies before and after enhancements made by the LLM. This comparison highlights the effectiveness of our strategy implementation. The improved parts by LLM are highlighted in yellow.

Below is the prompt for strategy implementation.

---

**Prompt for implementation**

```
<System prompt>
<Game rules>
```
Previously you generated the following function to evaluate the value of a state in the game:
```
<Previous guide>
```
Here is a possible way to improve this function:
```
<Improvement ideas>
```

---

#### H.3.1 EXAMPLE 1

Below is the GOPS Value Heuristics Function before the strategy improvement.

**GOPS Value Heuristics Function (Before)**

```python
def evaluate_state(state) -> tuple[tuple[float, float], dict]:
    score_cards = state[0]
    player_0_played_cards = state[1]
    player_1_played_cards = state[2]
    is_turn = state[3]
    player_0_score = state[4]
    player_1_score = state[5]
    score_deck = state[6]
    player_0_hand = state[7]
    player_1_hand = state[8]

    # Calculate the potential scores for each player based on the remaining score cards
    player_potential_score = player_0_score
    opponent_potential_score = player_1_score

    if len(score_deck) > 0:
        remaining_score_cards = len(score_deck)
        if len(player_0_played_cards) > len(player_1_played_cards):
            player_potential_score += remaining_score_cards / 2
        elif len(player_1_played_cards) > len(player_0_played_cards):
            opponent_potential_score += remaining_score_cards / 2

    player_advantage = player_potential_score - opponent_potential_score

    return (player_potential_score, opponent_potential_score), {'player_advantage': player_advantage}
```

Below is the selected idea to improve the above strategy.

**Selected GOPS Value Heuristics Improvement Idea**

Update the function to calculate the advantage of winning future rounds based on the potential scores each player could achieve with the remaining score cards and their current scores. This would involve considering the number of cards left in the deck, the players' hands, and the current scores to determine the advantage one player may have over the other in the upcoming rounds.

Below is the GOPS Value Heuristics Function after enhancement by LLM. The improved parts by LLM are highlighted in yellow.

---

**GOPS Value Heuristics Function (After)**

```python
def evaluate_state(state) -> tuple[tuple[float, float], dict]:
    score_cards = state[0]
    player_0_played_cards = state[1]
    player_1_played_cards = state[2]
    is_turn = state[3]
    player_0_score = state[4]
    player_1_score = state[5]
    score_deck = state[6]
    player_0_hand = state[7]
    player_1_hand = state[8]

    # Calculate the potential scores for each player based on the remaining score cards
    # and current scores
    player_potential_score = player_0_score + sum(score_deck) / 2
    opponent_potential_score = player_1_score + sum(score_deck) / 2

    # Determine the advantage of winning future rounds based on potential score
    if is_turn:

        player_potential_scores = [player_potential_score + card for card in player_0_hand]

        opponent_potential_scores = [opponent_potential_score + card for card in player_1_hand]

        player_advantage = sum(player_potential_scores) / len(player_potential_scores)

        - sum(opponent_potential_scores) / len(opponent_potential_scores)

    else:

        player_advantage = player_potential_score - opponent_potential_score

    return (player_potential_score, opponent_potential_score), {'player_advantage': player_advantage}
```

### H.3.2 EXAMPLE 2

Below is the Avalon Value Heuristics Function before improvements.

**Avalon Value Heuristics Function (Before)**

```python
def evaluate_state(state):
    num_successful_quests = sum(historical_quest_results)
    num_failed_quests = len(historical_quest_results) - num_successful_quests
    num_remaining_quests = len(num_participants_per_quest) - len(historical_quest_results)

    num_evil = len(players) - num_good
    num_evil_in_quest_team = len([player for player in quest_team if not is_good[player]])

    success_probability = 0.5
    if phase == 0:
        if num_successful_quests >= 3:
            success_probability = 0.9
        elif num_failed_quests >= 3:
            success_probability = 0.1
    elif phase == 1:
        success_probability = 0.8 if num_evil_in_quest_team == 0 else 0.2
    elif phase == 2:
        success_probability = 0.9 if num_successful_quests > num_failed_quests else 0.1
    elif phase == 3:
        if 'Merlin' in roles and 'Assassin' in roles:
            merlin_index = roles.index('Merlin')
            assassin_index = roles.index('Assassin')
            if assassin_index in quest_team:
                success_probability = 0.1
            else:
                success_probability = 0.9

    expected_winrates_per_player = dict()
    for player in players:
        if is_good[player]:
            expected_winrates_per_player[player] = success_probability
        else:
            expected_winrates_per_player[player] = 1 - success_probability

    intermediate_values = {
        'num_successful_quests': num_successful_quests,
        'num_failed_quests': num_failed_quests,
        'num_remaining_quests': num_remaining_quests,
        'num_evil_in_quest_team': num_evil_in_quest_team
    }

    return expected_winrates_per_player, intermediate_values
```

Below is the improvement idea pf Avalon Value Heuristics Function.

**Avalon Value Heuristics Function Improvement Idea**

Incorporate a more nuanced success probability calculation based on the roles of the players and the current phase of the game. For example, in phase 1 (team selection), consider the balance of Good and Evil players on the quest team and adjust the success probability accordingly. This can help better reflect the strategic considerations during team selection.

The following is Avalon Value Heuristics Function after enhancements, highlighted in yellow.

**Avalon Value Heuristics Function (After)**

```python
def evaluate_state(state: dict) -> tuple[dict[int, float], dict]:

    prob_good_victory = 0.5  # probability that Good will win

    if phase == 0:  # team selection phase
        num_successful_quests = sum(historical_quest_results)
        num_failed_quests = len(historical_quest_results) - num_successful_quests
        num_remaining_quests = len(num_participants_per_quest) - len(historical_quest_results)
        num_evil = len(players) - num_good
        num_evil_in_quest_team = len([player for player in quest_team if not is_good[player]])

        if num_successful_quests >= 3:
            prob_good_victory = 0.9
        elif num_failed_quests >= 3:
            prob_good_victory = 0.1

        intermediate_value1 = {'num_successful_quests': num_successful_quests, 'num_failed_quests':
          num_failed_quests,
                               'num_remaining_quests': num_remaining_quests}
        intermediate_value2 = {'num_evil_in_quest_team': num_evil_in_quest_team}

    elif phase == 1:  # team approval phase
        num_evil_in_quest_team = len([player for player in quest_team if not is_good[player]])

        prob_good_victory = 0.8 if num_evil_in_quest_team == 0 else 0.2

        intermediate_value1 = {'num_evil_in_quest_team': num_evil_in_quest_team}

        intermediate_value2 = {}  # Initialize empty dictionary to avoid the error

    elif phase == 2:  # quest phase
        num_successful_quests = sum(historical_quest_results)
        num_failed_quests = len(historical_quest_results) - num_successful_quests

        prob_good_victory = 0.9 if num_successful_quests > num_failed_quests else 0.1

        intermediate_value1 = {'num_successful_quests': num_successful_quests, 'num_failed_quests':
          num_failed_quests}
        intermediate_value2 = {}  # Initialize empty dictionary to avoid the error
```

```python
    elif phase == 3:  # assassination phase
        merlin_index = roles.index('Merlin')
        assassin_index = roles.index('Assassin')

        if assassin_index in quest_team:
            prob_good_victory = 0.1
        else:
            prob_good_victory = 0.9

    intermediate_value1 = {'merlin_index': merlin_index, 'assassin_index': assassin_index}

    intermediate_value2 = {}  # Initialize empty dictionary to avoid the error

    expected_winrates_per_player = dict()
    prob_evil_victory = 1 - prob_good_victory
    for player in players:
        if is_good[player]:
            expected_winrates_per_player[player] = prob_good_victory
        else:
            expected_winrates_per_player[player] = prob_evil_victory

    intermediate_values = {'intermediate_value1': intermediate_value1, 'intermediate_value2':
        intermediate_value2}

    return expected_winrates_per_player, intermediate_values
```

### H.3.3 EXAMPLE 3

Below is the GOPS Value Heuristics Function before the strategy improvement.

---

**GOPS Value Heuristics Function (Before)**

```python
def evaluate_state(state) -> tuple[tuple[float, float], dict]:
    score_cards = state[0]
    player_0_played_cards = state[1]
    player_1_played_cards = state[2]
    is_turn = state[3]
    player_0_score = state[4]
    player_1_score = state[5]
    score_deck = state[6]
    player_0_hand = state[7]
    player_1_hand = state[8]

    # Calculate initial potentials
    player_0_potential = sum(player_0_hand)
    player_1_potential = sum(player_1_hand)
    score_potential = sum(score_deck)

    # Update player potentials based on remaining cards and score deck
    player_0_potential += sum(card for card in player_0_hand if any(card > score for score in score_deck
        ))
    player_1_potential += sum(card for card in player_1_hand if any(card > score for score in score_deck
        ))

    # Add half of the score potential to the player who has the turn
    if is_turn:
        player_0_potential += score_potential / 2
    else:
        player_1_potential += score_potential / 2

    # Count the number of certain wins for each player
    player_0_certain_wins = sum(card > max(player_1_hand) for card in player_0_hand)
    player_1_certain_wins = sum(card > max(player_0_hand) for card in player_1_hand)

    rounds_left = len(score_deck)

    # Dynamic adjustment based on specific cards played
    player_0_certain_wins_adjust = 0
    player_1_certain_wins_adjust = 0
    for i in range(len(player_0_played_cards)):
        if player_0_played_cards[i] > player_1_played_cards[i]:
            player_0_certain_wins_adjust += 1
        elif player_1_played_cards[i] > player_0_played_cards[i]:
            player_1_certain_wins_adjust += 1

    player_0_certain_wins += player_0_certain_wins_adjust
    player_1_certain_wins += player_1_certain_wins_adjust

    # Add potential scores from certain wins
    if rounds_left <= player_0_certain_wins:
        highest_scores = sorted(score_deck)[-rounds_left:]
        player_0_potential += sum(highest_scores)

    if rounds_left <= player_1_certain_wins:
        highest_scores = sorted(score_deck)[-rounds_left:]
        player_1_potential += sum(highest_scores)
```

```python
    # Calculate expected scores
    player_0_expected_score = player_0_score + player_0_potential
    player_1_expected_score = player_1_score + player_1_potential

    # Calculate the difference in the sum of played cards
    sum_player_0_played_cards = sum(player_0_played_cards)
    sum_player_1_played_cards = sum(player_1_played_cards)
    sum_played_cards_diff = sum_player_0_played_cards - sum_player_1_played_cards

    # Adjust scores based on the difference in the sum of played cards
    adjustment_factor = 0.1  # Adjust this factor based on the level of adjustment desired
    player_0_expected_score += sum_played_cards_diff * adjustment_factor
    player_1_expected_score -= sum_played_cards_diff * adjustment_factor

    # Dynamic adjustment based on uncertainty in the game state
    uncertainty_factor = 0.1 * rounds_left  # Adjust this factor based on the level of uncertainty
        desired
    player_0_expected_score += player_0_potential * uncertainty_factor
    player_1_expected_score += player_1_potential * uncertainty_factor

    # Refine uncertainty factor based on specific conditions
    if len(player_0_hand) == 1:
        uncertainty_factor *= 0.5
    if len(player_1_hand) == 1:
        uncertainty_factor *= 0.5

    # Store intermediate values in a dictionary
    intermediate_values = {
        'player_0_potential': player_0_potential,
        'player_1_potential': player_1_potential,
        'score_potential': score_potential,
        'player_0_certain_wins': player_0_certain_wins,
        'player_1_certain_wins': player_1_certain_wins,
        'player_0_certain_wins_adjust': player_0_certain_wins_adjust,
        'player_1_certain_wins_adjust': player_1_certain_wins_adjust,
        'sum_played_cards_diff': sum_played_cards_diff,
        'uncertainty_factor': uncertainty_factor
    }

    return (player_0_expected_score, player_1_expected_score), intermediate_values
```

Below is the selected idea to improve the above strategy.

**Selected GOPS Value Heuristics Improvement Idea**

Incorporate a term in the value function that takes into account the distribution of score cards left in the deck. Certain score cards may have a higher value or strategic importance, so adjusting the potentials based on the presence of these cards can provide a more accurate evaluation of the state.

Below is the GOPS Value Heuristics Function after enhancement by LLM. The improved parts by LLM are highlighted in yellow.

**GOPS Value Heuristics Function (After)**

```python
def evaluate_state(state) -> tuple[tuple[float, float], dict]:
    score_cards = state[0]
    player_0_played_cards = state[1]
    player_1_played_cards = state[2]
    is_turn = state[3]
    player_0_score = state[4]
    player_1_score = state[5]
    score_deck = state[6]
    player_0_hand = state[7]
    player_1_hand = state[8]

    # Calculate initial potentials
    player_0_potential = sum(player_0_hand)
    player_1_potential = sum(player_1_hand)
    score_potential = sum(score_deck)

    # Update player potentials based on remaining cards and score deck
    player_0_potential += sum(card for card in player_0_hand if any(card > score for score in score_deck
        ))
    player_1_potential += sum(card for card in player_1_hand if any(card > score for score in score_deck
        ))

    # Add half of the score potential to the player who has the turn
    if is_turn:
        player_0_potential += score_potential / 2
    else:
        player_1_potential += score_potential / 2

    # Count the number of certain wins for each player
    player_0_certain_wins = sum(card > max(player_1_hand) for card in player_0_hand)
    player_1_certain_wins = sum(card > max(player_0_hand) for card in player_1_hand)

    rounds_left = len(score_deck)

    # Dynamic adjustment based on specific cards played
    player_0_certain_wins_adjust = 0
    player_1_certain_wins_adjust = 0
    for i in range(len(player_0_played_cards)):
        if player_0_played_cards[i] > player_1_played_cards[i]:
            player_0_certain_wins_adjust += 1
        elif player_1_played_cards[i] > player_0_played_cards[i]:
            player_1_certain_wins_adjust += 1

    player_0_certain_wins += player_0_certain_wins_adjust
    player_1_certain_wins += player_1_certain_wins_adjust

    # Add potential scores from certain wins
    if rounds_left <= player_0_certain_wins:
        highest_scores = sorted(score_deck)[-rounds_left:]
        player_0_potential += sum(highest_scores)

    if rounds_left <= player_1_certain_wins:
        highest_scores = sorted(score_deck)[-rounds_left:]
        player_1_potential += sum(highest_scores)

    # Incorporate distribution of score cards left in the deck

    important_score_cards = {7, 10, 13}  # Example: define important score cards

    player_0_potential += sum(card for card in player_0_hand if card in important_score_cards)

    player_1_potential += sum(card for card in player_1_hand if card in important_score_cards)

    # Calculate expected scores
    player_0_expected_score = player_0_score + player_0_potential
    player_1_expected_score = player_1_score + player_1_potential

    # Calculate the difference in the sum of played cards
    sum_player_0_played_cards = sum(player_0_played_cards)
    sum_player_1_played_cards = sum(player_1_played_cards)
    sum_played_cards_diff = sum_player_0_played_cards - sum_player_1_played_cards
```

```
# Adjust scores based on the difference in the sum of played cards
adjustment_factor = 0.1  # Adjust this factor based on the level of adjustment desired
player_0_expected_score += sum_played_cards_diff * adjustment_factor
player_1_expected_score -= sum_played_cards_diff * adjustment_factor

# Dynamic adjustment based on uncertainty in the game state
uncertainty_factor = 0.1 * rounds_left  # Adjust this factor based on the level of uncertainty
        desired
player_0_expected_score += player_0_potential * uncertainty_factor
player_1_expected_score += player_1_potential * uncertainty_factor

# Refine uncertainty factor based on specific conditions
if len(player_0_hand) == 1:
    uncertainty_factor *= 0.5
if len(player_1_hand) == 1:
    uncertainty_factor *= 0.5

# Store intermediate values in a dictionary
intermediate_values = {
    'player_0_potential': player_0_potential,
    'player_1_potential': player_1_potential,
    'score_potential': score_potential,
    'player_0_certain_wins': player_0_certain_wins,
    'player_1_certain_wins': player_1_certain_wins,
    'player_0_certain_wins_adjust': player_0_certain_wins_adjust,
    'player_1_certain_wins_adjust': player_1_certain_wins_adjust,
    'sum_played_cards_diff': sum_played_cards_diff,
    'uncertainty_factor': uncertainty_factor
}

return (player_0_expected_score, player_1_expected_score), intermediate_values
```

### H.3.4   EXAMPLE 4

Below is the GOPS Value Heuristics Function before the strategy improvement.

**GOPS Value Heuristics Function (Before)**

```python
def evaluate_state(state) -> tuple[tuple[float, float], dict]:
    score_cards = state[0]
    player_0_played_cards = state[1]
    player_1_played_cards = state[2]
    is_turn = state[3]
    player_0_score = state[4]
    player_1_score = state[5]
    score_deck = state[6]
    player_0_hand = state[7]
    player_1_hand = state[8]

    # Calculate initial potentials
    player_0_potential = sum(player_0_hand)
    player_1_potential = sum(player_1_hand)
    score_potential = sum(score_deck)

    # Add half of the score potential to the player who has the turn
    if is_turn:
        player_0_potential += score_potential / 2
    else:
        player_1_potential += score_potential / 2

    # Count the number of certain wins for each player
    player_0_certain_wins = sum(card > max(player_1_hand) for card in player_0_hand)
    player_1_certain_wins = sum(card > max(player_0_hand) for card in player_1_hand)

    rounds_left = len(score_deck)

    # Add potential scores from certain wins
    if rounds_left <= player_0_certain_wins:
        highest_scores = sorted(score_deck)[-rounds_left:]
        player_0_potential += sum(highest_scores)

    if rounds_left <= player_1_certain_wins:
        highest_scores = sorted(score_deck)[-rounds_left:]
        player_1_potential += sum(highest_scores)

    # New improvement: Incorporate a probabilistic approach based on the remaining score cards
    player_0_expected_score = player_0_score + player_0_potential
    player_1_expected_score = player_1_score + player_1_potential

    # Dynamic evaluation of opponent's potential moves
    opponent_potential = sum(player_0_hand) if is_turn else sum(player_1_hand)
    opponent_certain_wins = sum(card > max(player_1_hand) for card in player_0_hand) if is_turn else sum(
        (card > max(player_0_hand) for card in player_1_hand)

    # Adjust evaluation based on opponent's potential plays
    if rounds_left <= opponent_certain_wins:
        highest_scores = sorted(score_deck)[-rounds_left:]
        if is_turn:
            player_1_potential += sum(highest_scores)
        else:
            player_0_potential += sum(highest_scores)

    # Store intermediate values in a dictionary
    intermediate_values = {
        'player_0_potential': player_0_potential,
        'player_1_potential': player_1_potential,
        'score_potential': score_potential,
        'player_0_certain_wins': player_0_certain_wins,
        'player_1_certain_wins': player_1_certain_wins,
        'opponent_potential': opponent_potential,
        'opponent_certain_wins': opponent_certain_wins,
        'player_0_expected_score': player_0_expected_score,
        'player_1_expected_score': player_1_expected_score
    }

    return (player_0_expected_score, player_1_expected_score), intermediate_values
```

Below is the selected idea to improve the above strategy.

**Selected GOPS Value Heuristics Improvement Idea**

Introduce a penalty factor for potential scores that are excessively high, indicating potential overestimation. By capping the potential scores or applying a scaling factor based on certain thresholds, we can prevent the function from assigning unrealistic values and improve the accuracy of the value heuristic.

Below is the GOPS Value Heuristics Function after enhancement by LLM. The improved parts by LLM are highlighted in yellow.

**GOPS Value Heuristics Function (After)**

```python
def evaluate_state(state) -> tuple[tuple[float, float], dict]:
    score_cards = state[0]
    player_0_played_cards = state[1]
    player_1_played_cards = state[2]
    is_turn = state[3]
    player_0_score = state[4]
    player_1_score = state[5]
    score_deck = state[6]
    player_0_hand = state[7]
    player_1_hand = state[8]

    # Calculate initial potentials
    player_0_potential = sum(player_0_hand)
    player_1_potential = sum(player_1_hand)
    score_potential = sum(score_deck)

    # Add half of the score potential to the player who has the turn
    if is_turn:
        player_0_potential += score_potential / 2
    else:
        player_1_potential += score_potential / 2

    # Count the number of certain wins for each player
    player_0_certain_wins = sum(card > max(player_1_hand) for card in player_0_hand)
    player_1_certain_wins = sum(card > max(player_0_hand) for card in player_1_hand)

    rounds_left = len(score_deck)

    # Add potential scores from certain wins
    if rounds_left <= player_0_certain_wins:
        highest_scores = sorted(score_deck)[-rounds_left:]
        player_0_potential += sum(highest_scores)

    if rounds_left <= player_1_certain_wins:
        highest_scores = sorted(score_deck)[-rounds_left:]
        player_1_potential += sum(highest_scores)
```

```python
    # New improvement: Incorporate a probabilistic approach based on the remaining score cards
    player_0_expected_score = player_0_score + player_0_potential
    player_1_expected_score = player_1_score + player_1_potential

    # Dynamic evaluation of opponent's potential moves
    opponent_potential = sum(player_0_hand) if is_turn else sum(player_1_hand)
    opponent_certain_wins = sum(card > max(player_1_hand) for card in player_0_hand) if is_turn else sum
        (card > max(player_0_hand) for card in player_1_hand)

    # Adjust evaluation based on opponent's potential plays
    if rounds_left <= opponent_certain_wins:
        highest_scores = sorted(score_deck)[-rounds_left:]
        if is_turn:
            player_1_potential += sum(highest_scores)
        else:
            player_0_potential += sum(highest_scores)

    # Introduce a penalty factor for excessively high potential scores

    penalty_threshold = 100  # Define a threshold for potential scores to trigger penalty

    penalty_factor = 0.5  # Define a factor by which to reduce potential scores above threshold

    if player_0_potential > penalty_threshold:

        player_0_potential = penalty_threshold + (player_0_potential - penalty_threshold) *

            penalty_factor

    if player_1_potential > penalty_threshold:

        player_1_potential = penalty_threshold + (player_1_potential - penalty_threshold) *

            penalty_factor

    # Store intermediate values in a dictionary
    intermediate_values = {
        'player_0_potential': player_0_potential,
        'player_1_potential': player_1_potential,
        'score_potential': score_potential,
        'player_0_certain_wins': player_0_certain_wins,
        'player_1_certain_wins': player_1_certain_wins,
        'opponent_potential': opponent_potential,
        'opponent_certain_wins': opponent_certain_wins,
        'player_0_expected_score': player_0_expected_score,
        'player_1_expected_score': player_1_expected_score
    }

    return (player_0_expected_score, player_1_expected_score), intermediate_values
```

# I  DIALOGUE GUIDE LLM PROMPT AND OUTPUT EXAMPLES

This sections shows the system prompts of dialogue guidance on LLM and several examples, including system prompts, idea generation prompts, and strategy implementation examples.

## I.1  SYSTEM PROMPTS

Below is the Dialogue guide system prompt.

---

**Dialogue guide system prompt**

You are a coach trying to write a section of a strategy guide on how to play a game well.

The specific section of the strategy guide you are writing right now is on how to play the Merlin role effectively during the discussion phase so that they can win the game. Recall that players often use the discussion phase to (1) gather information about other players, (2) try to convince other players of their innocence or guilt, and (3) try to persuade other players of a particular course of action. The game you are interested in is called The Resistance: Avalon. The Resistance: Avalon is the game of hidden identities and social deduction. There are two teams in the game: Good and Evil. Each player has a hidden identity (role) and side.

---

Dialogue guide signature follows, specifying the structures and contents dialogues regarding role playing the Merlin.

---

**Dialogue guide signature**

Your guide should be in the form of a worksheet that the student can use to build their speech. You should order the worksheet questions in a way that makes logical sense, and you should have no more than six questions. Your questions should instruct the reader to write parts of their speech.

The title of your section should be "Questions to fill out before speaking as the Merlin role". Below is an example of how your worksheet should look like:

1. Questions to fill out before speaking as the Merlin role
2. Q1: Which player seems the most suspicious of you and why?
3. Q2: For the player that seems the most suspicious of you, produce a statement addressing their suspicious.
4. Q3: Which player is the quest leader?
5. Q4: Produce a statement addressing the quest leader to convince them to support your intended course of action/ desired team.
6. Q5: Which player is the most supportive of you?
7. Q6: Produce a statement addressing the supportive player to convince them to support your intended course of action/ desired team.

---

> **Example Generated Dialogue Strategy Guide for Assassin**
>
> 1. Q1: Which player seems to have a deeper understanding of the game flow than normal Servants should possess?
>
> 2. Q2: Develop a non-confrontational statement to subtly challenge this player. This should cause them to either prove their innocence or reveal more clues about their identity.
>
> 3. Q3: Who has been the most influential in the team selection and voting process?
>
> 4. Q4: Devise a statement to express agreement with this player's viewpoint subtly. This should make you less suspicious while enabling you to manipulate the discussion.
>
> 5. Q5: Which player seems the most supportive of your views and actions in the game?
>
> 6. Q6: Craft a statement subtly emphasizing your alignment with this supportive player's thoughts. This should increase your chances of being included in quest teams and reduce suspicion around you.

## I.2 IDEA GENERATION EXAMPLES

The following is the selected dialouge guide speaking as the Merlin role.

> **Select dialogue guide**
>
> Questions to fill out before speaking as the Merlin role
>
> 1. Who are the players you suspect are on the Evil side and why?
>
> 2. Draft a subtle statement that casts doubt on these players without directly accusing them.
>
> 3. Who is the current round leader and which players are you considering to suggest for the team?
>
> 4. Create a subtle suggestion that influences the leader's team decision towards your preferred team without revealing your knowledge.
>
> 5. What are the voting patterns you've observed and what do these patterns indicate about the players' identities?
>
> 6. Analyze the voting patterns and team compositions proposed by each player to identify potential alliances or suspicious behaviors. Craft a statement to subtly address these observations without revealing your role.
>
> 7. Which player seems the most supportive of Good and how can you align with them during the discussion phase to better ensure success for your team?

A feedback example is provided for better understanding of dialogue guidance to teach LLM how to role play Merlin.

---

**Feedback example description**

Example 2:

The student using your guide was playing as player 0 in this example filled out your guide as follows:

1. Q1: Based on the statements made so far, I suspect Players 2 and 4 are on the Evil side. Player 2's suggestion of a team consisting of Player 1 and Player 2 seems too convenient, possibly indicating a desire to control the team composition. Player 4's agreement with Player 1 and Player 2 without offering any alternative team suggestions raises suspicion as well.

2. Q2: "I can't help but wonder if some players are trying to steer the team selection process in a particular direction. It's important for us to carefully consider all perspectives and not be swayed by potentially self-serving suggestions."

3. Q3: The current round leader is Player 1. I would suggest a team consisting of Player 0 (Merlin) and Player 1, as this would allow me to closely observe the actions of the other players and potentially guide the team towards success.

4. Q4: "Player 1, I believe that a team comprised of Player 0 and Player 1 would provide a strong foundation for success. By working closely together, we can ensure that our mission is carried out effectively."

5. Q5: The voting patterns so far show agreement among Players 1, 2, and 4, indicating a potential alliance or shared goals. Player 3's suggestion of including themselves in the team may also indicate a desire to control the outcome of the quest.

6. Q6: "I've noticed a pattern of agreement among certain players in the team selection process. It's important for us to consider all perspectives and not be swayed by potentially biased suggestions. Let's focus on working together as a team to achieve our common goal."

7. Q7: Player 1 seems the most supportive of Good based on their consistent emphasis on unity and success for the Quests. I will align with Player 1 during the discussion phase to ensure that our team is selected and the quest is completed successfully.

Then they said the following during the discussion phase:

"Fellow players, as we embark on this quest, it is crucial that we approach the team selection process with caution and consideration. I have noticed a pattern of agreement among certain players, which could potentially indicate a desire to steer the outcome in a particular direction. It is important for us to carefully consider all perspectives and not be swayed by potentially biased suggestions.

I believe that a team comprised of Player 0 and Player 1 would provide a strong foundation for success. By working closely together, we can ensure that our mission is carried out effectively. Player 1 has consistently emphasized unity and success for the Quests, making them a trustworthy ally in this endeavor.

Let us focus on working together as a team to achieve our common goal, rather than allowing personal agendas to influence our decisions. Let us approach this quest with unity and determination, ensuring that Good prevails in the end. Thank you for your attention."

Below is some feedback from the other players on how player 0 performed during the discussion phase. Recall that Good players are trying to access which other players are likely to be Good, while Evil players are trying to access which Good player is likely to be Merlin.

Player 3 with role Servant had the following thoughts on this discussion round:

Thought:
Player 1's speech emphasizes unity and success for the Quests, which aligns with the goals of Good and decreases the probability of Player 1 being Evil. Player 2's statement also focuses on unity and careful consideration, indicating a commitment to the success of Good and decreasing the probability of Player 2 being Evil. Player 3's response as a Servant of Arthur is focused on the success of the Quests and unity, which aligns with the goals of Good and decreases the probability of Player 3 being Evil. Player 4's speech seems overly enthusiastic and focused on unity and cooperation, which could be a tactic to divert attention from their true identity as an Evil player. This raises suspicion and increases the probability of Player 4 being Evil.

Dictionary: {0: (-1, 'decreased slightly'), 1: (-1, 'decreased slightly'), 2: (0, 'stayed the same'), 3: (-2, 'decreased significantly'), 4: (2, 'increased significantly')}

Player 4 with role Assassin had the following thoughts on this discussion round:

Thought:
Player 2 seems to suspect the Evil players (Player 3 and Player 4) without any evidence and is trying to steer the conversation towards them. This aligns with the behavior of Merlin, who knows the identity of the Evil players and is trying to subtly guide the Good players towards the correct decisions. This increases the probability of Player 2 being Merlin. On the other hand, Player 3's speech seems generally clueless and lacks any strategic insight, which decreases the probability of Player 3 being Merlin.

Dictionary: {0: (0, 'stayed the same'), 1: (0, 'stayed the same'), 2: (2, 'increased significantly'), 3: (-2, 'decreased significantly'), 4: (-2, 'decreased significantly')}

This is the prompt for feedback reflections.

### Prompt for feedback reflections

```
<System prompt>
<Game rules>
```
You previously generated the following section of the strategy guide: `<Previous guide>`
Below is some feedback on how your guide performed when a student used it to play the game:
```
<Feedback examples>
```
Based on the feedback given and the guide section you generated previously, what are some conclusions you can draw from the feedback? Make sure to cite the specific examples in the feedback to justify your analysis.

The following are the feedback reflections generated by LLM.

---

**Generated feedback reflections**

In Example 0, Player 0 effectively followed the guide by suspecting certain players of being on the Evil side, crafting subtle statements to cast doubt on them, and aligning with players supportive of Good. This led to Player 0 being perceived positively by other players, with Player 2's probability of being Merlin decreasing significantly due to their behavior aligning more with the Evil side.

In Example 1, Player 0 again effectively followed the guide by suspecting players, making subtle statements, aligning with a supportive player, and emphasizing unity and success in their speech. This resulted in Player 0 being perceived positively by other players, with Player 4's probability of being Merlin decreasing significantly due to their lack of strategic insight.

In Example 2, Player 0 successfully identified suspicious behavior in other players, made subtle statements, and aligned with a supportive player. This led to Player 0 being perceived positively, with Player 2's probability of being Merlin decreasing significantly due to their behavior aligning more with the Evil side.

In Example 3, Player 0 effectively followed the guide by suspecting players, making subtle statements, aligning with a supportive player, and emphasizing unity. This resulted in Player 0 being perceived positively by other players, with Player 4's probability of being Merlin increasing significantly due to their behavior raising suspicion.

- Overall, the guide's emphasis on analyzing player behavior, making subtle statements, and aligning with supportive players has proven to be successful in helping players navigate the discussion phase effectively as the Merlin role in The Resistance: Avalon.
- Players who followed the guide were able to strategically assess the situation, cast doubt on potential Evil players, and align with Good players to increase their chances of success.

---

Below is the Prompt for idea generation regarding this dialogue guides

---

**Prompt for idea generation**

```
<System prompt>
<Game rules>
```
You previously generated the following section of the strategy guide:
```
<Previous guide>
```
Here is a possible way to improve your guide:
```
<Feedback reflections>
```

Based on the worksheet, feedback, and conclusions you drew, what are 2 improvements that you can make to the worksheet that you think will have the most impact?
Be as specific and concrete as possible, including what questions to add, edit, or remove, and write them out in the following format:

- Thoughts: <your thoughts here>

- Idea 1: <your idea here>

- Idea 2: <your idea here>

- ...

Here's an example of what this might look like for 3 improvement ideas:

- Thoughts: I should tell the reader to address each player individually in the guide.

- Idea 1: Add a question asking the reader who they think is most suspicious of them and produce a statement addressing their suspicions.

- Idea 2: Add a question asking the reader to consider which other player is most supportive of them and produce a statement addressing their support.

- Idea 3: Add a question asking the reader produce a statement addressed

---

Generated new improvement ideas are demonstrated below, showing the improvements made by LLM following the prompt for idea generation.

---

**Generated new improvement ideas**

Thoughts: The current worksheet focuses on analyzing player behavior, making subtle statements, and aligning with supportive players, which are all crucial strategies for playing the Merlin role effectively. To enhance the guide further, we can add questions that prompt the player to consider their own actions and statements more explicitly.

1. Idea 1: Add a question asking the player to reflect on how their proposed team composition would benefit the success of the Quest and provide a statement justifying their choices.

2. Idea 2: Add a question asking the player to assess their level of suspicion towards each player and craft a statement addressing how they will interact with those players during the discussion phase.

3. Idea 3: Add a question asking the player to consider how they can subtly guide the discussion towards a successful Quest.

---

## I.3 STRATEGY IMPLEMENTATION EXAMPLES

Recall the strategy implementation mentioned in the paper before. Here, we implement strategy improvements on dialogue guide. The improved dialogue guide is highlighted.

### I.3.1 EXAMPLE 1

Below is the dialogue guide before any improvements happened.

---

**Dialogue guide (previous)**

Questions to fill out before speaking as the Merlin role

1. Who are the players you suspect are on the Evil side and why?

2. Draft a subtle statement that casts doubt on these players without directly accusing them.

3. Who is the current round leader and which players are you considering to suggest for the team?

4. Create a subtle suggestion that influences the leader's team decision towards your preferred team without revealing your knowledge.

5. What are the voting patterns you've observed and what do these patterns indicate about the players' identities?

6. Analyze the voting patterns and team compositions proposed by each player to identify potential alliances or suspicious behaviors. Craft a statement to subtly address these observations without revealing your role.

7. Which player seems the most supportive of Good and how can you align with them during the discussion phase to better ensure success for your team?

---

Below is the improvement idea prompted into LLM.

---

**Improvement idea**

Add a question asking the player to assess their level of suspicion towards each player and craft a statement addressing how they will interact with those players during the discussion phase.

---

Here, implementation prompty follows to teach LLM how to improve the dialogue guide.

---

**Implementation prompt**

```
<System prompt>
<Game rules>
```
You previously generated the following section of the strategy guide:
```
<Previous guide>
```
Here is a possible way to improve your guide:
```
<Improvement idea>
```
Implement this improvement into the guide section as best as you can, but do not change the original guide too much. Make sure to stay within the scope of the guide section, which we reiterate below:
```
<Guide signature>
```

---

Finally, we have a improved dialogue guide, where the changes are highlighted below comparing to the previous guide.

---

**Dialogue guide (improved)**

Questions to fill out before speaking as the Merlin role

1. Who are the players you suspect are on the Evil side and why?
2. Draft a subtle statement that casts doubt on these players without directly accusing them.
3. Who is the current round leader and which players are you considering to suggest for the team?
4. Create a subtle suggestion that influences the leader's team decision towards your preferred team without revealing your knowledge.
5. What are the voting patterns you've observed and what do these patterns indicate about the players' identities?
6. Analyze the voting patterns and team compositions proposed by each player to identify potential alliances or suspicious behaviors. Craft a statement to subtly address these observations without revealing your role.
7. Which player seems the most supportive of Good and how can you align with them during the discussion phase to better ensure success for your team?
8. Assess your level of suspicion towards each player and craft a statement addressing how you will interact with those players during the discussion.

---

### I.3.2  EXAMPLE 2

Below is the dialogue guide before any improvements happened.

---

**Dialogue guide (previous)**

Questions to fill out before speaking as the Merlin role

1. Who are the players you suspect are on the Evil side and why?
2. Draft a subtle statement that casts doubt on these players without directly accusing them.
3. Who is the current round leader and which players are you considering to suggest for the team?
4. Create a subtle suggestion that influences the leader's team decision towards your preferred team without revealing your knowledge.
5. What are the voting patterns you've observed and what do these patterns indicate about the players' identities?
6. Craft a statement to share these observations subtly without giving away your role.

---

Below is the improvement idea prompted into LLM.

---

**Improvement idea**

Add a question asking the reader to identify which players they feel most aligned with in terms of strategy and decision-making, and how they can subtly align themselves with these players during the discussion phase.

---

Here, implementation prompt follows to teach LLM how to improve the dialogue guide.

---

**Implementation prompt**

```
<System prompt>
<Game rules>
```
You previously generated the following section of the strategy guide:
```
<Previous guide>
```
Here is a possible way to improve your guide:
```
<Improvement idea>
```
Implement this improvement into the guide section as best as you can, but do not change the original guide too much. Make sure to stay within the scope of the guide section, which we reiterate below:
```
<Guide signature>
```

---

Finally, we have a improved dialogue guide, where the changes are highlighted below comparing to the previous guide

---

**Dialogue guide (improved)**

Questions to fill out before speaking as the Merlin role

1. Who are the players you suspect are on the Evil side and why?
2. Draft a subtle statement that casts doubt on these players without directly accusing them.
3. Who is the current round leader and which players are you considering to suggest for the team?
4. Create a subtle suggestion that influences the leader's team decision towards your preferred team without revealing your knowledge.
5. What are the voting patterns you've observed and what do these patterns indicate about the players' identities?
6. Craft a statement to share these observations subtly without giving away your role.
7. Which player seems the most aligned with your strategic decisions and why?
8. Produce a statement to subtly align yourself with the player you feel most aligned with during the discussion phase.

---

### I.3.3  EXAMPLE 3

Below is the dialogue guide before any improvements happened.

---

**Dialogue guide (previous)**

Questions to fill out before speaking as the Merlin role

1. Who are the players you suspect are on the Evil side and why?
2. Draft a subtle statement that casts doubt on these players without directly accusing them.
3. Who is the current round leader and which players are you considering to suggest for the team?
4. Create a subtle suggestion that influences the leader's team decision towards your preferred team without revealing your knowledge.
5. What are the voting patterns you've observed and what do these patterns indicate about the players' identities?
6. Craft a statement to share these observations subtly without giving away your role.
7. Which player seems the most aligned with your strategic decisions and why?
8. Produce a statement to subtly align yourself with the player you feel most aligned with during the discussion phase.

---

Below is the improvement idea prompted into LLM.

---

**Improvement idea**

Add a question asking the reader to analyze the voting patterns and craft a statement subtly highlighting any inconsistencies or suspicious trends in the voting behavior of specific players.

---

Here, implementation prompt follows to teach LLM how to improve the dialogue guide.

---

**Implementation prompt**

```
<System prompt>
<Game rules>
```
You previously generated the following section of the strategy guide:
```
<Previous guide>
```
Here is a possible way to improve your guide:
```
<Improvement idea>
```
Implement this improvement into the guide section as best as you can, but do not change the original guide too much. Make sure to stay within the scope of the guide section, which we reiterate below:
```
<Guide signature>
```

---

Finally, we have a improved dialogue guide, where the changes are highlighted below comparing to the previous guide

> **Dialogue guide (improved)**
>
> Questions to fill out before speaking as the Merlin role
>
> 1. Who are the players you suspect are on the Evil side and why?
> 2. Draft a subtle statement that casts doubt on these players without directly accusing them.
> 3. Who is the current round leader and which players are you considering to suggest for the team?
> 4. Create a subtle suggestion that influences the leader's team decision towards your preferred team without revealing your knowledge.
> 5. What are the voting patterns you've observed and what do these patterns indicate about the players' identities?
> 6. Craft a statement to share these observations subtly without giving away your role.
> 7. Which player seems the most aligned with your strategic decisions and why?
> 8. Produce a statement to subtly align yourself with the player you feel most aligned with during the discussion phase.
> 9. Analyze the voting patterns and craft a statement subtly highlighting any inconsistencies or suspicious trends in the voting behavior of specific players.

# J    DIALOGUE GENERATION AND ANALYSIS EXAMPLE LLM INPUTS AND OUTPUTS

This sections shows the examples of dialogue generation and analysis, including descriptive prompts, dialogue generation, and dialogue analysis.

## J.1    DESCRIPTIVE PROMPTS

Below is the discussion history prompted into LLM for better knowledge of dialogue generation.

---

**Discussion history**

Here is a summary of previous rounds of discussion so far:

- Player 1: "Greetings, fellow players. As a Servant of Arthur, my primary goal is to ensure the success of our Quests. I am committed to working together with all of you to achieve victory for Good. I believe that for this first Quest, a team consisting of Player 1 and Player 4 would be a strong choice. I hope for your support in approving this team, as I am confident that we can work effectively together to accomplish our goal. I encourage everyone to share their thoughts and strategies openly. Let's work together to assess each other's intentions and make the best decisions for the success of our Quest. Remember, the fate of Camelot rests in our hands, and we must be vigilant against the forces of Evil."

- Player 2: "Greetings, noble companions. I echo Player 1's sentiment in working together for the success of our Quests. I believe it is essential to carefully consider our team choices to ensure victory for Good. I suggest that for this first Quest, a team consisting of Player 1 and Player 2 would provide a strong foundation for success. Together, we can overcome any challenges that come our way. Let us unite in our efforts and show our unwavering dedication to the cause. I urge all players to carefully evaluate each team proposal and remember that unity is key to thwarting the forces of Evil."

- Player 3: "Greetings, noble companions. I echo the sentiments shared by Player 1 and Player 2 regarding our shared goal of ensuring the success of our Quests. Unity and cooperation will be key in our endeavors to protect Camelot from the forces of Evil. As we deliberate on the composition of our team for this first Quest, I would like to offer a perspective that considers the balance and strength of our choices. A team comprised of Player 1 and Player 3 would bring together a diverse set of skills and viewpoints that can greatly contribute to our success. By including Player 3 in the team, we can ensure a comprehensive approach to overcoming any challenges that may arise. Let us work together in harmony, leveraging our individual strengths for the greater good of Camelot. I urge my fellow players to consider this team proposal thoughtfully, as I believe it holds great promise for the success of our Quest."

- Player 4: "Greetings, honorable players. I appreciate the dedication and commitment displayed by Player 1, Player 2, and Player 3 towards our common goal of ensuring the success of this Quest. Unity and cooperation are indeed vital in our quest to safeguard Camelot. I believe that the proposed team of Player 1 and Player 2 presents a strong foundation for success. Their combined skills and perspectives can greatly benefit our mission, and I urge all players to consider this team thoughtfully for the betterment of our cause. Let us unite our strengths and work together seamlessly to overcome any challenges that may arise. By standing united, we can ensure the triumph of Good over the forces of Evil. I urge all players to support the team of Player 1 and Player 2 for the success of this Quest."

---

State description shows LLM which state it will be for role playing and dialogue generation.

---

**State description**

You are Player 0, with identity Merlin. You are on the side of Good. The Evil players are Players 2 and 4. Please do not forget your identity throughout the game.

The current state of the game is as follows:

- The number of players in the game is: 5
- This is the quest number 0 which requires 2 players and 1 fails to fail
- This is the 0 round of discussion
- The previous results for the quest were () (True for Success, False for Fail)
- The current phase of the game is the team selection phase
- The current leader is player 1

You would like the following team to be approved: [0, 1]

---

## J.2 DIALOGUE GENERATION

Below is the prompt for filling out worksheet. Filling out this worksheet assists LLM generated better dialogue aligned with the role, Merlin as in the example.

---

**Prompt for filling out worksheet**

```
<System prompt>
<Game rules>
<Discussion history>
<State description>
```

Given the discussion history, current state, and your intended action, answer the above worksheet to the best of your ability. The answers should be based on the information you have and your own reasoning.

Six questions to fill out before speaking as the Merlin role

Q1: Who are the players you suspect are on the Evil side and why?

Q2: Draft a subtle statement that casts doubt on these players without directly accusing them.

Q3: Who is the current round leader and which players are you considering to suggest for the team?

Q4: Create a subtle suggestion that influences the leader's team decision towards your preferred team without revealing your knowledge.

Q5: What are the voting patterns you've observed and what do these patterns indicate about the players' identities?

Q6: Craft a statement to share these observations subtly without giving away your role.

---

The worksheet filled out by LLM follows, demonstrating the LLM capability and the effectiveness of prompting.

---

**LLM filled out worksheet**

Q1: I suspect Players 2 and 4 are on the Evil side because they both proposed the same team composition of Player 1 and Player 2, which seems too coordinated and potentially trying to exclude other players from the team.

Q2: "I've noticed a strong push for the same team composition from certain players, which may raise questions about their motivations and potential alliances within the group."

Q3: The current round leader is Player 1. I would suggest Player 0 and Player 1 for the team, as I believe this composition would provide a strong foundation for success and help me subtly guide the decision towards my preferred team.

Q4: "Player 1, I have a feeling that selecting Player 0 and Player 1 for the team could bring a diverse set of skills and viewpoints that may greatly contribute to our success. Let's consider all options carefully for the betterment of our cause."

Q5: The voting patterns have shown a strong push for the team composition proposed by Players 1 and 2, indicating potential coordination or alignment between these players. Player 3's suggestion of including themselves in the team also raises suspicions of wanting to secure a spot on the quest.

Q6: "I've noticed a pattern in the team composition suggestions and voting behaviors that may reveal certain alliances or motivations within the group. It's important for us to carefully evaluate all options and consider the implications for the success of our quest."

---

Below is the prompt for speech generation. Worksheet completion assists LLM to understand the states and role. The following prompt asks LLM to assemble responses to the questionaire filled out above.

---

**Prompt for speech generation**

```
<System prompt>
<Filled out worksheet>
```

Assemble your responses to the questionaire you just answered into a speech that would help you achieve your intent. Note that dialogue will be seen by all players in the game, so you should not reveal your identity.

Your speech:

---

Finally we have LLM generated speech after the preparations.

---

**LLM generated speech**

I've noticed a strong push for the same team composition from certain players, which may raise questions about their motivations and potential alliances within the group. The voting patterns have shown a strong push for the team composition proposed by Players 1 and 2, indicating potential coordination or alignment between these players. Player 3's suggestion of including themselves in the team also raises suspicions of wanting to secure a spot on the quest. It's important for us to carefully evaluate all options and consider the implications for the success of our quest. Let's consider all options carefully for the betterment of our cause. I have a feeling that selecting Player 0 and Player 1 for the team could bring a diverse set of skills and viewpoints that may greatly contribute to our success. As the current round leader, Player 1, I would suggest Player 0 and Player 1 for the team, as I believe this composition would provide a strong foundation for success and help me subtly guide the decision towards my preferred team. Thank you for considering my input.

---

J.3 DIALOGUE ANALYSIS

After dialogue generation, we continue have LLM assume the role of Merlin, and dialogue analysis is conducted.

Below is the prompt for dialogue analysis regarding to the role of Merlin. A thought is included to guide LLM analyze.

---

**Prompt for dialogue analysis: Merlin**

```
<System prompt>
<Game rules>
<Discussion history>
<State description>
```

Based on the discussions this round, for each player in players 0, 1, 2, 3, 4, do you think the probability of the player being Merlin (2) increased significantly (1) increased slightly (0) stayed the same (-1) decreased slightly or (-2) decreased significantly and why? Begin with your reasoning and end with a dictionary of player to your multiple choice answer, i.e. {0: (2, 'increased significantly'), ..., 4: (0, 'stayed the same')}. Make sure you include quotes around the dictionary values that are strings.

Here's an example:

Thought:
Player 2 seems to suspect the Evil players (Player 3 and Player 4) without any evidence and is trying to steer the conversation towards them. This aligns with the behavior of Merlin, who knows the identity of the Evil players and is trying to subtly guide the Good players towards the correct decisions. This increases the probability of Player 2 being Merlin. On the other hand, Player 4's speech seems generally clueless and lacks any strategic insight, which decreases the probability of Player 4 being Merlin.

Dictionary:
{0: (0, 'stayed the same'), 1: (0, 'stayed the same'), 2: (2, 'increased significantly'), 3: (-2, 'decreased significantly'), 4: (0, 'stayed the same')}

---

Generated dialogue analysis follows. The role is Merlin.

---

**LLM generated dialogue analysis: Merlin**

Thought:
Player 2 is actively participating in the discussion and proposing team compositions, which is a behavior that could align with the role of Merlin, who knows the Evil players and wants to guide the Good players towards success. This increases the probability of Player 2 being Merlin. Player 4, on the other hand, seems to be going along with the proposed teams without much strategic insight, which decreases the probability of Player 4 being Merlin.

Dictionary:
{0: (0, 'stayed the same'), 1: (0, 'stayed the same'), 2: (2, 'increased significantly'), 3: (0, 'stayed the same'), 4: (-2, 'decreased significantly')}

---

Under dialogue analysis, besides focusing on one specific role, we also have LLM target on good and evil in the game, which is broader in terms of role playing. Below is the prompt for dialogue analysis regarding to Good and Evil.

---

**Prompt for dialogue analysis: Good and Evil**

```
<System prompt>
<Game rules>
<Discussion history>
<State description>
```

Based on the discussions this round, for each player in players 0, 1, 2, 3, 4, do you think the probability of the player being Evil (2) increased significantly (1) increased slightly (0) stayed the same (-1) decreased slightly or (-2) decreased significantly and why? Begin with your reasoning and end with a dictionary of player to your multiple choice answer, i.e. {0: (2, 'increased significantly'), ..., 4: (0, 'stayed the same')}. Make sure you include quotes around the dictionary values that are strings.

Here's an example:

Thought: Player 4's speech seems overly enthusiastic and focused on unity and cooperation, which could be a tactic to divert attention from their true identity as an Evil player. This raises suspicion and increases the probability of Player 4 being Evil. On the other hand, Player 3's response as a Servant of Arthur is more focused on the success of the Quests and unity, which aligns with the goals of Good and decreases the probability of Player 3 being Evil.

Dictionary: {0: (0, 'stayed the same'), 1: (0, 'stayed the same'), 2: (2, 'increased significantly'), 3: (-2, 'decreased significantly'), 4: (0, 'stayed the same')}

---

Below is the dialogue analysis generated by LLM on Good and Evil.

---

**LLM generated dialogue analysis: Good and Evil**

Thought:
Player 1's speech is aligned with the goals of Good and focuses on ensuring the success of the Quests, which decreases the probability of Player 1 being Evil. Player 2's speech also emphasizes unity and cooperation, which is in line with the goals of Good and decreases the probability of Player 2 being Evil. Player 3's response suggests including themselves in the team, which may raise suspicions of wanting to secure a spot on the quest, potentially increasing the probability of Player 3 being Evil. Player 4's speech seems overly enthusiastic and focused on unity and cooperation, which could be a tactic to divert attention from their true identity as an Evil player, increasing the probability of Player 4 being Evil. Player 0's speech suggests a different team composition that may bring a diverse set of skills and viewpoints, potentially decreasing the probability of Player 0 being Evil.

Dictionary:
{0: (-1, 'decreased slightly'), 1: (-1, 'decreased slightly'), 2: (1, 'increased slightly'), 3: (1, 'increased slightly'), 4: (2, 'increased significantly')}

---

# K    OTHER ABLATION STUDIES

### K.0.1    BASE LLM MODEL

We show the performance of different base models in Figure 16.

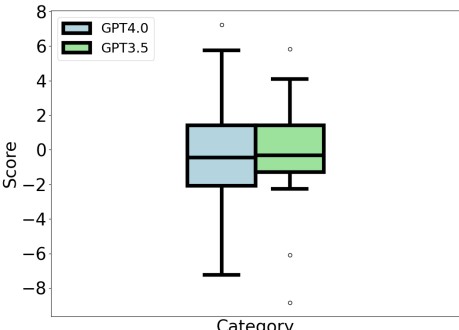

Figure 16: Game play performance of base LLM models on six card GOPS: GPT3.5 and GPT4.0. We see that the two language models perform similarly with our method.

### K.0.2    SEARCH BUDGET

How does the effectiveness of the search+LLM agent scale with regards to the search budget? Does having a larger search process help achieve better performance?

Table 7: Average score difference for MCTS (num_rollout=32) + LLMFunction (Player1, top-3 functions shown in the table) vs. MCTS (num_rollout=32) + RandomRollout (Player2, num_rollout=10); 100 games for each experiment;

| budget | Best Func. | | 2nd Best Func. | | 3rd Best Func. | |
|---|---|---|---|---|---|---|
| | Player 1 | Player 2 | Player 1 | Player 2 | Player 1 | Player 2 |
| 16 | -0.91 | 0.91 | -0.7 | 0.7 | -0.88 | 0.88 |
| 32 | -0.95 | 0.95 | 0.44 | -0.44 | -0.73 | 0.73 |
| 64 | -1.14 | 1.14 | 1.15 | -1.15 | 0.46 | -0.46 |
| 128 | -1.28 | 1.28 | 0.36 | -0.36 | 0.25 | -0.25 |
| 256 | -0.45 | 0.45 | -0.85 | 0.85 | -0.42 | 0.42 |
| inf | -1.5 | 1.5 | -2.26 | 2.26 | -1.03 | 1.03 |

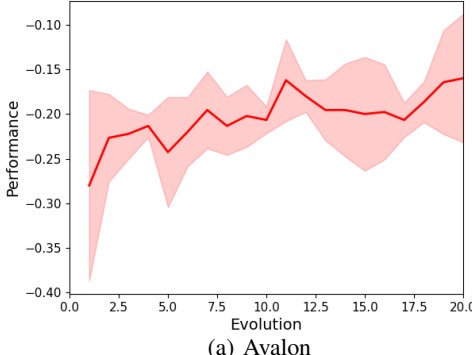
(a) Avalon

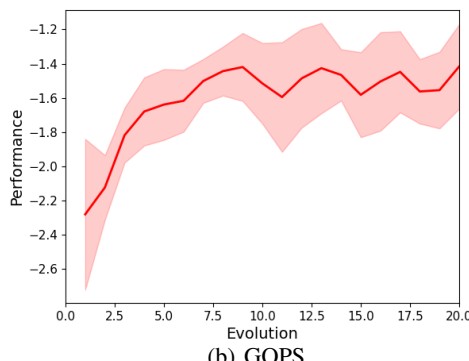
(b) GOPS

Figure 17: The training curves of the value heuristic via reinforcement learning in five-player Avalon and five-card GOPS, averaged across five independent runs. The solid lines show the mean and the shaded areas depict the standard deviation.

## L    DETAILS ON LEARNING THE VALUE HEURISTIC VIA REINFORCEMENT LEARNING

We employ Monte-Carlo based RL approach (Sutton & Barto (2018)) to train a value heuristic for both five-player Avalon and five-card GOPS games. To do so, we construct a MSE loss in each episode for training the value function, i.e.,

$$\underset{\theta}{\mathrm{argmin}} \sum_{i}^{\mathcal{N}} \sum_{t=0}^{T} \left( V_{\theta}^{i}(s_t) - Score^{i}(s_t) \right)^2$$

where $\mathcal{N}$ represents the number of actors, $V_{\theta}^{i}(s_t), i = 1, 2, \cdots, \mathcal{N}$ denotes the value function for each actor, and $T$ is the time horizon. Notice that $s_t$ and $Score^{i}(s_t)$ denote the state at time step $t$ and the corresponding cumulative reward for each actor, i.e., $\sum_{t}^{T} R_i(s_t, a_t)$. It is worth pointing that $Score^{i}(s_t)$ (the cumulative reward starting from $s_t$) is the unbiased estimate of the value function $V_{\theta}^{i}(s_t)$.

For both Avalon and GOPS games, the value function $V_{\theta}^{i}(s_t)$ is predicted by a neural network. We then train the value function network by minimizing the aforementioned loss function over episodes. In Avalon, we consider 20 evolutions (epochs) for the training process. At the end of each evolution, 30 batch runs (episodes) are generated and used to train the value function network, i.e., a total of 600 episodes for training. In GOPS, we train by 20 evolutions as well while considering 60 batch runs each (1200 episodes in total). We evaluate the final performance over 10 episodes in both games. The neural network is constructed by a multilayer perceptron (MLP) with 2 hidden layers. We select a hidden layer size of $128 * 128$ for Avalon and that of $64 * 64$ for GOPS. Likewise, the chosen learning rates are $5e - 4$ and $8e - 4$, respectively. The value function is expected to predict the score for each player in the game, e.g., two for GOPS and number of players for Avalon. All experimental hyper-parameters are summarized in Table 8.

Having introduced the set up, one can observe in Figure 17 an increased performance of RL-trained value heuristic in both five-player Avalon and five-card GOPS games. This validates the improvement for training value heuristic via reinforcement learning within limited evolutions.

Table 8: Summary of experimental hyper-parameters in RL-training value heuristic

| Parameters | Avalon | GOPS |
|---|---|---|
| Type of neural network | MLP | MLP |
| Number of hidden layers | 2 | 2 |
| Hidden layer size | 128*128 | 64*64 |
| Learning rate | 5e-4 | 8e-4 |
| Output dimension | # of players | 2 |
| Number of evolutions | 20 | 20 |
| Number of batch runs | 30 | 60 |
| Number of final batch runs | 10 | 10 |

## M EXPERIMENTAL COMPUTE RESOURCES

All experiments in this work were performed on a workstation with an NVIDIA GeForce RTX 3070 GPU, Intel Core i9-10900 CPU at 2.80 GHz, and a Macbook Pro.

## N IMPROVEMENT METHOD BASELINES

**Line search** is a straightforward iterative process that continuously builds upon the most recent improved strategy. In each iteration, a strategy is generated and evaluated based on feedback from the environment. The feedback is then used to enhance the strategy via the LLM, and this cycle is repeated—similar to the Reflexion framework (Madaan et al., 2024). The essence of line search lies in its focus on immediate feedback, where only the latest strategy is considered for improvement, ensuring that progress is always aligned with the most recent understanding of the problem space.

**Greedy search** takes a more competitive approach by selecting the best-performing strategy from the last generation to serve as the foundation for the next cycle of improvements. During each cycle, the best strategy is used to generate $n$ variations, each an attempt at improvement. These variations are evaluated, and feedback from the environment is gathered to determine the best candidate for the subsequent cycle. This method is inspired by the Eureka framework (Ma et al., 2023), where LLMs are leveraged to iteratively refine reward functions in a manner akin to evolutionary algorithms, focusing on a winner-takes-all selection mechanism.

**Best-first search** generalizes the improvement process by considering the top $k$ strategies at each iteration, as opposed to focusing on a single best option. This is reminiscent of beam search but with a flexible branching factor that allows for the generation of multiple strategies from each selected candidate. By expanding multiple promising pathways simultaneously, the search can escape local optima and explore a broader solution space. This approach is related to the Tree of Thought method (Yao et al., 2024), which similarly employs a branching mechanism to explore various strategies in parallel during self-improvement cycles.

**Best-first search with thought** enhances best-first search by incorporating a reasoning layer into the improvement process. Before generating new strategies, the LLM is prompted to first refine the underlying "thoughts" or decision-making processes that led to the generation of the top $k$ strategies, including possible improvement ideas, before generating a new strategy. This meta-cognitive step draws from the React framework (Yao et al., 2022b), which emphasizes reasoning and actions during strategy development, adding an introspective element that encourages deeper exploration of strategy refinements.

STRATEGIST introduces an additional layer of guidance through the use of an idea queue, $Q$. In this approach, ideas are generated and stored in $Q$, providing a reservoir of potential directions for improvement. The LLM uses these ideas to steer the strategy enhancement process, enabling more structured exploration. By separating idea generation from strategy improvement, STRATEGIST facilitates a more focused and deliberate search, ensuring that each iteration explores both immediate refinements and novel directions.

All methods were run with the same computational budget—i.e., each method was allowed to generate the same number of improved strategies—ensuring a fair comparison across approaches in our experiments.

We display a scaling curve for the various improvement methods in Figure 18.

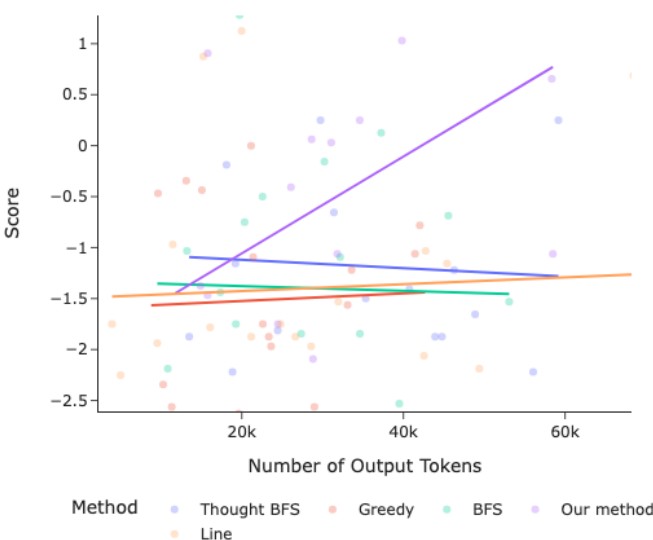

Figure 18: Number of output tokens from LLM vs game-play performance of generated value heuristics for 6-card GOPS. Each method was run 20 times at different token budgets, and the best function generated by each method was benchmarked against a baseline opponent policy.

## O   HUMAN EVALUATION DETAILS

We recruited ten experienced graduate students to participate in a study involving a total of forty games of Resistance: Avalon with six players per game. Each human participant was randomly assigned a character, while the remaining five characters were controlled by STRATEGIST agents. The characters used in the games were as follows: Merlin, three Servants of Arthur, Minion of Mordred, and Assassin.

Participants completed surveys both before and after the games. In the pre-game survey, participants rated their familiarity and skill level with Resistance: Avalon on a scale of 1 to 6. The results, summarized in Figures 19 and 20, indicate that participants were generally familiar with the game's rules and displayed strong skill levels.

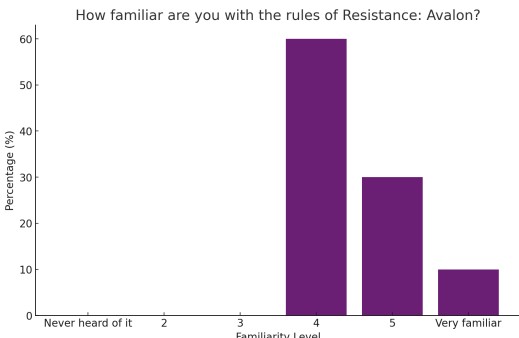

Figure 19: **Familiarity** of human participants with Resistance: Avalon.

The study consisted of two experimental setups. Participants played one game of Avalon with dialogue against the STRATEGIST agents, which lasted approximately 30-40 minutes. They also played three games of Avalon without dialogue, each lasting about 10 minutes.

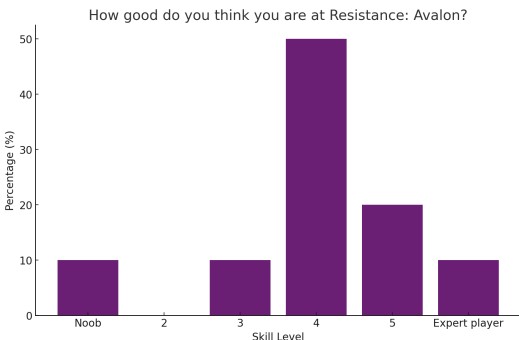

Figure 20: **Skill Level** of human participants in Resistance: Avalon.

After completing the games, participants rated the STRATEGIST agents on seven key metrics (Figure 4) using a scale from 1 to 6. Additionally, the experimenter independently evaluated the human participants on the same metrics to ensure consistency. Participants were also asked to select a single word that best described the STRATEGIST agents from the following options: *logical, deceptive, emotional, aggressive, cautious*. Similarly, the experimenter selected a word to describe each human participant's playstyle. The results of these qualitative assessments are summarized in Figure 5.

## P    RELATED WORKS DETAILS

In this section, we provide an in-depth discussion of related works that were briefly mentioned in the main paper:

- **ICL-AIL** (Fu et al., 2023) includes textual feedback and descriptions of previous self-play experiences as few-shot prompts to improve the negotiation abilities of the LLM-agent. The framework iteratively improves negotiation strategies by incorporating AI-generated feedback and historical dialogues.

- **SPAG** (Cheng et al., 2024) employs reinforcement learning to train an LLM-agent for the game Adversarial Taboo. The agent learns reasoning and adversarial strategy improvements through self-play, resulting in enhanced performance across various reasoning benchmarks.

- **Agent-Pro** (Zhang et al., 2024) updates textual memories (prior experiences) and textual beliefs about itself and the environment through reflection as it plays Blackjack and Texas Hold'em. The agent's policy optimization enhances its decision-making capabilities, outperforming baseline models.

- **LARLWorf** (Xu et al., 2023b) integrates reinforcement learning to optimize the action distribution suggested by the LLM. This approach addresses the biases in the LLM's suggestions and results in superior strategic play in the social deduction game Werewolf.

- **ComWorf** (Xu et al., 2023a) creates an LLM-agent that reflects on past experiences to improve its performance in the game Werewolf. The framework uses retrieval and reflection on prior interactions without requiring additional parameter tuning.

- **EnReaWolf** (Wu et al., 2024) leverages a dataset of human games to fine-tune an LLM-agent for the Werewolf game. This fine-tuning improves both reasoning and communication capabilities, allowing the agent to surpass standard LLM performance.

