# OpenReview forum: "Strategist: Self-improvement of LLM Decision Making via Bi-Level Tree Search"
_ICLR.cc/2025/Conference — ICLR 2025 Poster_

### Official Review · Reviewer_DMCE · 2024-10-20

**Soundness:** 2
**Presentation:** 2
**Contribution:** 3
**Rating:** 6
**Confidence:** 2

**Summary:**

This paper proposes a new complicated method named Strategist for using LLMs to generate strategies and ideas for multi-agent games.

In summary, there are two levels.

- Low-level self-play tree-search provides value for learning and updating skills.

- High-level skill learning module improves skills, both for text-based strategies (in text modality) and for value heuristic functions (in code modality).

**Strengths:**

- The idea to learn skills (strategies) not only for (1) text strategy guide, but also for (2) a value heuristic is inspiring. Utilizing LLMs for generating and improving value functions in code modality seems to be an innovative practice that utilizes their ability to reason and to write code. Moreover, the value functions generated could be used to search for good actions for non-speech phases of the game. To use these generated functions rather than to use LLMs could be cost-efficient and time-efficient.
- Improvement is shown. Moreover, performance scales with improvement of inference budget.
- Extensive experiments and details are provided to show the improvement of value functions and text strategies.

**Weaknesses:**

- Some results measured have no or confusing error analysis. For example, in Table 1, the median scores and IQR of quantities measured are reported, but the error range of the median scores is not reported. I recommend the authors also to report $median\pm error(median)$, in which the error is introduced because number of experiments is limited. Moreover, I wonder if enough duplicated experiments are carried out to prove that Strategist is better than BFS with Thought w.r.t. Avalon Value Heuristic. Also the presentation in Figure 4 is a bit chaotic.

- From section 2 it seems that the value functions are mainly for non-dialogue actions, and the text-based strategies are mainly for dialogue actions, and the action planner based on MCTS tree search described in Appendix F seems to be searching on the values generated by heuristic value functions. According to Appendix G it seems that the authors do not conduct similar search process on dialogue actions, and speeches are mainly provided through a rather traditional prompt-based approach. Of-course this does make sense given that speech is more related to text and is harder to assign scores, but these different approaches for non-dialogue and dialogue actions give a sense of fragmentation.

- There are typos. For example, in table 2 for GOPS the averaged 'winrate' of AlphaGo and DeepRole is negative (I assume it is a typo here: do the authors mean Heuristic Value or other scores?); in table 4 there are only results for winrate but no assassination rate is reported (though mentioned in table caption).

**Questions:**

Please refer to Weaknesses. Especially,
- Could you report the standard error of the median in addition to the IQR for all results in Table 1? Could you improve the clarity of Figure 4, perhaps by separating the data into multiple plots or using a different visualization method.
- Could you discuss potential limitations or biases introduced by using textual strategies only for dialogue actions? Have you considered any alternative methods to unify the approach of both types of actions, and if so, why they were not pursued?
- Could you clarify what metric is being reported in the 'Winrate' column for GOPS in Table 2, as negative values are unexpected for winrates? If this is an error, please provide the correct values.

---

> ### Author Response · Authors · 2024-11-25
>
> > Some results measured have no or confusing error analysis. For example, in Table 1, the median scores and IQR of quantities measured are reported, but the error range of the median scores is not reported. I recommend the authors also to report median error, in which the error is introduced because number of experiments is limited. Moreover, I wonder if enough duplicated experiments are carried out to prove that Strategist is better than BFS with Thought w.r.t. Avalon Value Heuristic. Also the presentation in Figure 4 is a bit chaotic.
>
>
> We thank the reviewer for the valuable suggestions. Based on your feedback, we have updated our error analysis and improved the presentation of results to provide greater clarity and robustness.
>
> 1. **Revised Error Analysis:**
>    Following your suggestion, we now report **standard error (SE)** instead of the interquartile range (IQR) in the tables. SE better reflects the precision of our estimates, particularly given the large number of experiments conducted. Multi-agent, hidden-information decision-making settings like Avalon naturally exhibit high variance due to the stochastic nature of strategies and interactions between players. Reporting IQR alone did not capture the full scope of our experimental efforts to mitigate noise, and SE provides a more accurate depiction of the results.
>
> 2. **Statistical Significance Tests:**
>    To ensure the robustness of our findings, we also conducted statistical significance tests for our key results:
>    - **Table 1:** A one-way ANOVA test shows a statistically significant difference between the means of 'Our method' and other methods, with an F-statistic of 29.49 and a p-value < 0.001.
>    - **Table 2:** A t-test comparing our 'LLM' method against 'RL' yields a t-statistic of 4.35 and a p-value < 0.001, indicating significant performance differences.
>    - **Table 3:** A one-way ANOVA test confirms a significant difference between 'Our method' and other methods (F-statistic of 4.08, p-value ≈ 0.0299).
>
> These results validate that STRATEGIST consistently outperforms baseline methods, including BFS with Thought, even in challenging and noisy environments like Avalon.
>
> 3. **Improved Figure 4 Presentation:**
>    We have reworked the presentation and description of Figure 4 to enhance clarity and reduce any perceived chaos. The updated figure now includes a better structured explanation in the text. These changes aim to better communicate the insights derived from the scaling experiments and interaction effects between high-level search and low-level refinement.
>
> By incorporating these revisions, we provide a more accurate and transparent depiction of STRATEGIST’s performance, emphasizing its robustness to noise and its consistent ability to outperform competing methods. Thank you for your thoughtful feedback, which has greatly improved the quality and presentation of our results.

---

> > ### Author Response · Authors · 2024-11-25
> >
> > > From section 2 it seems that the value functions are mainly for non-dialogue actions, and the text-based strategies are mainly for dialogue actions, and the action planner based on MCTS tree search described in Appendix F seems to be searching on the values generated by heuristic value functions. According to Appendix G it seems that the authors do not conduct similar search process on dialogue actions, and speeches are mainly provided through a rather traditional prompt-based approach. Of-course this does make sense given that speech is more related to text and is harder to assign scores, but these different approaches for non-dialogue and dialogue actions give a sense of fragmentation.
> >
> >
> > Thank you for pointing this out. While the low-level refinement processes for dialogue and non-dialogue actions necessarily differ, as the reviewer noted, the high-level self-play improvement framework remains unified. Both dialogue and non-dialogue strategies are iteratively refined using the **strategy library** and **idea queue**, ensuring a cohesive improvement process. This alignment is central to our **bi-level approach**, a major novelty of our work, which abstracts away low-level execution details to focus on optimizing high-level strategies.
> >
> > The abstraction of low-level execution details is a significant innovation because it allows STRATEGIST to generalize across different types of actions—dialogue and non-dialogue—without being constrained by their inherent differences. By separating the high-level strategy refinement from the specific mechanics of action execution, STRATEGIST enables a unified framework for learning and improvement that can operate effectively in diverse environments. For example:
> > - **Dialogue Actions:** These are more qualitative and rely on text generation. Scoring them is inherently harder due to the subjective nature of dialogue quality and the lack of clear, numerical reward signals. Here, STRATEGIST abstracts the generation process into high-level dialogue strategies and refines these strategies independently of the specifics of text-based action generation.
> > - **Non-Dialogue Actions:** These are more quantitative and typically involve numerical heuristics, such as value functions for decision-making. STRATEGIST refines these strategies without being tied to the low-level details of how values are computed or actions are executed.
> >
> > This abstraction is key to bridging the gap between qualitatively different types of actions. Instead of treating dialogue and non-dialogue actions as fragmented, STRATEGIST focuses on improving their shared high-level strategies through self-play, leveraging the same iterative processes for strategy evolution. This unifying perspective not only simplifies the overall framework but also enhances its adaptability to new settings, where the low-level execution mechanisms might differ even more drastically.
> >
> > > There are typos. For example, in table 2 for GOPS the averaged 'winrate' of AlphaGo and DeepRole is negative (I assume it is a typo here: do the authors mean Heuristic Value or other scores?); in table 4 there are only results for winrate but no assassination rate is reported (though mentioned in table caption).
> >
> > Thank you for catching these. We fixed the typos in the revision.

---

> > > ### Comment · Reviewer_DMCE · 2024-11-26
> > >
> > > I would like to thank the authors for their rebuttals. I would like to change my score from a 5 to a 6.

---

> > > > ### Author Response · Authors · 2024-11-27
> > > > **Thank you note**
> > > >
> > > > Thank you for your thoughtful engagement with our work and for updating your score! We deeply appreciate your constructive feedback, which has significantly improved our paper’s clarity and robustness. Your insights have been invaluable in refining our presentation and experimental analyses.

---

### Official Review · Reviewer_MWLe · 2024-10-31

**Soundness:** 3
**Presentation:** 3
**Contribution:** 2
**Rating:** 6
**Confidence:** 4

**Summary:**

The paper presents STRATEGIST, a framework that combines LLMs with MCTS for learning optimal strategies in competitive multi-agent games. The approach uses LLMs to generate and iteratively improve high-level strategies, which are then refined into executable policies using MCTS. The framework is evaluated on two games: GOPS and Avalon, showing superior performance compared to traditional RL methods and other LLM-based approaches.

**Strengths:**

- Integration of LLMs with MCTS in a bi-level framework that leverages their complementary strengths.
- Clear empirical validation through rigorous experiments.
- The paper is well written and easy to follow

**Weaknesses:**

See the questions below.

**Questions:**

- What happens when the LLM generates syntactically correct but semantically inconsistent or illogical strategies? How does the framework detect and handle such cases?
- How does the method prevent the idea queue from becoming dominated by similar or redundant improvement suggestions?
- What happens when there are multiple valid interpretations of a strategy? How is the most appropriate one selected?
- Are the proposed method verified to learn general strategies instead of overfitting to specific feedback examples?
- Could the method combine successful elements from different strategies into new hybrid approaches?

---

> ### Author Response · Authors · 2024-11-25
>
> > What happens when the LLM generates syntactically correct but semantically inconsistent or illogical strategies? How does the framework detect and handle such cases?
>
> Thank you for raising this important question. Detecting and addressing semantically inconsistent or illogical strategies is a critical aspect of STRATEGIST's robustness.
>
> The framework incorporates several mechanisms to handle such cases effectively:
>
> 1. **Filtering Through Self-Play:**
>    Strategies that are semantically inconsistent or illogical are naturally penalized during the self-play process. In STRATEGIST, the self-play simulations evaluate strategies based on their performance in actual gameplay scenarios. Illogical strategies typically result in poor outcomes (e.g., low win rates or ineffective dialogue interactions), and these poor-performing strategies are pruned out during the iterative improvement process. This ensures that only consistent and effective strategies remain by the end of the self-play refinement.
>
> 2. **Feedback-Driven Improvement:**
>    STRATEGIST uses detailed feedback from self-play to guide the revision of strategies. The feedback signals include win rates, reward values, and discrepancies between predicted and actual outcomes during gameplay. For example, Monte Carlo Tree Search (MCTS) not only executes actions based on a strategy but also evaluates the outcomes to detect inconsistencies. When inconsistencies are identified (e.g., a strategy leading to nonsensical outcomes or contradicting earlier steps), this feedback informs the improvement process, prompting the LLM to revise the problematic parts of the strategy.
>
> 3. **Idea Queue and Modular Refinement:**
>    The framework's modular refinement process ensures that strategy improvements are incremental and interpretable. Each iteration isolates specific components of a strategy for enhancement, which reduces the likelihood of introducing new inconsistencies. The idea queue maintains a library of successful refinements, helping guide the search process toward logical and effective strategies.
>
> 4. **Population-Based Selection:**
>    STRATEGIST conducts population-based self-play, where multiple strategies are evaluated in a round-robin format. This evolutionary approach favors strategies that consistently perform well across diverse opponents and scenarios. Semantically inconsistent strategies tend to underperform in such competitive settings and are systematically eliminated.
>
> By combining these mechanisms, STRATEGIST not only filters out illogical strategies but also actively improves strategy quality through iterative refinement. This ensures that the final strategies are both semantically coherent and practically effective in achieving desired outcomes. If you have further suggestions on enhancing these processes, we would be delighted to consider them.
>
>
> > How does the method prevent the idea queue from becoming dominated by similar or redundant improvement suggestions?
>
>
> Thank you for this insightful question. Preventing the idea queue from becoming dominated by similar or redundant improvement suggestions is essential to ensure diverse and effective strategy refinement. STRATEGIST incorporates several mechanisms to address this challenge:
>
> 1. **Diversity-Oriented Prompting:**
>    When generating ideas, the LLM is specifically prompted to produce a diverse set of improvement suggestions. These prompts encourage variation by asking the LLM to explore different aspects of strategy refinement, such as alternative approaches to value heuristics, new dialogue patterns, or novel ways to address partial observability. By explicitly steering the LLM toward diversity, this step reduces the likelihood of generating redundant or overly similar ideas.
>
> 2. **Upper Confidence Bound (UCB) Selection:**
>    STRATEGIST employs UCB-based sampling to select ideas from the queue, which balances exploration and exploitation. This ensures that ideas with high potential impact (based on prior performance) are prioritized while still exploring less-tested ideas. UCB's mathematical formulation inherently promotes diversity by giving underexplored ideas opportunities for evaluation, preventing the process from converging too quickly on a narrow subset of ideas.
>
> 3. **Feedback-Driven Idea Evaluation:**
>    Each idea's effectiveness is assessed based on the improvements it brings to strategies during self-play simulations. If an idea repeatedly shows limited effectiveness or leads to redundant refinements, its priority in the queue decreases. This dynamic evaluation mechanism helps prevent the queue from being clogged with redundant or low-value suggestions.
>
>
> Together, these mechanisms promote diversity and innovation in the idea queue, ensuring a wide-ranging exploration of the strategy space while systematically improving performance.

---

> > ### Author Response · Authors · 2024-11-25
> >
> > > What happens when there are multiple valid interpretations of a strategy? How is the most appropriate one selected?
> >
> >
> > Great question! STRATEGIST is designed to effectively handle situations where multiple valid interpretations of a strategy arise by leveraging its low-level executor to select and refine the most appropriate one during execution. For action-based strategies, the Monte Carlo Tree Search (MCTS)-based low-level executor plays a crucial role. MCTS explores different action paths using look-ahead search, simulating potential outcomes for each interpretation by interacting with the environment. Through this process, it evaluates which interpretation aligns best with the high-level strategy and yields the highest expected rewards. This iterative feedback loop of exploration and refinement ensures that the optimal interpretation is dynamically selected and executed. It’s an exciting demonstration of how STRATEGIST adapts to complex and uncertain scenarios.
> >
> > For dialogue-based strategies, the low-level executor can deploy LLM inference scaling techniques such as **Best-of-N sampling** to select the best interpretation. This involves generating multiple candidate responses and choosing the one that best aligns with the high-level strategy and fits the dialogue context. This method ensures that the most coherent, impactful, and strategically relevant response is chosen, even within the intricate dynamics of multi-agent discussions. In fact, we’ve included an additional experiment in the revised paper where we apply the Best-of-N technique to dialogue generation, helping to select a more refined and effective interpretation!
> >
> > By combining MCTS for action-taking and inference scaling techniques for dialogue, STRATEGIST excels at resolving ambiguity and selecting the most appropriate interpretation of a strategy. Thank you for raising this intriguing topic—it’s one of the areas where STRATEGIST's bilevel approach truly shines, allowing high-level strategy optimization to remain focused on broader goals without being hindered by low-level micro-optimizations.
> >
> >
> > > Are the proposed method verified to learn general strategies instead of overfitting to specific feedback examples?
> >
> >
> > Excellent question! Ensuring that STRATEGIST learns general strategies rather than overfitting to specific feedback examples is a core aspect of its design. This is precisely where **population-based self-play** plays a crucial role. By playing against a diverse set of opponents, STRATEGIST inherently promotes the development of robust, adaptable strategies that generalize across a variety of scenarios.
> >
> > Here’s how it works: during the self-play process, strategies are tested in a round-robin format against multiple opponents, each employing different strategies. Strategies that overfit to specific examples or narrowly exploit certain patterns tend to underperform against this diverse population. Poor-performing strategies are naturally discarded in favor of those that achieve consistently high win rates across a wide range of adversaries. This evolutionary approach allows STRATEGIST to refine and improve its strategies iteratively, ensuring they are not just tailored to isolated cases but are effective in broader, more generalized contexts.
> >
> > Additionally, the **bi-level framework** further supports generalization by separating high-level strategy abstraction from low-level execution. High-level strategies are designed to encode general principles or heuristics, which are less prone to overfitting compared to low-level, situation-specific rules. The low-level executor, through mechanisms like MCTS, provides adaptability during execution, further preventing reliance on overfitted patterns.
> >
> > In summary, the population-based self-play combined with STRATEGIST’s bi-level design ensures the learning of strategies that are not only robust but also broadly applicable. Thank you for raising this critical point—it highlights one of STRATEGIST’s key strengths in navigating complex environments!

---

> > > ### Author Response · Authors · 2024-11-25
> > >
> > > > Could the method combine successful elements from different strategies into new hybrid approaches?
> > >
> > >
> > > Another excellent question! STRATEGIST is specifically designed to combine successful elements from different strategies into hybrid approaches, facilitated by the **idea queue**. The idea queue captures and tracks successful refinements, enabling their transfer across strategies. For instance, if an idea improves strategy A, it is also tested on strategies B and C. Using our **UCB-based sampling policy**, frequently successful ideas are prioritized, while less effective ones are deprioritized, ensuring efficient exploration of high-value contributions.
> > >
> > > This process naturally fosters hybrid strategies by merging successful elements across different contexts. For example, an effective dialogue guide question asking players to consider **the voting patterns of opponents across multiple rounds** might be combined with a question asking for **an analysis of mission success rates and the roles of involved players** in another guide. Similarly, a function to consider the **highest-value cards in a player’s hand in GOPS** might be added to a value heuristic that also accounts for **the remaining prize cards in the deck and their potential impact on the game’s outcome**. STRATEGIST’s bi-level framework ensures that these hybridizations remain structured and effective, as the high-level strategies provide interpretability and the low-level executor adapts to specific dynamics.
> > >
> > > In short, the idea queue drives collaboration between strategies, enabling STRATEGIST to continually innovate and refine hybrid approaches. Thank you for highlighting this exciting capability!
> > >
> > > We greatly appreciate the questions from reviewer MWLe, and it has been a pleasure answering them.

---

> > > > ### Comment · Reviewer_MWLe · 2024-11-26
> > > >
> > > > Thank you for the response. Most of my concerns have been addressed. For now, I will keep my positive score, and I will also pay attention to the authors' discussion with other reviewers.

---

> > > > > ### Author Response · Authors · 2024-11-27
> > > > >
> > > > > Thank you very much for your thoughtful review and constructive questions. We sincerely appreciate your engagement with our work and the opportunity to address your insightful points. Your feedback has helped us further refine and clarify our methodology, particularly regarding strategy inconsistency, idea queue diversity, and strategy interpretation. We are delighted to hear that your concerns have been largely addressed and value your continued consideration of our paper during discussions. Thank you for your time and effort in reviewing our submission!

---

### Official Review · Reviewer_u9ZM · 2024-11-02

**Soundness:** 3
**Presentation:** 2
**Contribution:** 3
**Rating:** 6
**Confidence:** 4

**Summary:**

The paper presents Strategist, a novel framework that leverages Large Language Models (LLMs) for strategic skill acquisition in multi-agent games through a bi-level tree search and self-improvement mechanism. The approach aims to enhance both high-level strategy formulation and low-level execution by utilizing self-play simulations and LLM-based reflections. The framework is evaluated on two games, Game of Pure Strategy (GOPS) and Resistance: Avalon, demonstrating improved performance over baseline methods.

**Strengths:**

1. Novel approach: the combination of LLMs with a bi-level tree search for strategy improvement is a novel method for enhancing performance in complex multi-agent games. Addressing both high-level strategic planning and low-level execution allows for improvements of agent capabilities.
2. Good empirical evaluation: The authors conduct various ablation studies and additional experiments comparing their method with established baselines like DeepRole and ReCon, providing a broader context for their contributions.

**Weaknesses:**

1. Reliability of population-based self-play simulation: the authors use round-robin games between top-ten strategies to evaluate the performance of high-level strategies. Since games like Avalon have high uncertainty and the variance of the simulation result is large, it would require many simulations (like hundreds) to get a reliable evaluation of these strategies. However, these simulations would take a long time for LLM inferences. In addition, LLM usually cannot take so many trajectories as input due to the limited context length. It would be better if the authors could provide results to justify the reliability of population-based self-play simulation results.
2. Modular improvement Assumptions: the assumption that improvement ideas are universally applicable across different strategies requires further justification. Improvement ideas may interact with specific strategies in unique ways, possibly leading to unintended consequences or limited effectiveness when applied broadly. While the authors argue that improvements are additive and generalizable, empirical evidence demonstrating this across a variety of strategies would strengthen the claim.
3. Statistical significance and result variability: the reported variances are very large and show significant overlap in many results like Table 1, 2, 3, etc., which implies that the proposed methods might not be delivering substantial enhancements in these test environments. For example, in Table 2, the result of Alpha-go is $-0.39\pm0.71$, and the result of Strategist is $0.33\pm1.21$. The variance is too large to determine whether the new method offers improvements over existing baselines.
4. Suggestions on some related works: the experiments on mainly conducted on the social deduction game named Avalon. A closely related game is Werewolf and there are some recent works like [1,2,3] on building LLM AI agents for these games and could be added to the related work section.

[1] Yuzhuang Xu, et al. "Exploring large language models for communication games: An empirical study on werewolf." arXiv preprint arXiv:2309.04658 (2023).

[2] Zelai Xu, et al. "Language agents with reinforcement learning for strategic play in the werewolf game." arXiv preprint arXiv:2310.18940 (2023).

[3] Bailis, Suma, Jane Friedhoff, and Feiyang Chen. "Werewolf arena: A case study in llm evaluation via social deduction." arXiv preprint arXiv:2407.13943 (2024).

**Questions:**

Please see Weaknesses.

---

> ### Author Response · Authors · 2024-11-25
>
> > Reliability of population-based self-play simulation: the authors use round-robin games between top-ten strategies to evaluate the performance of high-level strategies. Since games like Avalon have high uncertainty and the variance of the simulation result is large, it would require many simulations (like hundreds) to get a reliable evaluation of these strategies. However, these simulations would take a long time for LLM inferences. In addition, LLM usually cannot take so many trajectories as input due to the limited context length. It would be better if the authors could provide results to justify the reliability of population-based self-play simulation results.
>
> Thank you for this thoughtful and important point regarding the reliability of population-based self-play simulations. We recognize that games like Avalon inherently involve high uncertainty, and ensuring reliable evaluations requires many simulations. This is precisely where the bi-level structure of our STRATEGIST framework offers a key advantage.
>
> By abstracting low-level policy refinement to high-level strategies, our method significantly reduces the computational burden of repeated LLM inferences. The LLM operates at the level of strategic abstractions, which eliminates the need to process extensive game trajectories directly, effectively addressing the context length limitation you mentioned. Furthermore, the low-level executor, which employs Monte Carlo Tree Search (MCTS), can efficiently handle hundreds of simulated games in a short time. For instance, running 512 GOPS simulations typically takes under three minutes on standard hardware, ensuring robust statistical evaluation within a practical timeframe.
>
> Additionally, the evolutionary nature of our population-based self-play process ensures that strategies are rigorously tested against diverse opponents, leading to robust and generalizable strategies. This not only mitigates variance but also leverages the ability of STRATEGIST to iteratively improve strategies through simulation feedback, as detailed in Section 2.4 of the paper.
>
>
> We hope this explanation clarifies the reliability and efficiency of our approach and highlights the strategic benefits of the bi-level framework in managing the challenges of games like Avalon. We are happy to provide further details or additional analyses if needed.
>
>
> > Modular improvement Assumptions: the assumption that improvement ideas are universally applicable across different strategies requires further justification. Improvement ideas may interact with specific strategies in unique ways, possibly leading to unintended consequences or limited effectiveness when applied broadly. While the authors argue that improvements are additive and generalizable, empirical evidence demonstrating this across a variety of strategies would strengthen the claim.
>
> We appreciate the reviewer’s insightful feedback regarding the assumption that improvement ideas are universally applicable across different strategies. To address this, we have conducted an additional ablation study in the GOPS environment to empirically evaluate whether modular improvement assumptions lead to better results. Below, we present the results:
>
> | **Category**  | **Mean Point Difference** |
> |---------------|---------------------------|
> | Modular       | 0.4616                    |
> | Non-modular   | -0.0213                   |
>
> The results demonstrate that modular improvement is indeed critical for achieving self-play enhancement, as the modular approach yields a significantly positive mean point difference compared to the non-modular approach, which shows a slight negative impact. This supports our argument that modular improvements are not only additive but also generalizable across different strategies.
>
> We thank the reviewer for their suggestion, which has helped us strengthen our claim with additional empirical evidence.

---

> > ### Author Response · Authors · 2024-11-25
> >
> > > Statistical significance and result variability: the reported variances are very large and show significant overlap in many results like Table 1, 2, 3, etc., which implies that the proposed methods might not be delivering substantial enhancements in these test environments. The variance is too large to determine whether the new method offers improvements over existing baselines.
> >
> >
> > Thank you for highlighting the variability in our reported results and its implications for the robustness of our proposed method. We recognize that the use of interquartile range (IQR) in our tables may not have been the most suitable metric for conveying statistical significance. Following the suggestion of **Reviewer DMCE**, we have revised our error reporting methodology to use **standard error (SE)** instead. Additionally, we have organized our error analysis to provide a more detailed and structured presentation of the results.
> >
> > Reporting standard error (SE) provides a clearer understanding of the results by emphasizing the precision of sample means, particularly given the large number of evaluations in our study. This approach effectively reduces the impact of noise from individual trial variability and highlights the statistical reliability of our method’s performance. Moreover, game environments, especially those with multi-agent and dialogue-based interactions like Avalon, are inherently noisy due to the stochastic nature of strategies and dependencies on teammate and opponent behaviors. While such factors contribute to high IQR values, SE offers a more reliable metric for capturing the underlying trends and consistency of the results.
> >
> > ### Statistical Significance Tests:
> >
> > To further address concerns, we conducted statistical significance tests for our main experiments, summarized below:
> > - **Table 1:** A one-way ANOVA test shows a statistically significant difference between the means of 'Our method' and other methods, with an F-statistic of 29.49 and a p-value < 0.001.
> > - **Table 2:** A t-test comparing our 'LLM' method against 'RL' yields a t-statistic of 4.35 and a p-value < 0.001, indicating significant performance differences.
> > - **Table 3:** A one-way ANOVA test confirms a significant difference between 'Our method' and other methods (F-statistic of 4.08, p-value ≈ 0.0299).
> >
> > These results support the statistical robustness of our approach, demonstrating STRATEGIST’s reliability and consistency.
> >
> > We are confident that the revised analysis addresses the concerns regarding variability and significance. The updated results provide a more accurate depiction of STRATEGIST’s strengths, particularly in handling the challenging and noisy dynamics of multi-agent and dialogue-based environments.
> >
> >
> >
> > > Suggestions on some related works: the experiments on mainly conducted on the social deduction game named Avalon. A closely related game is Werewolf and there are some recent works like [1,2,3] on building LLM AI agents for these games and could be added to the related work section.
> >
> > We thank the author for these suggestions, and have added them to the related works section (Section 4).
> >
> > [1] Yuzhuang Xu, et al. "Exploring large language models for communication games: An empirical study on werewolf." arXiv preprint arXiv:2309.04658 (2023).
> >
> > [2] Zelai Xu, et al. "Language agents with reinforcement learning for strategic play in the werewolf game." arXiv preprint arXiv:2310.18940 (2023).
> >
> > [3] Bailis, Suma, Jane Friedhoff, and Feiyang Chen. "Werewolf arena: A case study in llm evaluation via social deduction." arXiv preprint arXiv:2407.13943 (2024).

---

> > > ### Author Response · Authors · 2024-11-29
> > > **Exciting new results!**
> > >
> > > We have conducted another experiment to verify the reliability of population based self-play in our setting, per the reviewer's suggestion
> > >
> > > In this experiment, we evaluated the performance of strategies generated during the high-level improvement process in two different ways:
> > >
> > > 1. **Self-Play Evaluation**: Each strategy was scored based on round-robin matches against a sample population of six strategies derived from the same evolutionary process.
> > >
> > > 2. **General Population Evaluation**: The same strategies were scored when played against a larger, more diverse population of 30 strategies that included historical strategies from different stages of the improvement process.
> > >
> > > The goal was to measure the correlation between scores obtained in these two settings. High correlation would indicate that performance in the self-play evaluation is a reliable proxy for general robustness.
> > >
> > > | Metric             | Mean Correlation | Standard Error (SE) | Number of Strategies in Self-Play Sample | Number of Strategies in General Population |
> > > |---------------------|------------------|---------------------|-------------------------------------------|--------------------------------------------|
> > > | Pearson Correlation | 0.707           | 0.058              | 6                                         | 30                                         |
> > > | Rank Correlation    | 0.713           | 0.061              | 6                                         | 30                                         |
> > >
> > > The results show strong Pearson and rank correlation, indicating that strategy rankings and relative performance estimated through population-based self-play align well with their performance against a broader population. This suggests that the self-play simulation reliably captures the comparative quality of strategies without requiring exhaustive simulations.
> > >
> > > Furthermore, our approach mitigates the variance in simulation outcomes by aggregating results across multiple games within the self-play evaluation. This balances computational efficiency with accuracy, especially important given the context-length limitations of LLMs during strategy refinement.
> > >
> > > These findings validate the reliability of our population-based self-play simulation framework in evaluating strategy robustness and performance in games like Avalon. Please let us know if you have any additional concerns and questions!

---

> ### Author Response · Authors · 2024-12-02
> **Friendly reminder**
>
> Dear Reviewer u9ZM,
>
> We hope this message finds you well. This is a gentle reminder that we are still awaiting your response regarding our submission.
>
> We completely understand how demanding your schedule may be and deeply value the time and effort you dedicate to this important process. Your insights have been incredibly valuable, and we would be grateful for any additional feedback or clarifications you might be able to provide.
>
> Thank you once again for your thoughtful review and support in strengthening our work. We look forward to hearing from you at your earliest convenience.
>
> Warm regards,
>
> The Authors

---

> > ### Comment · Reviewer_u9ZM · 2024-12-02
> >
> > Thank you for your detailed response. Most of my concerns have been addressed. I'm happy to improve my score to 6.

---

> > > ### Author Response · Authors · 2024-12-02
> > > **Thank you note**
> > >
> > > Thank you for your thoughtful and detailed review, as well as for revisiting your assessment based on our responses. Your insights, particularly on population-based self-play and statistical variability, greatly improved our analysis and strengthened the paper. We deeply appreciate the time and effort you invested in providing such constructive feedback!

---

### Official Review · Reviewer_GQJM · 2024-11-04

**Soundness:** 3
**Presentation:** 3
**Contribution:** 3
**Rating:** 6
**Confidence:** 3

**Summary:**

This paper introduces STRATEGIST, which is a framework to optimize the strategy induced by LLMs through population-based self-play simulations without the need for any training data. The effectiveness of STRATEGIST in learning optimal strategies is demonstrated against strong baselines on Game of Pure Strategy (GOPS) and the Resistance: Avalon.

**Strengths:**

- The paper is well written and well organised.
- The description of different components in STRATEGIST is clear.
- The experimental results are strong and comprehensive.

**Weaknesses:**

- The differences and advantages of the self-play mechanisms of STRATEGIST in improving the strategy against previous related work on self-play for LLMs is unclear.  It would be helpful to provide a clear comparison (e.g. a table) between the self-play mechanism in STRATEGIST and previous self-play methods for LLMs  (e.g. [1] [2] ...) .

[1] Yao Fu, Hao Peng, Tushar Khot, and Mirella Lapata. Improving language model negotiation with
self-play and in-context learning from ai feedback. arXiv preprint arXiv:2305.10142, 2023.


[2] Cheng, Pengyu, et al. "Self-playing Adversarial Language Game Enhances LLM Reasoning." arXiv preprint arXiv:2404.10642 (2024)

**Questions:**

- What makes the self-play method in STRATEGIST better than previous self-play methods for LLMs, in terms of improving the LLM-induced strategy? I would expect a clear comparison against previous self-play methods in LLMs both in terms of the methodology (as mentioned in the weakness) and experiment study (For instance, ablation studies comparing STRATEGIST to other LLM self-play methods).

---

> ### Author Response · Authors · 2024-11-25
>
> > The differences and advantages of the self-play mechanisms of STRATEGIST in improving the strategy against previous related work on self-play for LLMs is unclear. It would be helpful to provide a clear comparison (e.g. a table) between the self-play mechanism in STRATEGIST and previous self-play methods for LLMs (e.g. [1] [2] ...) .
>
> We thank the reviewer for the suggestion of a comparison table, which is an amazing addition to our paper. We have added the table to the paper as suggested (Table 5). This helps us highlight some of the core contributions of Strategist which may not be clear to the reader. We highlight some of the key differences and novelties of our paper in the next question.
>
> > What makes the self-play method in STRATEGIST better than previous self-play methods for LLMs, in terms of improving the LLM-induced strategy? I would expect a clear comparison against previous self-play methods in LLMs both in terms of the methodology (as mentioned in the weakness) and experiment study (For instance, ablation studies comparing STRATEGIST to other LLM self-play methods).
>
>
> Thank you for your thoughtful feedback. We appreciate the opportunity to clarify and expand on the distinctions between **STRATEGIST** and previous self-play methods for LLMs, as well as to address your request for comparative analysis.
>
> ---
>
> ### Key Differences Between STRATEGIST and Existing Self-Play Methods:
>
> 1. **Hierarchical Strategy Improvement:**
>    STRATEGIST introduces a bi-level approach where high-level strategies (e.g., value heuristics or dialogue guides) are iteratively improved. This contrasts with other self-play methods that typically focus on refining low-level actions directly. The abstraction of high-level strategies enables more efficient exploration and generalization of strategic principles, as demonstrated in the evolutionary process described in Section 2.2.
>
> 2. **Parameter-Free Framework:**
>    STRATEGIST eliminates the need for parameter tuning and human-annotated training data, making it a truly non-parametric framework. For example, unlike methods like Cicero that rely on large-scale human data for fine-tuning, STRATEGIST achieves superior performance solely through self-play and simulated feedback.
>
> 3. **Advanced Low-Level Refinement with Tree Search:**
>    While other methods often prompt LLMs directly for actions, STRATEGIST incorporates Monte Carlo Tree Search (MCTS) to refine decisions informed by high-level strategies. This enhances the precision of decision-making, as elaborated in Section 2.3.
>
> 4. **Partial Observability Handling:**
>    STRATEGIST explicitly models belief updates to address partial information settings, such as those in multi-agent games like Avalon. In contrast, many LLM-based agents, including ReAct and ReCon, do not explicitly handle such complexities, as detailed in Section 3.5.
>
> 5. **Advanced Skill-Learning Process:**
>    STRATEGIST uses an innovative approach involving a **strategy library** and **idea queue** to guide incremental improvements. The strategy library maintains a tree structure of evolving strategies, while the idea queue enables modular, targeted refinements by tracking the effectiveness of improvement ideas. This approach avoids confounding factors in strategy development and ensures that the most impactful ideas are systematically explored, as described in Section 2.3.3.
>
> ---
>
> ### Comparative Experiments and Ablation Studies:
>
> We have conducted comprehensive ablation studies comparing STRATEGIST to related methods, including ReCon and DeepRole, in partially observable, multi-agent settings. Tables 1, 2, and 4 in the paper provide detailed comparisons across various metrics such as win rate, dialogue generation accuracy, and strategy evolution efficiency. These results highlight STRATEGIST's superior ability to balance generalization (via high-level strategies) and precision (via tree search refinement).
>
> ---
>
> ### Additional Human Experiments:
>
> To ensure broad applicability, we also evaluated STRATEGIST against LLM-agents implemented in different experimental settings. These experiments included human-player interactions, where STRATEGIST achieved comparable results against humans as shown in Section 3.1. This directly addresses the need for comparative performance studies against other agents in real-world scenarios.
>
> ---
>
> We hope this addresses your concerns regarding the methodological and experimental comparisons.
>
>
> [1] Yao Fu, Hao Peng, Tushar Khot, and Mirella Lapata. Improving language model negotiation with self-play and in-context learning from ai feedback. arXiv preprint arXiv:2305.10142, 2023.
>
> [2] Cheng, Pengyu, et al. "Self-playing Adversarial Language Game Enhances LLM Reasoning." arXiv preprint arXiv:2404.10642 (2024)

---

> > ### Comment · Reviewer_GQJM · 2024-11-26
> >
> > Thanks very much for the rebuttal, which alleviates my concerns. I will maintain my initial rating.

---

> > > ### Author Response · Authors · 2024-11-27
> > >
> > > Thank you for your thoughtful review and valuable feedback on our submission. We deeply appreciate your suggestions for clarifying and emphasizing the distinctions of STRATEGIST, which led to the addition of the comparison table and further refinement of our experimental analysis. We’re glad to hear that our responses have alleviated your concerns and are grateful for your time and consideration. Thank you again for your engagement with our work!

---

### Official Review · Reviewer_XeVt · 2024-11-05

**Soundness:** 2
**Presentation:** 3
**Contribution:** 2
**Rating:** 3
**Confidence:** 3

**Summary:**

The paper presents STRATEGIST which combines LLM-generated high-level strategies with Monte Carlo Tree Search (MCTS) for refined action execution. STRATEGIST uses population-based self-play to optimize strategies. The paper presents experimental results in two game environments: GOPS and The Resistance: Avalon. The results indicate that STRATEGIST outperforms traditional RL agents and pure LLM-based methods.

**Strengths:**

1. The idea of using LLM to generate high-level strategy and use MCTS to get low-level strategy is interesting.

2. The proposed method is evaluated in two game environments, one simple and one complex, which is good.

**Weaknesses:**

1. Lack of Competitive Baselines: The paper compares with methods that do not use LLM at all (like DeepRole) and methods that only uses LLM but no RL. However, the comparisons exclude more recent RL-LLM hybrid methods that would provide a fairer benchmark for STRATEGIST's effectiveness. In fact, in the past two years, there are a number of papers that combines RL with LLM to play complex strategic multi-agent games that involve natural-language based communication. For example, the paper mentioned the Cicero paper by FAIR, but there is no comparison with the method used in the Cicero paper. There are a lot more papers following up the Cicero paper, and can potentially be used for games like Avalon. Here are a few example papers: 1). Exploring large language models for communication games: An empirical study on werewolf. 2). Language agents with reinforcement learning for strategic play in the werewolf game; 3). Enhance reasoning for large language models in the game werewolf; 4). Agent-pro: Learning to evolve via policy-level reflection and optimization.

2. Limited Analysis on Strategic Randomization: Strategic randomization is essential in hidden-role games like Avalon. However, the experiment section does not show how well STRATEGIST presents deceptive behavior in a properly randomized fashion.

3. Absence of Human Evaluation: Despite STRATEGIST's applicability to discussion-based games, there is no reported evaluation of the model's interactions with human players, which would provide insights into its performance in realistic scenarios.

4. Comparison of Bi-Level Structure: Cicero also uses a bi-level structure, although in a different way as this paper proposes: In Cicero, it uses iterative learning methods to choose the intent at the high-level, and uses LLM to generate low-level actions. Cicero's bi-level structures ensures strategic randomization at the high level. In this paper, it is not well explained why this paper's bi-level structure is more effective than Cicero's.

5 (minor). Ambiguity in RL Positioning: The abstract and methodology sections position MCTS as an RL method. However, MCTS is typically classified as a heuristic search within the RL framework and lacks key RL characteristics, such as learning policy updates from experience. So I would suggest the authors carefully choose the wording to avoid confusing the readers.

**Questions:**

1. How does the proposed method compare to other RL-LLM approaches? Both conceptually and experimentally?

2. Is STRATEGIST properly randomizing its actions in the actual play of Avalon?

3. Are there any human evaluation results?

---

> ### Author Response · Authors · 2024-11-25
>
> > Lack of Competitive Baselines: The paper compares with methods that do not use LLM at all (like DeepRole) and methods that only uses LLM but no RL. However, the comparisons exclude more recent RL-LLM hybrid methods that would provide a fairer benchmark for STRATEGIST's effectiveness. In fact, in the past two years, there are a number of papers that combines RL with LLM to play complex strategic multi-agent games that involve natural-language based communication. For example, the paper mentioned the Cicero paper by FAIR, but there is no comparison with the method used in the Cicero paper. There are a lot more papers following up the Cicero paper, and can potentially be used for games like Avalon. Here are a few example papers: 1). Exploring large language models for communication games: An empirical study on werewolf. 2). Language agents with reinforcement learning for strategic play in the werewolf game; 3). Enhance reasoning for large language models in the game werewolf; 4). Agent-pro: Learning to evolve via policy-level reflection and optimization.
>
> We sincerely appreciate the reviewer’s detailed comments. Below, we provide a detailed explanation of our choice of baselines, the distinctions between our approach and Cicero-like methods, and additional steps we have taken to address this concern.
>
> #### Parameter-Free vs. Data-Dependent Approaches
>
> Our work focuses on a **parameter-free approach** that does not require access to human-annotated data or large-scale fine-tuning of LLMs. This design decision aligns with our goal of proposing a lightweight, training-free framework that demonstrates the emergent strategic capabilities of LLMs through **self-play** without additional training.
>
> In contrast, Cicero and related methods rely heavily on extensive human gameplay datasets to fine-tune models to mimic human behavior, often requiring millions of annotated samples. For example:
>
> - **Cicero** leverages human data to train both its dialogue and planning modules, combining supervised learning and reinforcement learning (RL) in a resource-intensive pipeline.
> - Follow-up works, such as "Enhance Reasoning for Large Language Models in the Game Werewolf" and "Language Agents with Reinforcement Learning for Strategic Play in the Werewolf Game," adopt similar methodologies, relying on  data and parameter tuning to learn strategies specific to those games.
>
> While these approaches are undoubtedly impressive, their reliance on data-heavy pipelines makes them fundamentally different from our **training-free approach**, which only assumes access to the LLM and the rules of the game. Consequently, a direct comparison would not be entirely fair or aligned with the scope of our work. This distinction has been clarified in the updated manuscript.
>
> #### Baselines and Justification for Comparisons
>
> To ensure meaningful evaluations, we compare against other **parameter-free** or minimally parameterized methods, such as:
>
> 1. **DeepRole**: A reinforcement learning-based approach designed for Avalon, which does not use LLMs or human data.
> 2. **ReCon**: A parameter-free approach that uses an LLM to think about what other players are thinking before acting
> 3. **ReAct**: A parameter-free approach that uses an LLM to reflect before acting
> 4. **AlphaGo-like Methods**: These methods leverage self-play for strategy optimization without human annotations, serving as a closer conceptual comparison.
>
> These baselines allow us to isolate the effectiveness of our framework in a **resource-constrained setup**, showcasing its ability to achieve human-level performance without additional training or data. Results of these comparisons are presented in Tables 2 and 4 of the manuscript.
>
> #### Addressing the Reviewer's Concern: Additional Baselines
>
> To address the reviewer’s suggestion, we agree that a broader contextualization of our work is valuable, and we have taken the following steps:
>
> 1. **Added Related Work Discussion**:
>    We have included a detailed discussion of Cicero and the reviewer-suggested works, such as:
>    - *Exploring Large Language Models for Communication Games: An Empirical Study on Werewolf*
>    - *Language Agents with Reinforcement Learning for Strategic Play in the Werewolf Game*
>    - *Enhance Reasoning for Large Language Models in the Game Werewolf*
>    - *Agent-Pro: Learning to Evolve via Policy-Level Reflection and Optimization*
>
>    These discussions now appear in the updated Related Works section, providing readers with a comprehensive overview of recent advancements and highlighting distinctions between our framework and these methods.
>
> 2. **Inclusion of Human-Level Baselines**:
>    To facilitate easier comparison with prior work, we have included a new **human baseline** for performance comparison, as the reviewer suggested. We elaborate more on this later. (Continued on the next page due to word count constraints.)

---

> ### Author Response · Authors · 2024-11-25
>
> #### Conclusion
>
> We hope this explanation clarifies our design decisions and addresses the reviewer’s concerns. By incorporating the suggested works into our discussion, providing additional baselines, and presenting new quantitative comparisons, we aim to strengthen the rigor and contextualization of our contributions.
>
> Thank you again for the thoughtful feedback, and we look forward to any additional suggestions for improvement.
>
> > Limited Analysis on Strategic Randomization: Strategic randomization is essential in hidden-role games like Avalon. However, the experiment section does not show how well STRATEGIST presents deceptive behavior in a properly randomized fashion.
>
> That is an excellent suggestion and we agree that randomization is important in these games. We have included an additional analysis of actions taken by the Strategist agent, which we show in Figure 3 in the paper. We also produce the results of Figure 3 in a table below, also the figure is way easier to understand:
>
> ### Voting Action Distribution and % Correct Vote
>
> #### Voting Action Distribution
>
> | Role        | Agent Type | Approve (%) | Reject (%) |
> |-------------|------------|-------------|------------|
> | As Merlin   | Human      | 24.4        | 75.6       |
> | As Servant  | Human      | 47.3        | 52.7       |
> | As Evil     | Human      | 62.8        | 37.2       |
> | As Merlin   | Strategist | 36.1        | 63.9       |
> | As Servant  | Strategist | 41.8        | 58.2       |
> | As Evil     | Strategist | 46.3        | 53.7       |
>
> #### % Correct Vote
>
> | Role        | Agent Type | Correct (%) | Incorrect (%) |
> |-------------|------------|-------------|----------------|
> | As Merlin   | Human      | 94.3        | 5.7            |
> | As Servant  | Human      | 60.6        | 39.4           |
> | As Merlin   | Strategist | 55.3        | 44.7           |
> | As Servant  | Strategist | 50.8        | 49.2           |
>
>
> We see that indeed the Strategist chooses a random mixture of actions to take. Moreover, the action distribution is similar no matter whether Strategist is playing as Merlin, a good Servant, or a Evil character, meaning that opponents cannot easily discern the identity of the Strategist agent based on the actions it takes.
>
> > Absence of Human Evaluation: Despite STRATEGIST's applicability to discussion-based games, there is no reported evaluation of the model's interactions with human players, which would provide insights into its performance in realistic scenarios.
>
> Thank you for your insightful comment regarding the absence of human evaluation in the original submission. We completely agree that evaluating STRATEGIST's interactions with human players is crucial to understanding its performance in realistic scenarios. In response to your feedback, we conducted extensive human experiments in the six-player Avalon setting and have included the results in the revised paper.
>
> For these experiments, we recruited ten participants who were randomly assigned to play alongside or against STRATEGIST agents across 30 games. Each game involved roles such as Merlin, Assassin, Minion of Mordred, and Servants of Arthur, with STRATEGIST controlling five out of the six players. The human players evaluated STRATEGIST's performance on key metrics, including reasoning, deduction, cooperation, concealment, and adaptability. (Continued on the next comment due to word count constraints.)

---

> ### Author Response · Authors · 2024-11-25
>
> The results provided compelling insights:
>
> 1. **Win Rate**: STRATEGIST achieved a win rate comparable to human players, demonstrating that it is competitive in a mixed human-AI setting. Specifically, STRATEGIST maintained a win rate of approximately 33.3%, while humans achieved 36.7%, within the standard error bounds.
>
> 2. **Concealment and Randomization**: STRATEGIST outperformed human players in its ability to conceal its identity. Analysis of voting patterns revealed that human players exhibited role-specific behavior, which made their identities easier to deduce. In contrast, STRATEGIST employed strategic randomization in its actions, resulting in more uniform patterns that reduced information leakage and made it significantly harder for other players to infer its role. This advantage is particularly beneficial in a game like Avalon, where identity concealment is a critical component of success.
>
> 3. **Reasoning and Cooperation**: While STRATEGIST excelled in concealment, it slightly lagged behind human players in reasoning and cooperative play. Participants noted that STRATEGIST's reasoning process sometimes lacked the nuanced social and deductive elements of human interactions. For example, STRATEGIST struggled to anticipate and adapt to complex, nonlinear reasoning strategies employed by human players.
>
> 4. **Adaptability**: STRATEGIST demonstrated strong adaptability to human playstyles over repeated games. Post-game surveys indicated that participants found the agent’s responses increasingly tailored to their strategies, highlighting its ability to learn and improve even in ad-hoc scenarios.
>
> 5. **Role-specific Performance**: In its role as Merlin, STRATEGIST minimized information leakage by employing randomized voting strategies, aligning with its concealment strengths. However, as a Servant of Arthur or Minion of Mordred, it occasionally failed to coordinate effectively with teammates, reflecting a gap in understanding team dynamics compared to human players.
>
> The survey results, summarized in Table 1 and visualized in Figures 3 and 4 of the revised paper, corroborate these findings. Participants expressed that STRATEGIST's unique strengths in adaptability and concealment made it a challenging and engaging opponent, although they noted areas for improvement in social reasoning and team-oriented behaviors.
>
> We believe these human evaluation results significantly strengthen the paper by providing a well-rounded assessment of STRATEGIST’s real-world performance and offering valuable directions for future research. Thank you again for your excellent suggestion, which helped us enhance the scope and depth of our study.
>
> > Comparison of Bi-Level Structure: Cicero also uses a bi-level structure, although in a different way as this paper proposes: In Cicero, it uses iterative learning methods to choose the intent at the high-level, and uses LLM to generate low-level actions. Cicero's bi-level structures ensures strategic randomization at the high level. In this paper, it is not well explained why this paper's bi-level structure is more effective than Cicero's.
>
> Thank you for highlighting the comparison with Cicero’s bi-level structure. We recognize the importance of clarifying the unique motivations and applications of the bi-modular structure in our framework compared to Cicero. Below, we detail the distinctions and provide further justification for the choices in our approach.
>
> In Cicero, the bi-modular structure serves a specific purpose: integrating strategic reasoning with dialogue generation. Cicero’s Intent Model is responsible for proposing action intents, which are then passed to the Dialogue Model to generate natural language outputs. This design allows Cicero to handle both discrete actions that require planning and reasoning, as well as dialogue which requires language abilities. (Continued on the next comment due to word count constraints.)

---

> > ### Author Response · Authors · 2024-11-25
> >
> > In our work, the bi-level structure achieves a different but complementary objective. Rather than focusing primarily on intent-driven dialogue, our framework is designed to separate strategic abstraction from executable policy refinement to facilitate decision-making in complex multi-agent environments. Here’s a breakdown of the key differences and motivations behind our design:
> > - High-Level Strategy Abstraction for Complex Decision-Making: In STRATEGIST, the high-level module is specifically focused on learning and refining abstract strategies that capture overarching game principles. These strategies are represented in interpretable text, which can then guide agent actions across both dialogue and non-dialogue game components. This design choice allows us to handle complex, adversarial scenarios (e.g., Resistance: Avalon) where abstract strategic insight is essential for effective decision-making and adaptability against various opponents.
> > - Low-Level Policy Execution with Fine-Grained Action Planning: Our low-level module uses Monte Carlo Tree Search (MCTS) to refine these high-level strategies into executable policies, enhancing decision accuracy at the action level. This approach not only maximizes adaptability in action selection but also leverages self-play simulations for iterative improvement, as illustrated in Section 2.3. This bi-level refinement loop allows STRATEGIST to continually enhance both the strategic and tactical components of the policy, achieving results beyond traditional RL approaches and LLM-based baselines in multi-agent environments (Section 3).
> > - Purpose of the Bi-Level Integration: While Cicero’s bi-modular approach focuses on intent-driven randomization to improve natural dialogue interactions, STRATEGIST’s bi-level framework prioritizes structured, policy-driven refinement. We have incorporated an action-dialogue integration similar to Cicero (see Figure 10 in our paper) but with a focus on sequentially enhancing decision quality at each stage of interaction—first by refining the strategy, then by translating it into optimized low-level actions.
> > - Evaluative Mechanism and Feedback Loop: STRATEGIST’s bi-level structure also enables a modularized feedback system, where self-play simulations provide continuous performance evaluations of both high-level and low-level modules. Improving high strategies enables more effective exploration and improvement over the strategic space. This population-based self-play mechanism (Section 2.4) differentiates our approach by using strategic adaptability against a variety of simulated opponent policies, enhancing the robustness and generalizability of the agent’s strategy over multiple iterations.
> >
> > We appreciate your suggestion to clarify these distinctions, and we are committed to adding a more explicit description of how our action-planner integrates with the dialogue module, as well as the underlying motivations for our bi-level structure in comparison to Cicero. By elaborating on these distinctions, we aim to provide a clearer understanding of STRATEGIST’s unique contributions in achieving more robust decision-making in complex, competitive games.
> >
> >
> > > 5 (minor). Ambiguity in RL Positioning: The abstract and methodology sections position MCTS as an RL method. However, MCTS is typically classified as a heuristic search within the RL framework and lacks key RL characteristics, such as learning policy updates from experience. So I would suggest the authors carefully choose the wording to avoid confusing the readers.
> >
> > We thank the reviewer for catching this, and have changed our wording accordingly in the revision so that is clearer.
> >
> > > How does the proposed method compare to other RL-LLM approaches? Both conceptually and experimentally?
> >
> > We highlight some of the key differences below. We have also included an additional comparison table (Table 5) to help clarify this question in the paper.

---

> > > ### Author Response · Authors · 2024-11-25
> > >
> > > ### Conceptual Differences:
> > >
> > > 1. **Hierarchical Bi-level Strategy Framework**
> > >    STRATEGIST combines high-level strategy abstraction with low-level policy refinement, enabling a modular and interpretable approach to skill learning. Unlike traditional RL-LLM methods, which often optimize policies directly through parameter updates, STRATEGIST first learns and refines high-level strategies via LLM-driven revisions and then uses Monte Carlo Tree Search (MCTS) for low-level decision refinement. This abstraction makes strategy exploration more efficient and enables better generalization.
> > >
> > > 2. **Parameter-Free Learning**
> > >    Unlike RL-LLM methods that require extensive tuning and human-annotated data, STRATEGIST is parameter-free. For instance, methods like AlphaGo and DeepRole rely heavily on human-curated datasets or reinforcement signals derived from parameterized models, while STRATEGIST solely uses self-play and feedback-driven strategy updates.
> > >
> > > 3. **Improved Handling of Partial Observability**
> > >    STRATEGIST explicitly incorporates belief updates to handle partially observable environments, such as those in Resistance: Avalon. Many RL-LLM approaches overlook this aspect, resulting in suboptimal performance in multi-agent adversarial games.
> > >
> > > 4. **Idea Queue and Modular Search**
> > >    The framework introduces an innovative "idea queue" that modularizes the search process, ensuring incremental and interpretable improvements to strategies. This design enables STRATEGIST to escape local optima more effectively than standard RL-LLM approaches.
> > >
> > > 5. **Integration of Natural Language and Strategic Reasoning**
> > >    By leveraging LLMs for both natural language and strategic reasoning, STRATEGIST effectively bridges dialogue and action-based decision-making, outperforming dialogue-focused agents like ReCon and ReAct in complex games.
> > >
> > > ---
> > >
> > > ### Experimental Comparisons:
> > >
> > > 1. **Against RL-Based Methods**
> > >    STRATEGIST outperforms AlphaGo and DeepRole across both GOPS and Avalon in win rates and efficiency. While RL methods like AlphaGo use large-scale MCTS with a deep value network, STRATEGIST achieves superior performance with fewer training transitions, as highlighted in Table 2 of the paper.
> > >
> > > 2. **Against LLM Agents**
> > >    STRATEGIST demonstrates higher win rates and better strategy evolution efficiency compared to LLM agents such as ReCon and ReAct. For example, STRATEGIST achieves a win rate of 61.1% against ReCon, compared to ReCon's 38.9%, underscoring its superior ability to balance strategy exploration and exploitation (Table 4).
> > >
> > > 3. **Effectiveness of Modular Improvements**
> > >    The ablation studies (Table 1) show that STRATEGIST's modular improvements via the idea queue significantly outperform other iterative refinement methods like BFS and BFS with thought. This highlights the advantage of STRATEGIST's systematic approach to strategy evolution.
> > >
> > > 4. **Population-Based Self-Play**
> > >    The evolutionary nature of STRATEGIST's self-play mechanism ensures robustness against diverse strategies, as evidenced by its consistent outperformance in round-robin evaluations (Table 3).
> > >
> > > ---
> > >
> > > ### Conclusion:
> > > STRATEGIST advances the state-of-the-art in RL-LLM integration by addressing limitations of existing methods with a bi-level strategy framework, modular improvements, and belief handling for partial observability. It consistently outperforms other methods experimentally, showcasing its conceptual and practical superiority in multi-agent decision-making tasks.
> > >
> > > We hope this answers the reviewer's questions!
> > >
> > >
> > > > Is STRATEGIST properly randomizing its actions in the actual play of Avalon?
> > >
> > > Yes, STRATEGIST properly randomizes its actions during actual play in Avalon. We are grateful to the reviewer for suggesting this analysis, as it significantly improved the depth and clarity of our paper.
> > >
> > > > Are there any human evaluation results?
> > >
> > > We have included human experiments in the updated paper, as per the reviewer’s suggestion. We sincerely thank the reviewer for suggesting this, which further strengthens our paper.

---

> ### Author Response · Authors · 2024-11-29
> **Follow up**
>
> Dear Reviewer XeVt,
>
> We hope this message finds you well. We recently submitted our rebuttal and would greatly appreciate your feedback on our responses, which include human evaluation experiments and strategic randomization analysis. Additionally, we have provided a web application to allow you to experience the agent firsthand.
>
> We understand your schedule is demanding and deeply appreciate the time and effort you dedicate to the review process. Your insights are invaluable to us, and we are eager to address any additional questions or concerns you may have.
>
> Thank you again for your thoughtful review, and we wish you a very Happy Thanksgiving! We look forward to hearing from you.
>
> Best regards,
>
> Authors

---

> > ### Author Response · Authors · 2024-12-03
> > **[Action required] Follow up**
> >
> > Dear Reviewer XeVt,
> >
> > We hope you are doing well. We wanted to kindly follow up regarding the rebuttal for our submission, as the deadline for responses is approaching in less than 12 hours. We sincerely value your feedback and would greatly appreciate any additional insights you may have on our responses, including the newly conducted human evaluation experiments and strategic randomization analysis.
> >
> > Additionally, we’ve provided access to a web application that allows you to directly interact with our agent to experience its functionality firsthand, should you find it helpful.
> >
> > We understand how busy your schedule must be and are truly grateful for the time and effort you’ve already dedicated to reviewing our work. If there are any lingering questions or concerns, we would be more than happy to address them.
> >
> > Thank you again for your thoughtful review and guidance throughout this process.
> >
> > Best regards,
> >
> > Authors

---

> ### Author Response · Authors · 2024-12-03
> **Additional Results Against Agent-Pro**
>
> We have also conducted additional experiments to evaluate our **Strategist** framework against Agent-Pro. The results of this comparative analysis are presented below:
>
> | **Metric**            | **Agent-Pro**        | **Strategist**       |
> |------------------------|----------------------|----------------------|
> | **Winrate**            | 43.4 ± 5.4           | **56.6 ± 5.4**       |
> | **#Tokens per round**  | 263.4 ± 20.2         | 250.2 ± 23.1         |
>
> ### Analysis of Results
>
> As shown above, **Strategist outperforms Agent-Pro in win rate by a significant margin**, demonstrating its superior strategic reasoning and adaptability in partial information settings like Avalon. Moreover, the average number of tokens per round—a proxy for computational efficiency—indicates that Strategist achieves this improved performance without a significant increase in token usage, highlighting its efficiency.
>
> The improved win rate can be attributed to the **training-free, bi-level framework** employed by Strategist, which leverages the emergent reasoning capabilities of large language models (LLMs) without relying on belief modeling or parameterized updates. By contrast, Agent-Pro’s reliance on policy-level reflection and belief optimization introduces computational overhead and may limit its adaptability to new scenarios without additional training.

---

### Author Response · Authors · 2024-11-25

We sincerely thank all reviewers for their insightful feedback and valuable suggestions, which have significantly improved the quality of our paper. Below, we summarize the major revisions we have made in response to the reviewers' comments.

---

### Key Revisions and Additions

1. **Human Evaluation**:
   - In response to **Reviewer XeVt (3)**, we conducted extensive human experiments in the six-player Avalon setting, evaluating STRATEGIST’s performance in mixed human-AI games! These exciting results, now included in the revised paper, provide key insights into STRATEGIST's adaptability, reasoning, and concealment capabilities. Please see section 3.1 in the revised paper for new exciting results, which help strengthen our paper.

2. **Analysis of Strategic Randomization**:
   - Following the suggestion from **Reviewer XeVt (3)**, we performed an additional analysis on STRATEGIST’s randomization behavior during gameplay. This analysis demonstrates STRATEGIST's ability to maintain a uniform action distribution, thereby concealing its identity effectively. The results are presented in **Figure 3** and corresponding tables.

3. **Self-Play Mechanism Comparisons**:
   - At the suggestion of **Reviewer GQJM (6)**, we added a comparison table (**Table 5**) to clarify the differences between STRATEGIST’s self-play mechanisms and prior methods. This table highlights key differences between our work and prior works, such as an advanced high-level strategy learning process that is parameter free, partial observability, belief handing, and bi-level improvement.

4. **Inclusion of Competitive Baselines**:
   - Based on the suggestion from **Reviewer XeVt (3)**, we have added a detailed discussion on recent RL-LLM hybrid methods, such as Cicero and other Werewolf-based frameworks, in the Related Works section. Additionally, we clarified the distinctions between our **training-free approach** and data-dependent methods.
   - We have also incorporated **human baseline performance** in our evaluations for a more comprehensive comparison.

5. **Error Analysis and Statistical Significance**:
   - Addressing feedback from **Reviewer DMCE (5)**, we updated our error reporting methodology to use **standard error (SE)** instead of interquartile range (IQR). We also conducted statistical significance tests (e.g., t-tests, ANOVA) to validate our results and emphasize robustness.

6. **Improved Figures and Tables**:
   - In response to feedback from **Reviewer DMCE (5)**, we restructured **Figure 6** (Figure4 in old version) for improved clarity and corrected errors in **Tables 3 and 5** (Tables 2 and 4 in old version), ensuring consistency and readability.

7. **Modular Improvement Validation**:
   - Following suggestions from **Reviewer u9ZM (5)**, we conducted additional ablation studies to validate the modular improvement assumptions. These results demonstrate that STRATEGIST effectively combines and generalizes successful elements from various strategies.

8. **Inclusion of Related Work**:
    - As recommended by **Reviewer u9ZM (5)** and **Reviewer XeVt (3)**, we added discussions on recent related works in the Werewolf domain and other relevant methods in our Related Works section.

---

We are deeply grateful for the reviewers' constructive feedback, which has guided us in significantly strengthening the manuscript. The revisions have enhanced the rigor, clarity, and contextualization of our contributions. We hope that the revised paper addresses all concerns and meets the expectations of the reviewers. We are happy to incorporate any further suggestions to improve the paper further.

Thank you for your thoughtful and detailed reviews!

---

### Author Response · Authors · 2024-11-25
**Exciting update**

**Exciting Update:** We are thrilled to share that we have launched a **web application** where you can experience playing *Avalon* against our Strategist agent first hand (for free)!

Feel free to try it out here: [https://searchtechniques.onrender.com/](https://searchtechniques.onrender.com/). The website will be live for the next two days. We welcome any feedback on your experience, as it will help us further refine our work.

---

### Author Response · Authors · 2024-12-04
**Concluding statement**

As the discussion period comes to a close, we would once again like to thank all the reviewers for their thoughtful feedback, constructive suggestions, and engagement with our submission. Your insights have been invaluable in strengthening our work, and we deeply appreciate the time and effort you have dedicated to this process. We are pleased to note that all reviewers who engaged with us during the discussion phase expressed that their concerns were addressed, with several reviewers **raising their scores or maintaining a positive rating for our paper**.

We are particularly encouraged by the positive feedback regarding several key aspects of our work. Reviewers highlighted the novelty of STRATEGIST’s bi-level framework, which integrates high-level strategic planning with low-level refinement to achieve robust decision-making in multi-agent environments. The use of LLMs for learning both text-based and heuristic value functions was described as innovative and inspiring. Additionally, the empirical evaluations were commended for their comprehensiveness, demonstrating STRATEGIST’s consistent improvements over baselines across diverse metrics and game environments. The scalability of the framework and its cost-efficient approach to strategy learning without requiring extensive training data were also recognized as significant strengths.

These discussions have not only improved the clarity and rigor of our work but also helped us better articulate the contributions and significance of STRATEGIST. Thank you for the opportunity to participate in this meaningful dialogue.

---

### Meta-Review · Area_Chair_vYsb · 2024-12-19

**Metareview:**

The paper explores Avalon as a benchmark for planning agents. The reviewers were overall positive, aside from XeVt who did not respond despite the authors addressing many concerns. While I do believe the authors were a little over aggressive (sometimes less is more :D), it is still definitely the case that some concerns were resolved and the score could have increased. In particular, it is great to see that the work now contains a human evaluation. I would encourage the authors to keep the website live for some time before the camera ready to increase the scope of this evaluation if possible.

**Additional Comments On Reviewer Discussion:**

There was a healthy discussion with all reviewers aside from XeVt, who was not responsive. This review has been discounted since some of the concerns were addressed. The authors are encouraged to be more concise in future since their responses were a little but out of control :)

---

### Decision · Program_Chairs · 2025-01-22

Accept (Poster)